Resource

# Demographic and genetic factors shape the epitope specificity of the human antibody repertoire against viruses

Axel Olin [1,2], Christian Pou[3], Anthony Jaquaniello[1,4], Jack Crook[5], Ziyang Tan[3], Maguelonne Roux[1,6,61], Florian Dubois[7,8], Bruno Charbit[7,8], Dang Liu [1], Françoise Donnadieu[9], Laura Garcia [9], Camille Lambert [9], Emma Bloch[9], Emmanuel Clave [10], Itauá Leston Araujo[10], Antoine Toubert [10], Maxime Rotival [1], Etienne Simon-Lorière [5], Michael White[9], Petter Brodin [3,11], Darragh Duffy [7,8], Lluis Quintana-Murci [1,12,60], Etienne Patin [1,60] & on behalf of the Milieu Intérieur Consortium*

Antibodies are central to immune defenses. Despite advances in understanding the mechanisms of antibody generation, a comprehensive model of how intrinsic and external factors shape human humoral responses to viruses has been lacking. Here we apply phage immunoprecipitation sequencing to investigate the effects of demographic factors—including 108 lifestyle and health-related variables—and genetic variation on antibody reactivity to over 97,000 viral peptides in 1,212 healthy adults. We demonstrate that age, sex and continent of birth extensively affect not only the viruses but also the specific viral epitopes targeted by the antibody repertoire. Notably, we find that antibodies against rapidly evolving epitopes of influenza A virus decrease with age, whereas immunoreactivity to conserved epitopes increases. Furthermore, we identify strong associations between antibodies against 34 viruses and genetic variants at *HLA*, *FUT2*, *IGH* and *IGK* loci, some of which increase autoimmune disease risk. These findings offer a valuable resource for understanding the factors affecting antibody-mediated immunity, laying the groundwork for optimizing vaccine strategies.

Antibodies are central effectors of adaptive immunity and serve as correlates of protection following natural infection or vaccination. The large inter-individual variability in antibody repertoires indicates that antibody production and maintenance are shaped by multiple factors. Family- and population-based studies have revealed marked differences in antibody titers according to sex and age. For example, women exhibit higher titers against human papillomavirus[1] and Epstein–Barr virus (EBV)[1,2] and mount stronger vaccine responses than men[3]. Furthermore, antibodies against persistent herpesviruses, such as herpes simplex virus 1 (HSV-1) and cytomegalovirus (CMV), tend to increase with age, reflecting cumulative exposure[1,2,4,5], whereas antibodies against viruses that primarily infect children (for example, respiratory syncytial virus (RSV) and varicella–zoster virus (VZV)) typically persist at high levels into adulthood[1,2]. Additional non-genetic factors associated with antibody levels include socioeconomic status[1,2] and smoking[4].

Human genetic factors also contribute to variation in antibody responses. Total and virus-specific antibody titers against CMV, EBV and influenza A virus (IAV) have been shown to be heritable[2,6]. At the genome-wide scale, the *HLA* locus presents strong associations with antibody titers against numerous viruses, such as EBV, IAV, rubella virus and VZV[2,7–10]. Other loci, including *IGH*, *FUT2*, *STING1* and *MUC1*, have been associated with responses to IAV, norovirus and polyomaviruses[4,11].

A full list of affiliations appears at the end of the paper. e-mail: quintana@pasteur.fr; epatin@pasteur.fr

Despite these advancements, most studies have focused on a limited number of viruses, hindering a comprehensive understanding of the determinants of humoral immunity across the broad spectrum of viruses infecting humans. Furthermore, although antibodies targeting a single virus can recognize numerous epitopes—the portion of an antigen bound by the immune system—inter-individual variation in epitope reactivity remains poorly characterized, leaving the determinants of viral antigenic specificity largely unknown.

In this Resource, we address these questions using phage immunoprecipitation sequencing (PhIP-seq), a high-throughput approach for assessing antibody–epitope interactions[12,13]. PhIP-seq has been applied to characterize antibody repertoire changes across diseases[5,14] and evaluate humoral immunity to bacteria[4,15]. A virus-focused implementation, VirScan[16], which spans the complete peptidome of all known human viruses, has recently allowed investigation of the impact of measles infection on antibody profiles[17] and immune development in neonates[18]. Using the VirScan library, we profiled 97,978 viral peptides in 1,212 healthy adults and integrated these data with extensive demographic and genetic information. This approach enabled us to characterize differences in the viruses, viral proteins and epitopes targeted by individual antibody repertoires and to identify key factors shaping both the breath and epitope specificity of antiviral humoral immunity.

## Results

### Extensive diversity in the antiviral antibody repertoire of healthy adults

To assess the viral peptidome-wide antibody repertoire, we performed PhIP-seq on 900 plasma samples from the Milieu Intérieur cohort[19], comprising individuals of European ancestry with balanced sex and age distribution (20–69 years; Fig. 1a). To validate our findings and explore population-level differences in humoral immunity, we analyzed an additional 312 samples from the EvoImmunoPop (EIP) cohort[20], including 100 and 212 Belgian residents born in Central Africa (AFB) and Europe (EUB), respectively—all male, aged 20–50 years (Fig. 1b). For both cohorts, we used the VirScan V3 library, encompassing 115,753 microbial peptides[16]. After filtering for unique viral sequences, we retained 97,978 peptides, representing a wide range of viral families and species (Extended Data Fig. 1a,b). PhIP-seq read counts for each viral peptide were converted into standardized, batch-corrected $Z$ scores (Extended Data Fig. 2 and Supplementary Note), which have been shown to correlate strongly with antibody titers[16].

The total numbers of positive peptides per individual were normally distributed (Fig. 1c,d,f,g), averaging 881 and 1,044 peptides for Milieu Intérieur and EIP individuals, respectively, due to differences in cohort demographics or sampling protocols. Approximately 97% of peptides were positive in <5% of individuals, reflecting individual-specific immunity (denoted private peptides) or false positives[4,15] (Fig. 1e,h). We therefore conducted all subsequent analyses on peptides positive in >5% of individuals, requiring at least two peptides to be positive per virus (denoted public peptides). This yielded 2,608 and 3,210 public peptides in Milieu Intérieur and EIP individuals, respectively, originating from 113 viral species, with EBV, IAV and enterovirus B being the most prevalent in both cohorts (Extended Data Fig. 1c).

Given the high sequence identity among some of the peptides tested, cross-reactivity can lead to false-positive signals. To address this, we used the antiviral antibody response deconvolution algorithm (AVARDA), which estimates the breadth of antibody responses for each viral species—defined as the largest number of reactive viral peptides that show low sequence identity—while accounting for cross-reactivity and the unbalanced peptide representation in the PhIP-seq library[21]. As expected, seroprevalence determined by AVARDA breadth scores was highest for common viruses such as EBV, HSV-1, CMV, rhinovirus B and adenovirus C (Fig. 1i). We validated the resolution, sensitivity and serostatus prediction accuracy of both peptide-level $Z$ scores and

virus-level AVARDA breadth scores through extensive comparisons with enzyme-linked immunosorbent assays (ELISA) and Luminex assays (Extended Data Figs. 3 and 4, Supplementary Note and Supplementary Tables 1 and 2). Together, these analyses underscore the specificity and sensitivity of PhIP-seq and reveal the remarkable diversity of human antibody repertoires against viruses causing common infections.

### Age and sex affect the breadth and epitope specificity of the antibody repertoire

We investigated the effects of non-genetic and genetic factors on antiviral humoral immunity by systematically testing their associations with both peptide-level $Z$ scores and virus-level AVARDA breadth scores. This combined approach allowed us to capture inter-individual variation in antigenic specificity and to identify associations that can be missed at the virus level while accounting for between-species cross-reactivity (Supplementary Note). Before conducting the main analyses, we verified whether antibody reactivity scores were associated with the numbers of different B cell subsets in blood or with B cell output from bone marrow (Supplementary Table 3), which could mediate the effects of age or smoking on humoral responses[22]. No such associations were detected (adjusted $P$ value ($P_{adj}$) > 0.05; Extended Data Fig. 5 and Supplementary Note), suggesting that the factors examined below act directly on antibody levels.

We first examined the effects of age and sex on the antiviral antibody repertoire in the Milieu Intérieur cohort. As no significant nonlinear effects of age or age × sex interactions were observed, we focused only on linear and additive effects (Methods). Linear regression modeling revealed that age is strongly associated with antibody reactivity against a broad range of viruses (Fig. 2a), consistent with previous studies[4,5]. Antibodies against 565 peptides increased with age (false-discovery-rate-adjusted $P$ < 0.05; Supplementary Table 4), primarily from the herpesviruses HSV-1, HSV-2 and EBV, which can reactivate throughout life[23]. These associations were robust to cross-reactivity, as supported by AVARDA (Extended Data Fig. 6a), and were replicated in the EIP cohort for HSV-1 and EBV (Extended Data Fig. 6b–d). The strongest age effects were observed for antibodies targeting HSV-1 envelope glycoprotein D (Extended Data Fig. 6e) and EBV nuclear antigens 3, 4 and 6 (EBNA-3, EBNA-4 and EBNA-6) (Extended Data Fig. 6f). Both peptide-level $Z$ scores and AVARDA breadth scores also showed positive associations with age for hepatitis A virus and Aichi virus A (Fig. 2a and Extended Data Fig. 6a), the latter of which is a kobuvirus that was initially isolated during a 1989 gastroenteritis outbreak in Japan and has subsequently been detected in Europe[24]. Conversely, antibodies against 766 peptides decreased with age, primarily from rhinoviruses, enteroviruses and adenoviruses ($P_{adj}$ < 0.05; Fig. 2a). After accounting for cross-reactivity, AVARDA confirmed age-related decreases for antibodies against rhinoviruses A and B, enteroviruses B and C and adenovirus D (Extended Data Fig. 6a), suggesting higher exposure in younger individuals and/or faster antibody waning in older adults.

Notably, antibodies against different IAV peptides either strongly increase or decrease with age (Fig. 2a,b). In younger individuals, antibodies primarily target amino acid positions 1–100 and 300–400 of hemagglutinin (HA) (for example, peptide 1: $\beta_{VirScan} = -0.029$ and $P_{VirScan} = 1.48 \times 10^{-19}$; Fig. 2c,f,g), corresponding to the more variable globular head domain[25] (Fig. 2d). By contrast, in older individuals, antibodies preferentially target positions 450–550 within the more conserved stalk domain (for example, peptide 3: $\beta_{VirScan} = 0.023$ and $P_{VirScan} = 1.11 \times 10^{-21}$; Fig. 2c,d,f,g). Antibodies showing the greatest decrease with age tend to target the most variable HA peptides (Spearman's $\rho = -0.53$ and $P = 7.4 \times 10^{-5}$; Fig. 2e), suggesting that antibodies against rapidly evolving viral epitopes wane faster than those targeting more evolutionarily stable epitopes. These observations were validated with Luminex immunoassays (H3N2 head: $\beta_{Luminex} = -0.033$ and $P_{Luminex} = 4.42 \times 10^{-76}$; H1N1 stalk: $\beta_{Luminex} = 0.014$

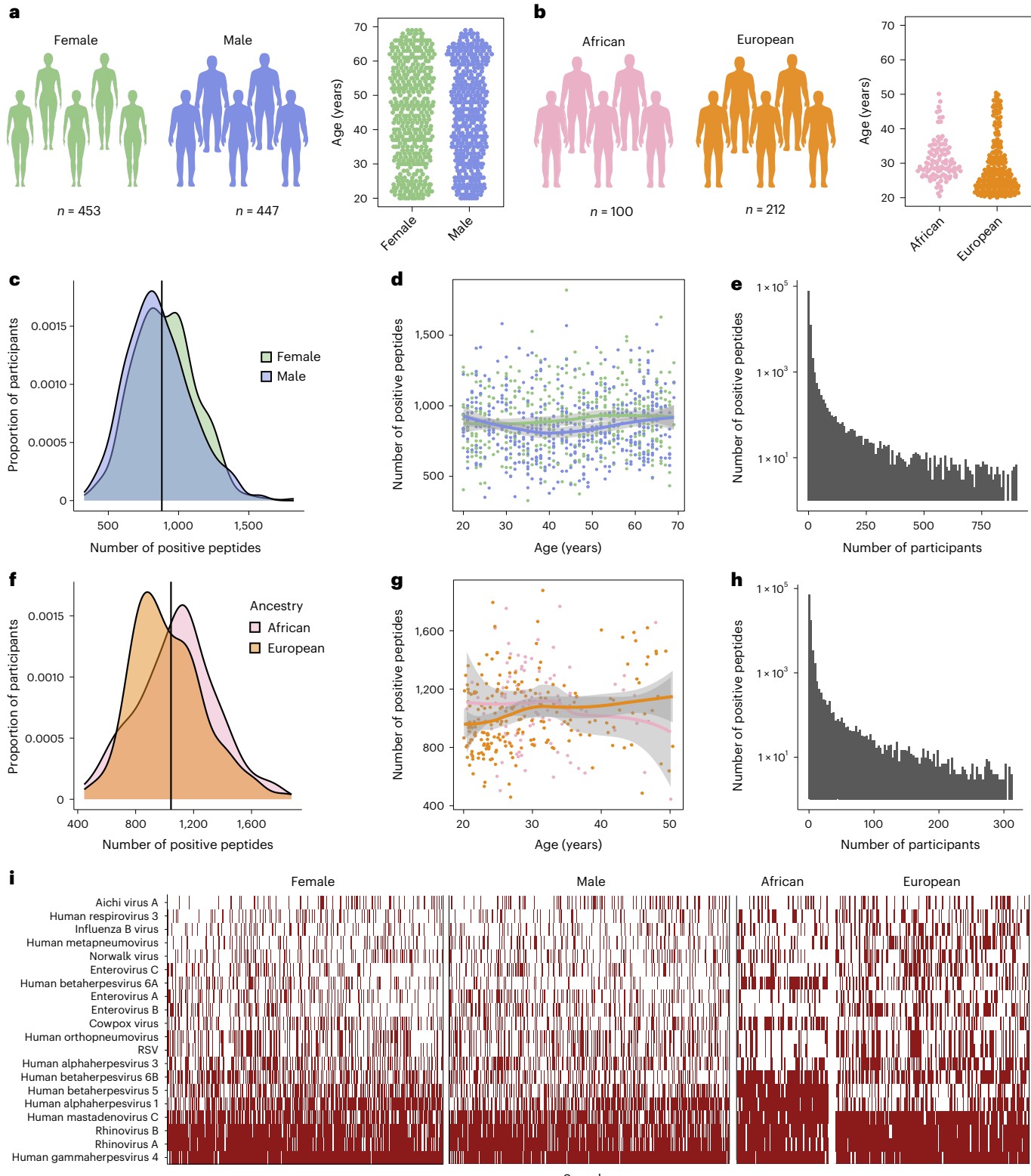

**Fig. 1 | Assessing antibody repertoire variation in the Milieu Intérieur and EIP cohorts. a**, Sample sizes and age distribution by sex within the Milieu Intérieur cohort. **b**, Sample sizes and age distribution by continent of birth within the EIP cohort. **c**, Density distributions of Milieu Intérieur donors as a function of the number of peptides they react against, categorized by sex. The black vertical line indicates the average number of positive peptides in the entire cohort. **d**, Number of positive peptides per Milieu Intérieur donor as a function of age and sex. **e**, Number of positive peptides as a function of the number of Milieu Intérieur donors. **f**, Density distributions of EIP donors as a function of the number of peptides they react against, categorized by continent of birth. The black vertical line indicates the average number of positive peptides in the entire cohort. **g**, Number of positive peptides per EIP donor as a function of age and continent of birth. **h**, Number of positive peptides as a function of the number of EIP donors. **i**, Heatmap indicating the predicted infection status of each Milieu Intérieur and EIP donor for the 20 most prevalent viruses, as determined by AVARDA ($P_{adj} < 0.05$ after Benjamini–Hochberg correction). In **d** and **g**, the solid curve and shaded area indicate the locally estimated scatterplot smoothing curve and 95% confidence interval, respectively.

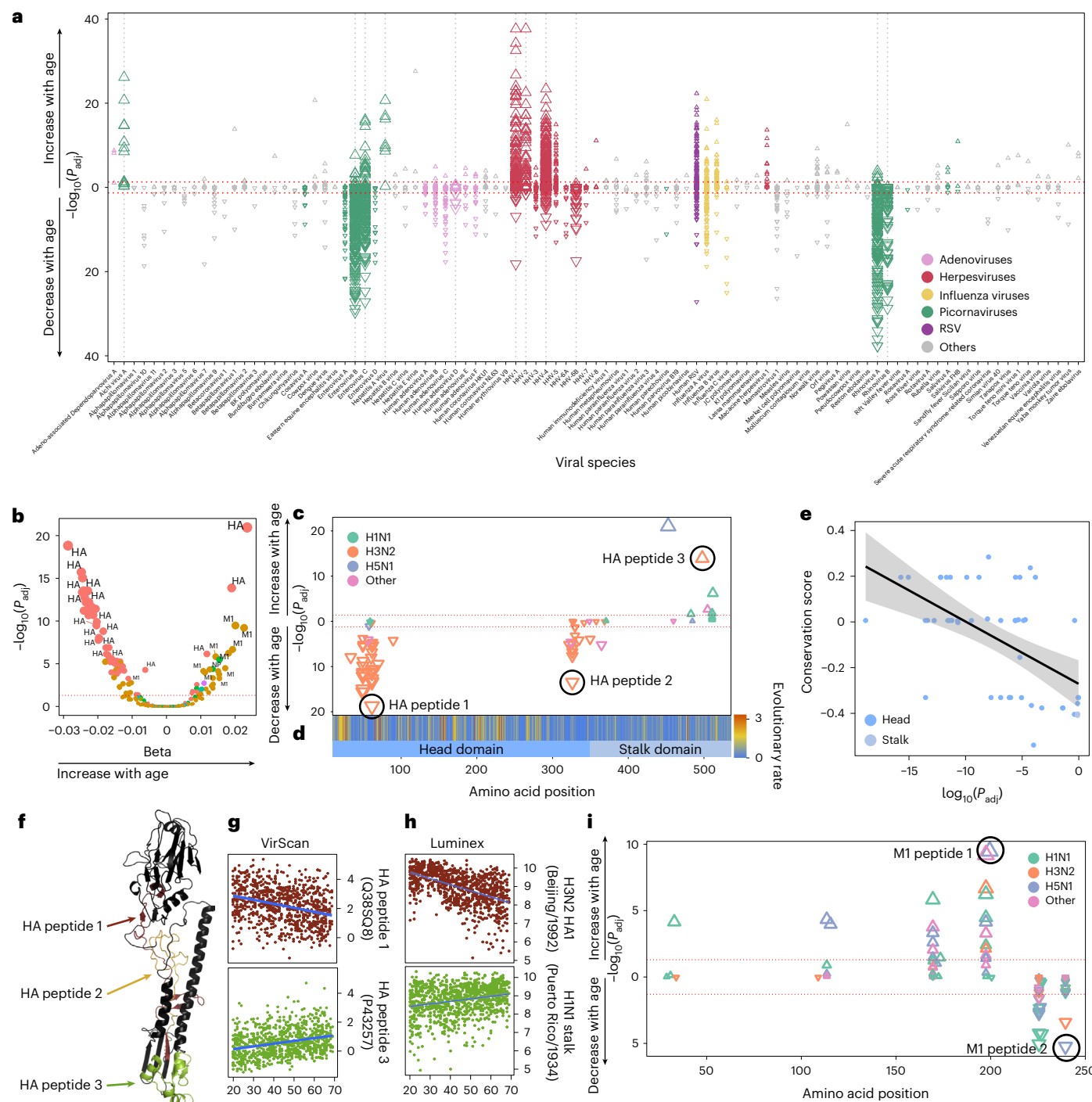

**Fig. 2 | Age impacts the epitope-specific antiviral antibody repertoire.**
**a**, $-\log_{10}(P_{adj})$ values and the directions of associations between all public peptide $Z$ scores and age in the Milieu Intérieur cohort, by viral species (two-sided Wald test). The dotted gray vertical lines indicate viruses for which the AVARDA breadth score is significantly associated with age. The dotted red horizontal lines indicate the significance threshold ($P_{adj} < 0.05$). **b**, $-\log_{10}(P_{adj})$ values against effect sizes of associations between IAV peptide $Z$ scores and age in the Milieu Intérieur cohort, colored by viral protein (two-sided Wald test). **c**, Amino acid positions of the midpoint of public HA peptides associated with age within the full IAV (HA) protein. Significance and the directions of associations with age are indicated on the $y$ axis and by the direction of triangles, respectively. **d**, Per-residue evolutionary rates of HA, computed from all H3 viral sequences sampled between 1975 and 2012. Higher values (in red) indicate higher viral evolutionary rates. **e**, HA peptide evolutionary rates as a function of $-\log_{10}(P_{adj})$ values for

negative associations between antibody reactivity and age (two-sided Wald test). The black line and shaded area indicate the regression line and 95% confidence interval, respectively. **f**, Locations of the peptides of interest indicated in **c** within the three-dimensional structure of HA (PDB ID: 4N5Y). **g**, PhIP-seq-based $Z$ score as a function of age for HA peptides 1 and 3, highlighted in **c**. **h**, Luminex-based antibody titers targeting globular head (H3N2 HA1 Beijing/1992) and stalk (H1N1 Puerto Rico/1934) HA domains as a function of age, replicating the results shown in **g**. The blue line indicates the regression line. **i**, Amino acid positions of the midpoint of public M1 peptides associated with age within the full IAV M1 protein for the Milieu Intérieur cohort. In **c** and **i**, significance and the directions of associations with age are indicated on the $y$ axis and by the direction of triangles, respectively. The triangle color indicates the IAV subtype. The most significant peptides for each epitope are circled and labeled.

and $P_{Luminex} = 8.62 \times 10^{-15}$; Fig. 2h). A similar pattern was observed for the IAV matrix protein 1 (M1) (Fig. 2b,i): antibodies against less conserved positions (200–250) decrease with age ($\beta_{VirScan} = -0.018$; $P_{VirScan} = 5.90 \times 10^{-6}$; $\beta_{Luminex} = -0.0068$; $P_{Luminex} = 4.04 \times 10^{-4}$), whereas antibodies targeting more conserved positions (150–200) increase ($\beta_{VirScan} = 0.02$; $P_{VirScan} = 3.33 \times 10^{-10}$). These effects result in a significant relationship between M1 evolutionary rates and age-related changes in immunoreactivity (Spearman's $\rho = -0.70$; $P = 1.8 \times 10^{-4}$).

To determine whether these peptide-specific effects were driven by age-related variations in exposure to different IAV strains, we compared antibodies targeting protein domains from the same viral strain. Consistent opposing age effects were observed for the HA head and stalk domains from the same H3N2 strain, particularly in individuals over 40 years of age (A/Victoria/3/1975 HA head: $\beta_{Luminex} = -0.035$ and $P_{Luminex} = 3.50 \times 10^{-21}$; stalk: $\beta_{VirScan} = 0.019$ and $P_{VirScan} = 1.34 \times 10^{-14}$) and for the same H1N1 strain (A/Puerto Rico/8/1934 HA head: $\beta_{Luminex} = -0.012$ and $P_{Luminex} = 0.013$; stalk: $\beta_{Luminex} = 0.014$ and $P_{Luminex} = 8.62 \times 10^{-15}$), as well as for M1 domains from the same H1N1 strain (A/Jamesburg/1942 150–200 region: $\beta_{VirScan} = 0.013$ and $P_{VirScan} = 2.4 \times 10^{-5}$; 200–250 region: $\beta_{VirScan} = -0.015$ and $P_{VirScan} = 2.4 \times 10^{-7}$). Furthermore, although past influenza vaccination was associated with higher total anti-IAV antibody titers in the Milieu Intérieur cohort ($\beta = 0.34$; $P = 2.62 \times 10^{-14}$), vaccination was only weakly associated with age (odds ratio = 0.012; $P = 0.048$), supporting the view that vaccination is unlikely to account for the observed age-related patterns. Notably, the AVARDA breadth score for IAV was not associated with age (Extended Data Fig. 6a), as it aggregates peptides with opposing age effects. Together, these findings indicate that epitope specificity of anti-IAV humoral responses varies with age.

Sex effects were more modest than age effects: 330 peptides showed higher antibody levels in women and 236 peptides showed higher antibody levels in men ($P_{adj} < 0.05$; Extended Data Fig. 7a and Supplementary Table 4). Accounting for cross-reactivity, AVARDA supported higher reactivity in women for antibodies against CMV, human herpesvirus 6A (HHV-6A) and HHV-6B (Extended Data Fig. 7b). These results suggest higher exposure and/or stronger humoral responses to herpesviruses in women, in contrast with bacterial antibody responses that show no sex differences[4]. As with age, we observed sex-specific antibody reactivity to different proteins of influenza viruses (Extended Data Fig. 7c,d): women preferentially target the HA protein ($\beta_{VirScan} = -0.71$; $P_{VirScan} = 1.26 \times 10^{-13}$; $\beta_{Luminex} = -0.10$; $P_{Luminex} = 0.038$), whereas men showed higher reactivity to M1 ($\beta_{VirScan} = 0.71$; $P_{VirScan} = 2.10 \times 10^{-14}$; $\beta_{Luminex} = 0.092$; $P_{Luminex} = 0.02$) and NP ($\beta_{VirScan} = 0.60$; $P_{VirScan} = 3.55 \times 10^{-10}$; $\beta_{Luminex} = 0.10$; $P_{Luminex} = 0.03$) proteins from IAV and IBV, respectively. For comparison, we inferred serostatus for M1 from H3N2 and found that 10.0% of women and 16.1% of men were seropositive (odds ratio = 1.72; 95% confidence interval = 1.16–2.57; $P_{VirScan} = 0.0077$), consistent with previous findings[26]. As influenza vaccination rates did not differ between women and men in the Milieu Intérieur cohort (20.2 versus 18.6%, respectively; $P = 0.51$) and vaccines do not typically target the M1 protein, these findings point to intrinsic sex differences in humoral responses to influenza viruses.

## Antibody profiles differ according to continent of origin

To investigate how the antiviral antibody repertoire varies between populations, we leveraged the EIP cohort, comprising individuals born in Central Africa (AFB) or Europe (EUB). Although all samples were collected in Belgium, AFB individuals had relocated to Europe shortly before sample collection (2.45 years before, on average[27]), implying that differences compared with EUB probably result from previous viral exposures and/or genetic ancestry. Antibody levels against 898 viral peptides were increased in EUB, predominantly from rhinoviruses, adenoviruses and IAV ($P_{adj} < 0.05$; Fig. 3a), although the significance was weak when considering AVARDA scores ($P_{adj} > 0.001$). By contrast, higher antibody reactivity in AFB was observed for 647 peptides, of which 61% were derived from herpesviruses. Elevated reactivity in AFB was

strongly supported by AVARDA for CMV ($\beta = -14.71$; $P_{adj} = 1.29 \times 10^{-19}$), HHV-6A ($\beta = -7.18$; $P_{adj} = 6.18 \times 10^{-17}$), HHV-6B ($\beta = -5.49$; $P_{adj} = 1.34 \times 10^{-10}$) and HHV-8 ($\beta = -8.44$; $P_{adj} = 6.93 \times 10^{-20}$; Extended Data Fig. 8a–c), consistent with previous reports[20,28,29]. Adjusting the statistical models for genetic determinants of the antiviral antibody repertoire (Supplementary Table 5) had minimal impact on the differences observed between AFB and EUB (Extended Data Fig. 8d), supporting differential viral exposure—rather than genetic ancestry—as the primary driver of population-level variation in antibody repertoires.

Population differences were also evident at the epitope level within individual viral species. Despite similar overall EBV reactivity between populations ($P_{adj} > 0.05$; Extended Data Fig. 8a), AFB and EUB targeted different EBV peptides (Fig. 3a,b,f). Antibodies from AFB more frequently targeted the EBNA-4 viral protein, whereas those from EUB preferentially targeted EBNA-6. The four EBNA-4 peptides most associated with African origin clustered between amino acid positions 600 and 800 and derived from the AG876 strain, a type 2 EBV strain that is prevalent in Africa[30] (Fig. 3b–e). Conversely, EBNA-6 peptides associated with European origin localized to positions 750–850 and derived from the GD1 and B95-8 cosmopolitan strains (Fig. 3f–i). These findings, validated by Luminex immunoassays for both EBNA-4 ($\beta = -1.28$; $P = 9.12 \times 10^{-14}$; Fig. 3d) and EBNA-6 ($\beta = 1.02$; $P = 8.2 \times 10^{-13}$; Fig. 3h), suggest that population-specific epitope targeting reflects past exposure to distinct EBV strains. Similarly, antibodies against IAV from AFB primarily targeted NP from H1N1, whereas those from EUB favored HA from H3N2 (Extended Data Fig. 8e–g). Collectively, these results reveal population disparities in antibody reactivity against epitopes of common viruses, highlighting the limitations of single-antigen assays to assess seroprevalence in global epidemiological studies.

## Smoking exerts reversible effects on antibody reactivity against rhinoviruses

To identify new non-genetic factors affecting antiviral antibody repertoires, we searched for associations in the Milieu Intérieur cohort with 108 variables related to socioeconomic status, health-related habits, medical history and disease biomarkers (Supplementary Table 3) while controlling for age and sex. Besides weak associations with socioeconomic status and health biomarkers (Fig. 4a, Supplementary Table 4 and Supplementary Note), the only strongly significant associations were found for tobacco smoking, correlating with antibodies against 134 peptides ($P_{adj} < 0.05$; Fig. 4a,b) primarily derived from rhinoviruses A and B and enteroviruses A–D. AVARDA confirmed the association between cigarette consumption and antibodies targeting rhinoviruses A and B ($\beta = 0.032$; $P_{adj} = 1.99 \times 10^{-4}$). Rhinoviruses are a prevalent cause of the common cold, which occurs more frequently and with greater severity in smokers, although the underlying mechanisms remain debated[31,32].

These associations with smoking were not reproduced using immunoassays based on whole-rhinovirus lysates (Supplementary Table 1), suggesting that peptide-specific effects are masked when the full rhinovirus peptidome is assessed. Consistently, antibody reactivity was associated with smoking status for only 29% (63 out of 218) of the rhinovirus peptides included in the VirScan library. However, the five peptides most strongly associated with smoking—derived from a rhinovirus polyprotein containing capsid proteins ($\beta > 0.040$; $P_{adj} < 1.0 \times 10^{-10}$; Fig. 4c)—included the peptide showing the strongest smoking association in the LifeLines-DEEP cohort ($n = 1,443$; $\beta = 1.09$; $P_{adj} = 3.29 \times 10^{-9}$) using a custom PhIP-seq library[4], validating that these associations are both genuine and peptide specific. We found that anti-rhinovirus B reactivity was not associated with smoking duration in active smokers ($\beta = 0.012$; $P = 0.383$; Fig. 4d), suggesting constant, non-cumulative exposure to rhinoviruses. Interestingly, ex-smokers exhibited antibody levels comparable to those of people who had never smoked ($\beta = -0.17$; $P = 0.084$; Fig. 4c). Accordingly, anti-rhinovirus B antibodies decreased with years after quitting smoking in former smokers ($\beta = -0.022$; $P = 8.74 \times 10^{-3}$; Fig. 4e). These findings collectively

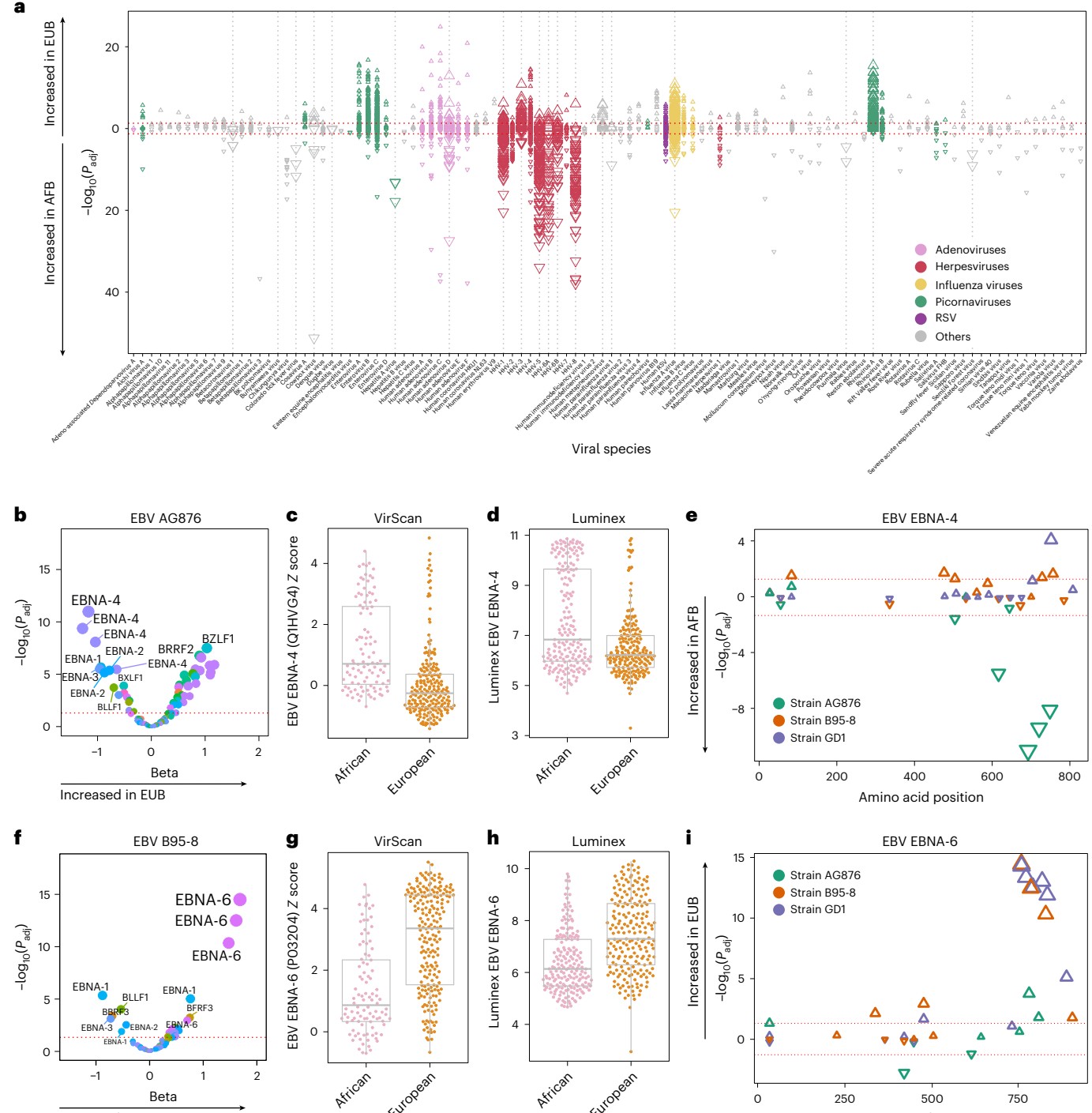

**Fig. 3 | Antiviral antibody repertoire in relation to continent of birth.**
**a**, $-\log_{10}(P_{adj})$ values and the directions of associations between all public peptide $Z$ scores and continent of birth in the EIP cohort, separated by viral species (two-sided Wald test). The dashed gray vertical lines indicate viruses for which the AVARDA breadth score is significantly associated with continent of birth. The dashed red horizontal lines indicate the significance threshold ($P_{adj} < 0.05$). **b**, $-\log_{10}(P_{adj})$ values against effect sizes of associations between continent of birth and peptide $Z$ scores from the EBV AG876 strain in the EIP cohort (two-sided Wald test). Colors indicate the viral protein. **c,d**, PhIP-seq reactivity scores (**c**) and Luminex-based titers (**d**) for antibodies targeting the most significant EBNA-4 peptide (UniProt ID: Q1HVG4) from the EBV AG876 strain, in AFB and EUB separately. Horizontal lines, box edges and whiskers indicate the median value, interquartile range and 1.5× the interquartile range, respectively. **e**, Amino acid positions of the midpoint of public EBNA-4 peptides associated with continent of

birth within the full EBV EBNA-4 protein for the EIP cohort (two-sided Wald test). Significance and the directions of associations with age are indicated on the $y$ axis and by the direction of triangles, respectively. Triangle colors indicate the EBV strain. **f**, $-\log_{10}(P_{adj})$ values against effect sizes of associations between continent of birth and peptide $Z$ scores from the EBV B95-8 strain in the EIP cohort (two-sided Wald test). Colors indicate the viral protein. **g,h**, PhIP-seq reactivity scores (**g**) and Luminex-based titers (**h**) for antibodies targeting the most significant EBNA-6 peptide (UniProt ID: P03204) from EBV B95-8, in AFB and EUB separately. Horizontal lines, box edges and whiskers indicate the median value, interquartile range and 1.5× the interquartile range, respectively. **i**, Amino acid positions of the midpoint of all public EBNA-6 peptides associated with continent of birth within the full EBV EBNA-6 protein for the EIP cohort (two-sided Wald test). Significance and the directions of associations with age are indicated on the $y$ axis and by the direction of triangles, respectively. Triangle colors indicate the EBV strain.

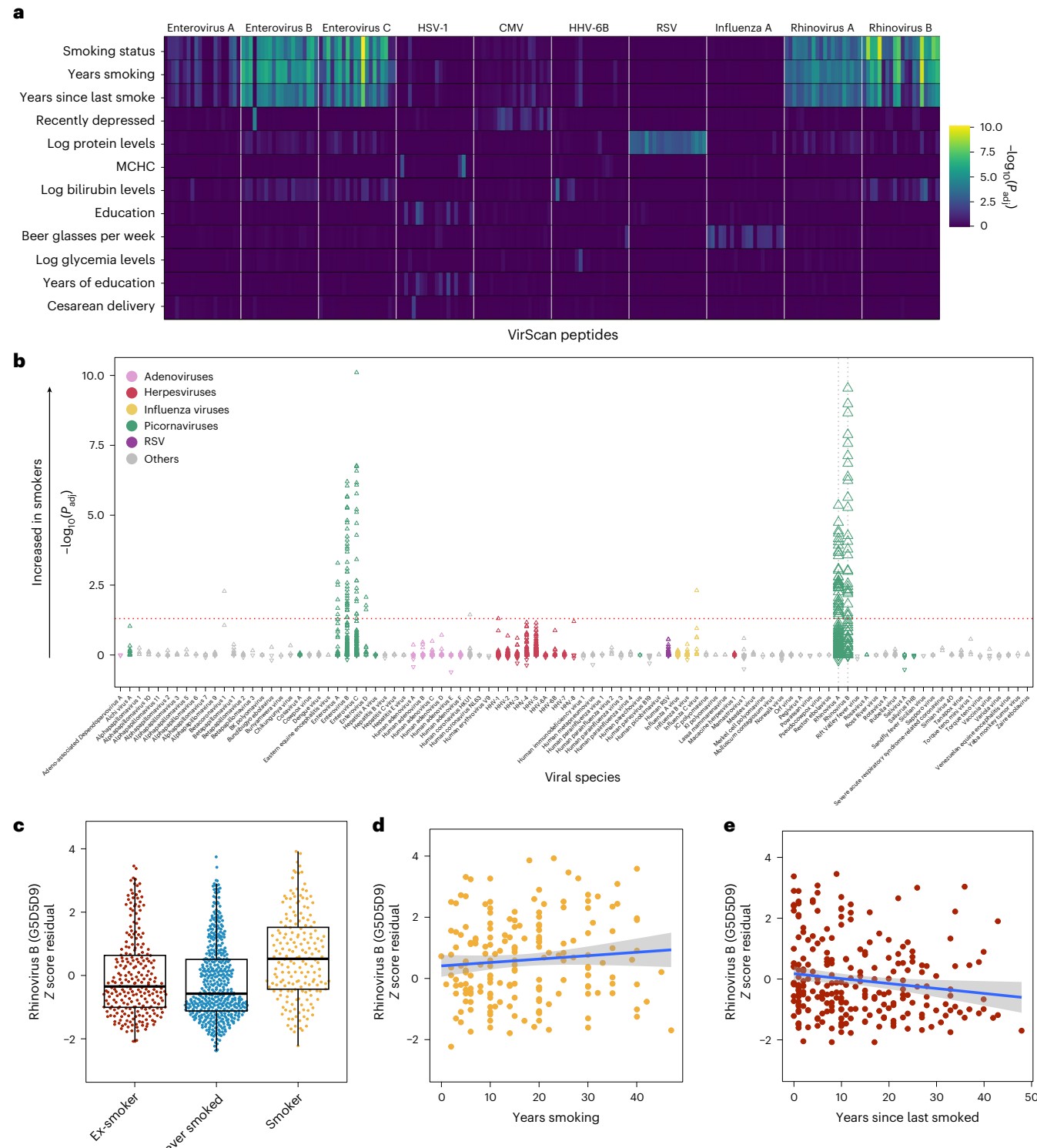

**Fig. 4 | Tobacco smoking elicits strong, reversible effects on antiviral antibody responses. a**, $-\log_{10}(P_{adj})$ values for associations between public peptide $Z$ score and health- and lifestyle-related variables (two-sided Wald test for continuous and binary variables; analysis of covariance for categorical variables). Only the 20 most significant peptides from the ten viruses with the most significant associations are shown. Only variables with an association of $P_{adj} < 0.01$ are shown. **b**, $-\log_{10}(P_{adj})$ values and directions of associations between all public peptide $Z$ scores and smoking status in the Milieu Intérieur cohort, separated by viral species (two-sided Wald test). The direction indicates a positive or negative association with smoking compared with non-smokers. The dotted gray vertical lines indicate viruses for which the AVARDA breadth score is significantly associated with smoking status. The dotted red horizontal line indicates the significance threshold ($P_{adj} < 0.05$). **c**, Antibody reactivity for the rhinovirus B peptide most significantly associated with smoking status, categorized by smoking status. Horizontal lines, box edges and whiskers indicate the median value, interquartile range and 1.5× the interquartile range, respectively. **d**, Antibody reactivity for the rhinovirus B peptide most significantly associated with smoking status, as a function of years of smoking in active smokers. **e**, Antibody reactivity for the rhinovirus B peptide most significantly associated with smoking status, as a function of years since last smoking in former smokers. In **d** and **e**, the blue line indicates the linear regression line and the shaded area represents the 95% confidence interval. MCHC, mean corpuscular hemoglobin concentration.

suggest that smoking exerts a strong, yet reversible, effect on the antibody repertoire against rhinoviruses.

## Germline variants in immunoglobulin genes shape the antiviral antibody repertoire

To identify genetic factors affecting the antiviral antibody repertoire, we conducted a genome-wide association study (GWAS) of 2,608 public peptide $Z$ scores in the Milieu Intérieur cohort, testing associations with 5,699,237 imputed common single-nucleotide polymorphisms (SNPs)[22] while controlling for age, sex and genetic structure (Methods). The EIP cohort was used as a replication cohort. Given the incomplete coverage of B cell receptor loci by the imputed SNPs, we performed next-generation sequencing of the *IGH*, *IGK* and *IGL* genes in all Milieu Intérieur donors at a depth of ~35× coverage, identifying 30,503 additional common variants. In total, we detected strong genome-wide significant associations for 225 viral peptides at four independent loci, including *HLA*, *FUT2*, *IGH* and *IGK* genes ($P < 1.31 \times 10^{-10}$; Fig. 5a and Supplementary Table 5).

We found associations between *HLA* variants and antibody reactivity against 112 peptides from 15 viruses, including EBV, HSV-1 and adenoviruses A–F, consistent with previous studies[2,4,7–10] and replicated in the EIP cohort ($P_{EIP} < 0.05$; Supplementary Table 5). To account for linkage disequilibrium and facilitate comparisons with disease studies, we imputed *HLA* alleles from genotype data and tested for associations between peptide $Z$ scores and allele dosages. This analysis identified 85 associations (Supplementary Table 6), including HLA-DRB1*04 ($\beta = 0.96$; $P = 2.0 \times 10^{-16}$) and HLA-DQA1*03:01 ($\beta = 0.66$; $P = 2.3 \times 10^{-15}$) with adenovirus peptides (Fig. 5b,c), as well as *HLA-DRB1*13 with EBV peptides ($\beta = 1.04$; $P = 7.5 \times 10^{-19}$; Fig. 5d). These alleles have been associated with an increased risk for type 1 diabetes and rheumatoid arthritis[33], potentially explaining the link between these immune-mediated diseases and EBV or adenovirus infections[34,35].

Variants near *FUT2* were associated with antibodies against norovirus peptides ($\beta = 0.31$; $P = 1.10 \times 10^{-10}$; Extended Data Fig. 9a). Mutations in *FUT2* determine the non-secretor phenotype, which confers resistance to norovirus infection and susceptibility to type 1 diabetes and inflammatory bowel disease[36,37]. The most significant variants include the *FUT2* rs601338G>A stop mutation defining the non-secretor status[38] ($\beta = -0.31$; $P = 2.01 \times 10^{-10}$), the protective allele A being associated with reduced anti-norovirus antibody levels. Variants in strong linkage disequilibrium with rs601338 were replicated in the EIP cohort ($P_{EIP} = 5.98 \times 10^{-9}$; $r^2 = 0.998$). We additionally identified a novel association between variants in near-complete linkage disequilibrium ($r^2 = 0.995$) with rs601338 and antibodies against two salivirus strains for both Milieu Intérieur ($\beta = -0.33$; $P < 1.58 \times 10^{-14}$) and EIP ($P_{EIP} < 1.36 \times 10^{-10}$; Extended Data Fig. 9b). Saliviruses, discovered in 2009 in diarrheal samples, are known to cause gastroenteritis[39], although their cellular tropism and entry mechanisms remain unclear. Associations between *FUT2* non-secretor status and anti-salivirus antibodies are unlikely to reflect cross-reactivity with norovirus, as corresponding $Z$ scores were uncorrelated (Extended Data Fig. 9c,d).

Genetic variation at the *IGH* locus was associated with 107 peptides from 21 viruses (Fig. 5a and Supplementary Table 5). This locus encodes the antibody heavy chain and has previously been associated

with antibody levels against various bacteria, IAV and norovirus[4]. Our analyses extended these findings to additional viruses, including herpesviruses (HSV-2, EBV, CMV and HHV-6), RSV, IAV, HBV, coronavirus NL63, rubella virus, sandfly fever Sicilian virus, enteroviruses and rhinoviruses. Several newly identified variants influence *IGHV* clonal gene usage by V(D)J somatic recombination, as assessed by adaptive immune receptor repertoire sequencing in a previous study[40]. For example, we found that a variant associated with anti-rubella antibodies (rs1024350; $\beta = 0.33$; $P = 1.90 \times 10^{-11}$) and suggestively associated with anti-IAV antibodies ($\beta = -0.28$; $P = 5.38 \times 10^{-10}$) affects *IGHV1-69* usage[40] ($P = 1.14 \times 10^{-16}$). *IGHV1-69* gene usage partially determines the quality of anti-influenza antibodies[41] and has been associated with lupus and type 1 diabetes[42]. Another variant, rs9671760, associated with antibodies against rubella virus ($\beta = 0.37$; $P = 3.34 \times 10^{-14}$; Fig. 5e) and sandfly fever Sicilian virus ($\beta = -0.38$; $P = 1.46 \times 10^{-11}$; Fig. 5f), regulates *IGHV3-64* usage[40] ($P = 1.32 \times 10^{-8}$).

The fourth locus included *IGK*, encoding the κ light chain of antibodies, and was associated with antibody levels targeting adenovirus B peptides ($\beta = 0.81$; $P = 1.51 \times 10^{-23}$; Extended Data Fig. 9e). Together, these findings underscore the pervasive impact of host genetic factors, including germline mutations in immunoglobulin genes, on humoral responses to multiple viruses.

## Demographic and genetic factors differentially affect reactivity across viral epitopes

Finally, to quantify the relative contributions of demographic (non-genetic) and genetic factors to antibody variability, we estimated the proportion of variance explained by age, sex, smoking and GWAS lead variants for each of the 2,608 public peptides of the Milieu Intérieur cohort. Together, these factors explained an average of 7.39% (range = 0.91–25.50%) of inter-individual variation in antibody reactivity (Fig. 6a). Demographic factors accounted for 3.81% (range = 0.007–20.68%) of the variance, whereas genetic factors contributed to 3.44% (range = 0.48–23.02%) of the variance. These relative contributions varied substantially across viruses (Extended Data Fig. 10a,b): antibody levels against rhinovirus peptides were dominated by age effects (Fig. 2a), those against CMV were dominated by sex and those against EBV were dominated by genetic variation (Supplementary Table 5).

We also observed substantial variation in the factors explaining the variance of peptide $Z$ scores for the same virus. For example, antibody reactivity to the HA protein of IAV was predominantly explained by age, whereas anti-M1 antibodies were primarily affected by *IGH* genetic variation (Extended Data Fig. 10c). Similarly, anti-EBV antibodies targeting the EBNA-5 protein were strongly shaped by *HLA* genotype, whereas those targeting EBNA-4 and tegument proteins varied primarily in an age-dependent manner (Extended Data Fig. 10d).

A comparable pattern emerged for anti-RSV antibodies, but at the level of a single protein: variance of antibodies against different peptides of the immunogenic glycoprotein G was primarily explained by either age or *IGH* genetic variants (Fig. 6b). Age-associated peptides derived from RSV strain A, whereas *IGH*-associated peptides originated from strain B—two phylogenetic RSV lineages differing in the protein G sequence (Fig. 6c). Specifically, antibodies increasing with age primarily target positions 150–200 of protein G in RSV-A ($\beta = 0.016$;

---

**Fig. 5 | GWAS of antibody reactivity against public peptides. a**, Manhattan plot of associations between 2,608 public peptides and common human genetic variants (minor allele frequency > 5%) in the Milieu Intérieur cohort (two-sided Wald test). Only results with $P < 0.005$ are displayed. The red dashed horizontal line indicates the significance threshold ($P < 1.31 \times 10^{-10}$), as determined by permutations. The top hit of each peak is annotated with the closest gene or gene locus. **b**, Antibody reactivity against the pV protein of adenovirus D as a function of the number of copies of the *HLA-DRB1*04* allele. **c**, Antibody reactivity against the L2 protein of adenovirus B as a function of the number of copies of the HLA-DQA1*03:01 allele. **d**, Antibody reactivity against the EBNA-5 protein of EBV as a function of the number of copies of the HLA-DRB1*13 allele.

In **b**–**d**, horizontal lines, box edges and whiskers indicate the median value, interquartile range and 1.5× the interquartile range, respectively. **e**,**f**, LocusZoom plots for the associations between *IGH* variants and antibody reactivity against: the rubella virus (UniProt ID: D5KJ87) (**e**) and the sandfly fever Sicilian virus (UniProt ID: A7KCL0) (**f**) (two-sided Wald test). The variant most significantly associated with antibody reactivity and the closest gene usage quantitative trait locus variant (rs9671760) are indicated by gray vertical lines. *IGHV* segment locations are indicated at the bottom, and the V segment targeted by the gene usage quantitative trait locus variant (*IGHV3-64*) is labeled. *IGHV* gene, immunoglobulin heavy chain variable gene; lncRNA, long non-coding RNA; miRNA, microRNA; rRNA, ribosomal RNA.

$P = 4.85 \times 10^{-23}$; Fig. 6d)—a pattern consistent with a previous study[43] and replicated in the EIP cohort for EUB individuals only ($\beta_{EIP} = 0.036$; $P_{EIP} = 0.015$; Extended Data Fig. 10e)—whereas antibodies associated with the *IGH* variant (rs59595881) target positions 225–275 of protein G in RSV-B ($\beta = 0.60$; $P = 1.11 \times 10^{-12}$). The position-specific effects of age

and genetics are unlikely to be driven solely by differential exposure to RSV, as Europeans are seasonally exposed to both RSV-A and RSV-B[44]. Overall, these findings indicate that the effects of demographic and genetic factors largely differ among viruses, viral strains, proteins and epitopes targeted by the antibody repertoire.

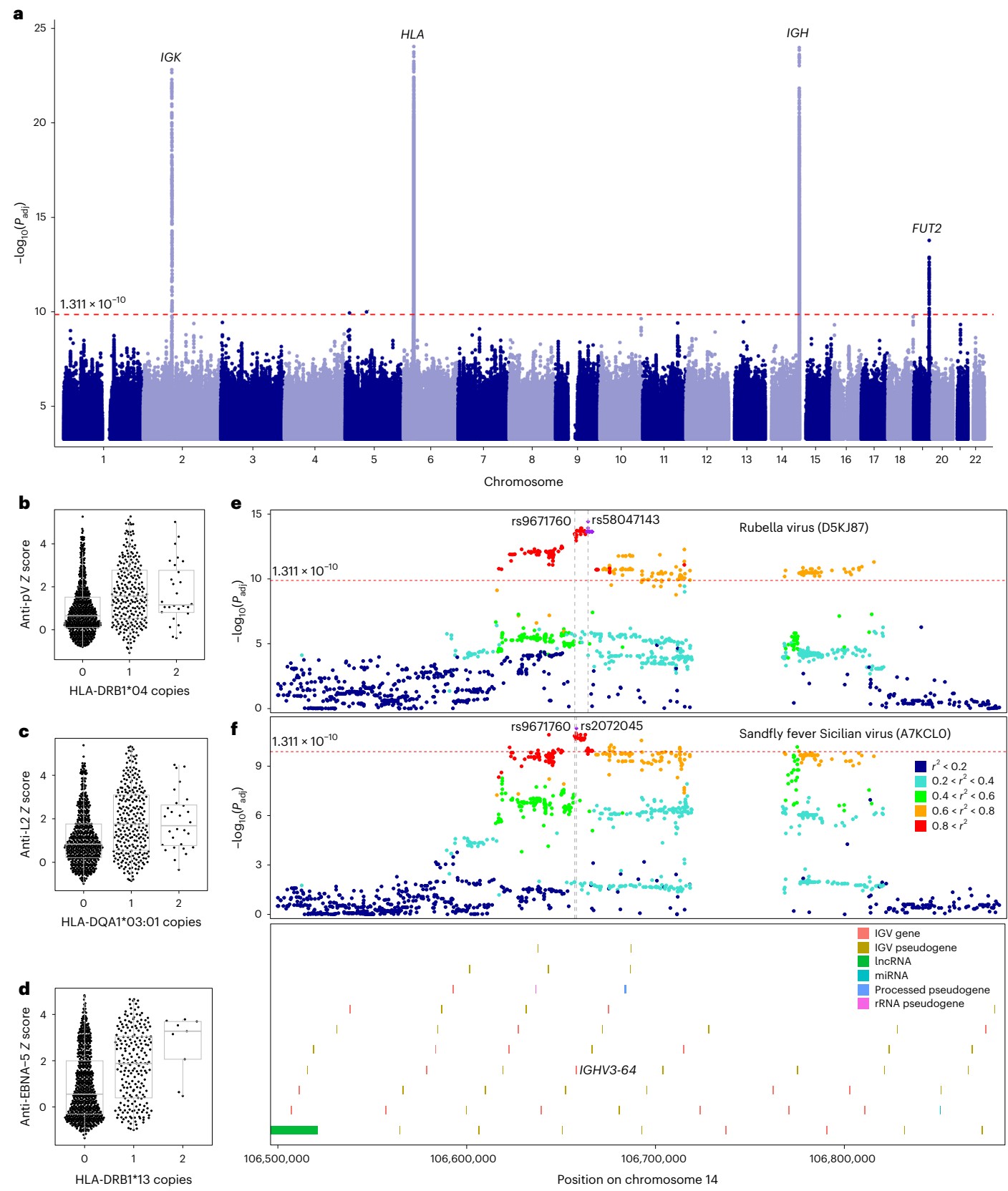

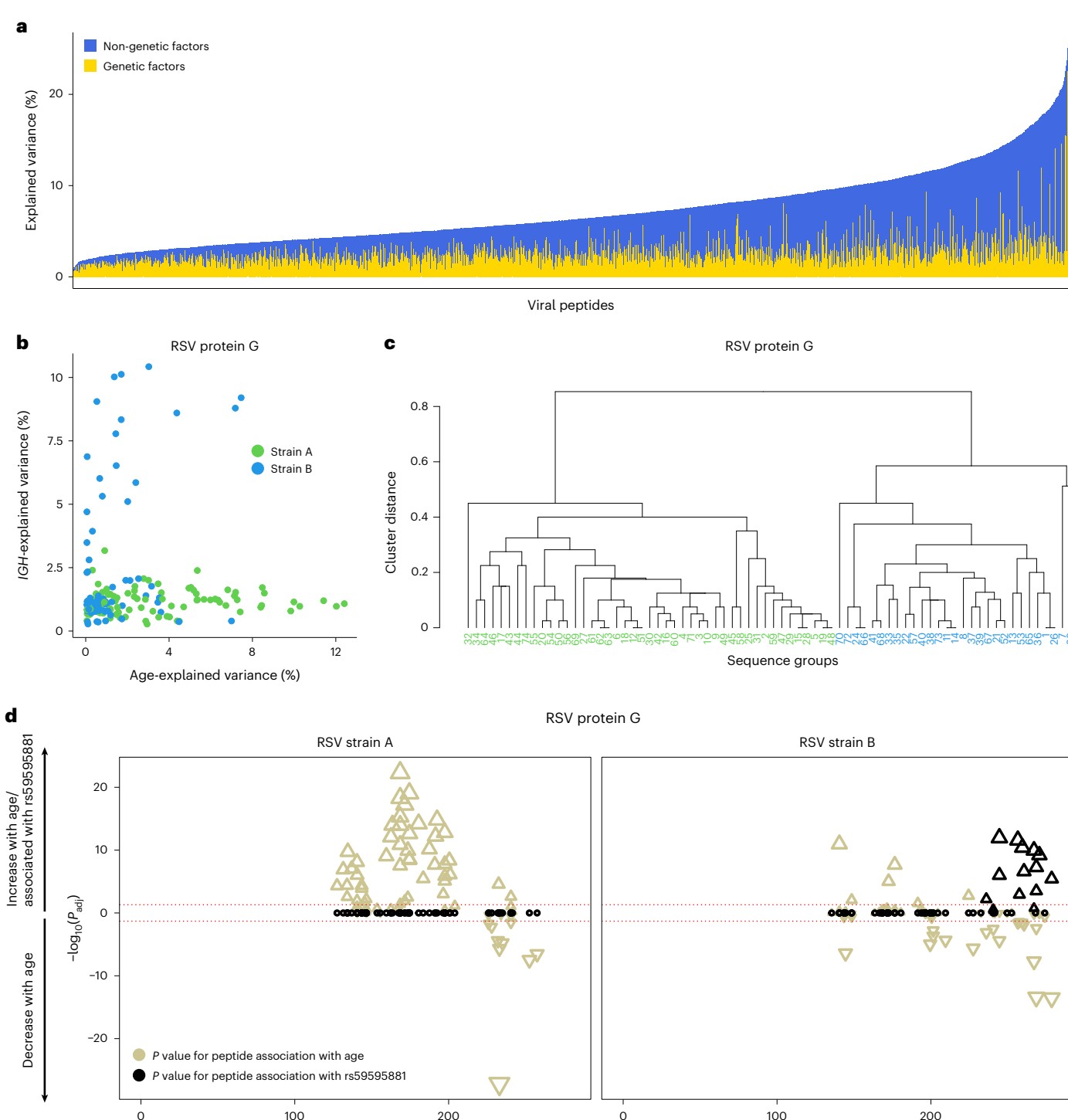

**Fig. 6 | Variance in antiviral antibody reactivity explained by demographic and genetic factors. a**, Proportion of variance explained by demographic (that is, age, sex and smoking) and genetic factors for antibody reactivity against 2,608 public peptides in the Milieu Intérieur cohort. The peptides are sorted by total variance explained. **b**, Variance explained by age and *IGH* genetic variation for RSV protein G peptides in the Milieu Intérieur cohort, colored according to RSV strain, as in **c**. **c**, Hierarchical clustering of peptide sequences from RSV protein G, separating peptides affiliated with the RSV-A (green) and RSV-B (blue) strains. **d**, Amino acid positions of the midpoint of protein G peptides associated with age and *IGH* genetic variation within the full RSV protein G for the Milieu Intérieur cohort (two-sided Wald test). *P* values for the association with age (beige) and the most significant *IGH* variant (black) are indicated, separated by RSV strain. Significance and directions of associations are indicated on the *y* axis and by the direction of triangles, respectively. The dotted red horizontal lines indicate the significance threshold ($P_{adj} < 0.05$).

## Discussion

In this study, we generated a comprehensive dataset of plasma antibody levels against over 97,000 viral peptides, providing a valuable resource to investigate the intrinsic, environmental and genetic factors shaping the antibody repertoire in healthy adults. All of the results can be explored via a dedicated web-based browser (https://mirepertoire. pasteur.cloud/). Among these factors, age had the most profound and widespread effect. Age-related increases in antibody response may reflect higher exposure in older adults (for example, hepatitis A virus and Aichi virus A), reactivation of latent viruses (for example, HSV-1, HSV-2,

EBV and CMV) or reinfections by viruses causing recurrent infections (for example, IAV, IBV and RSV). Conversely, age-related decreases probably reflect higher exposure during young adulthood and/or faster antibody waning (for example, rhinoviruses A–C and enteroviruses B and C).

Importantly, our study shows that aging is associated with differential epitope recognition within the same viral protein. For example, anti-IAV antibodies of younger and older adults preferentially target different domains of viral proteins, this pattern persisting within the same viral strains and for the M1 protein, which is not a typical vaccine target. Therefore, our results suggest that these differences are not solely due to age-related disparities in natural or vaccine-induced exposure to diverse viral strains, but reflect the waning and recall of antibodies targeting variable versus conserved influenza epitopes[45], respectively. Alternatively, the accessibility of certain viral protein domains may require multiple reinfections to elicit antibodies, consistent with proposed mechanisms for age-related differences in neutralizing titers against HA globular head and stalk domains of IAV[46,47]. We propose that age-dependent antigenic specificity, observed here across several IAV proteins, may therefore be a broader phenomenon than was previously appreciated. Similarly, we show that sex influences immunodominance, with women's antibodies preferentially targeting the HA protein of IAV and IBV, whereas men's antibodies disproportionally target NP and M1. These differences are unlikely to reflect vaccination rates, which were similar between sexes, although we cannot exclude that women received greater numbers or more recent vaccine doses than men. Further studies are needed to elucidate mechanisms and implications for age- and sex-related differences in influenza infection risk and vaccine response.

Antibody profiles also vary markedly according to the continent of birth, probably due to differences in viral exposure[16]. Antibodies from individuals born in Central Africa and Europe preferentially target different EBV proteins, suggesting that regional variation in EBV strains[30] contributes to population differences in antibody responses at the epitope level. Among environmental factors affecting the antibody repertoire, we identified a strong association between smoking and anti-rhinovirus antibodies, consistent with previous findings[4] and the higher risk of common cold in smokers[31]. Notably, ex-smokers exhibited antibody levels against rhinoviruses comparable to those of people who had never smoked, suggesting that altered viral clearance and/or heightened exposure in smokers are reversible upon smoking cessation.

Finally, our GWAS confirmed that *HLA* and *IGH* affect antibody levels against a range of viruses[2,4,7–10] and largely expanded the list of associated viruses by revealing novel associations with herpesviruses 2–6, RSV, HBV, rhinoviruses, enteroviruses, coronavirus NL63 and rubella virus. Sequencing of the immunoglobulin genes was critical in discovering these associations, as well as the association with *IGK*, since SNP arrays poorly cover these complex regions. We also identified a strong association between antibodies against the recently discovered and poorly understood saliviruses and *FUT2*, previously linked to norovirus infection, suggesting that saliviruses may exploit similar host infection mechanisms as noroviruses.

Several of these associated genetic variants have previously been linked to increased autoimmune disease risk[33,36,37]. Individuals with these diseases often show higher seroprevalence for common viruses, leading the authors of previous studies to suggest a causal role of these viral infections in autoimmunity[34,35]. However, our results suggest that associations between autoimmune conditions and antibody levels against viruses may instead result from a shared genetic etiology that affects both traits independently. Our study also supports the hypothesis of antagonistic pleiotropy, whereby variants that once conferred resistance to infection now predispose to non-infectious immune diseases[48]. Consistently, *HLA* and *FUT2* alleles associated with antiviral humoral responses and autoimmunity have increased in frequency under natural selection in Europe over the past millennia[49]. Detailed sequencing-based studies in large biobanks are now required to clarify the role of genetic variation in shaping antibody repertoires in immune disorders.

This study has several limitations. First, although the VirScan library offers broad coverage, it is restricted to linear peptides, potentially missing antibodies recognizing conformational epitopes. Second, antibody cross-reactivity between peptides complicates the precise attribution of responses to specific viruses. Although we addressed this risk by using AVARDA, this method may also lead to false negatives. Third, the large number of tests required to analyze the full viral peptidome, combined with the cohort size, may further increase the false negative rate. Fourth, PhIP-seq does not determine whether antibodies are protective against viral infection, which requires dedicated experiments. Finally, although we identified robust associations between antibody repertoires and non-genetic and genetic factors, these explain only a fraction of inter-individual variation. Longitudinal studies integrating the human viral exposome and virome, alongside genome-to-genome association studies[50], are needed to fully elucidate the determinants of human variation in humoral responses to viruses.

Despite these challenges, our study provides high-resolution insights into the widespread effects of age, sex, continent of birth and genetics on the antibody repertoire. Crucially, it reveals that these factors differentially affect antibodies targeting specific epitopes within the same virus or viral protein, deepening our understanding of antibody generation and maintenance. We anticipate that these findings and the accompanying dataset will prompt mechanistic studies of antiviral immunity, with the potential to inform vaccine and therapeutic strategies.

## Online content

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

[1]Human Evolutionary Genetics Unit, Institut Pasteur, Université Paris Cité, CNRS UMR 2000, Paris, France. [2]Division of Micro and Nanosystems, School of Electrical Engineering and Computer Science, KTH Royal Institute of Technology, Stockholm, Sweden. [3]Department of Women's and Children's Health, Karolinska Institutet, Solna, Sweden. [4]Data Management Platform, Institut Pasteur, Paris, France. [5]Evolutionary Genomics of RNA Viruses unit, Institut Pasteur, Université Paris Cité, CNRS UMR 2000, Paris, France. [6]Bioinformatics and Biostatistics Hub, Institut Pasteur, Université Paris Cité, Paris, France. [7]Translational Immunology Unit, Institut Pasteur, Université Paris Cité, Paris, France. [8]Single Cell Biomarkers UTechS, Institut Pasteur, Université Paris Cité, Paris, France. [9]Infectious Disease Epidemiology and Analytics G5 Unit, Institut Pasteur, Université Paris Cité, INSERM U1347, Paris, France. [10]Institut de Recherche Saint Louis, Université Paris Cité, INSERM UMR1342, Paris, France. [11]Department of Immunology and Inflammation, Imperial College London, London, UK. [12]Human Genomics and Evolution, Collège de France, Paris, France. [60]These authors jointly supervised this work: Lluis Quintana-Murci, Etienne Patin. [61]Deceased: Maguelonne Roux. *A list of authors and their affiliations appears at the end of the paper. ✉e-mail: quintana@pasteur.fr; epatin@pasteur.fr

## on behalf of the Milieu Intérieur Consortium

Laurent Abel[13,14], Andres Alcover[15], Hugues Aschard[16], Philippe Bousso[17], Nollaig Bourke[18], Petter Brodin[3,11], Pierre Bruhns[19], Nadine Cerf-Bensussan[20], Ana Cumano[21], Christophe D'Enfert[22], Caroline Demangel[23], Ludovic Deriano[24], Marie-Agnès Dillies[6], James Di Santo[25], Gérard Eberl[26], Jost Enninga[27], Jacques Fellay[28,29], Ivo Gomperts-Boneca[30], Milena Hasan[8], Gunilla Karlsson Hedestam[31], Serge Hercberg[32,33], Molly A. Ingersoll[34,35], Olivier Lantz[36,37], Rose Anne Kenny[18,38], Mickaël Ménager[39,40], Frédérique Michel[41], Hugo Mouquet[42], Cliona O'Farrelly[43,44], Etienne Patin[1,60], Antonio Rausell[45,46], Frédéric Rieux-Laucat[47], Lars Rogge[48], Magnus Fontes[49], Anavaj Sakuntabhai[50,51], Olivier Schwartz[52], Benno Schwikowski[53], Spencer Shorte[62], Frédéric Tangy[54], Antoine Toubert[10], Mathilde Touvier[32,33], Marie-Noëlle Ungeheuer[55], Christophe Zimmer[56,57,58], Matthew L. Albert[59], Darragh Duffy[7,8] & Lluis Quintana-Murci[1,12,60]

[13]Laboratory of Human Genetics of Infectious Diseases, Necker Branch, INSERM U1163, Necker Hospital for Sick Children, Paris, France. [14]St. Giles Laboratory of Human Genetics of Infectious Diseases, Rockefeller Branch, Rockefeller University, New York, NY, USA. [15]Unité Biologie Cellulaire des Lymphocytes, Institut Pasteur, Université Paris Cité, INSERM U1224, Ligue Nationale Contre le Cancer, Équipe Labellisée Ligue-2018, Paris, France. [16]Statistical Genetics Unit, Institut Pasteur, Université Paris Cité, CNRS, Paris, France. [17]Dynamics of Immune Responses Unit, Institut Pasteur, Université Paris Cité, INSERM U1223, Paris, France. [18]Discipline of Medical Gerontology, School of Medicine, Trinity Translational Medicine Institute, Trinity College Dublin, Dublin, Ireland. [19]Antibodies in Therapy and Pathology, Institut Pasteur, Université Paris Cité, INSERM UMR1222, Paris, France. [20]Laboratory of Intestinal Immunity, Imagine Institute, Université Paris Cité, INSERM UMR1163, Paris, France. [21]Unit of Lymphocytes and Immunity, Institut Pasteur, Université Paris Cité, INSERM U1223, Paris, France. [22]Unité Biologie et Pathogénicité Fongiques, Institut Pasteur, Université Paris Cité, INRAE USC2019, Paris, France. [23]Immunobiology and Therapy Unit, Institut Pasteur, Université Paris Cité, INSERM U1224, Paris, France. [24]Genome Integrity, Immunity and Cancer Unit, Institut Pasteur, Université Paris Cité, INSERM U1223, Équipe Labellisée Ligue Contre Le Cancer, Paris, France. [25]Innate Immunity Unit, Institut Pasteur, Université Paris Cité, INSERM U1223, Paris, France. [26]Microenvironment and Immunity Unit, Institut Pasteur, Université Paris Cité, INSERM U1224, Paris, France. [27]Dynamics of Host–Pathogen Interactions Unit, Institut Pasteur, Université Paris Cité, CNRS UMR3691, Paris, France. [28]School of Life Sciences, Ecole Polytechnique Fédérale de Lausanne, Lausanne, Switzerland. [29]Precision Medicine Unit, Lausanne University Hospital and University of Lausanne, Lausanne, Switzerland. [30]Unité Biologie et Génétique de la Paroi Bactérienne, Institut Pasteur, Université Paris Cité, CNRS UMR6047, INSERM U1306, Paris, France. [31]Department of Microbiology, Tumor and Cell Biology, Karolinska Institutet, Stockholm, Sweden. [32]Nutritional Epidemiology Research Team, Centre for Research in Epidemiology and Statistics, Université Sorbonne Paris Nord and Université Paris Cité, INSERM, INRAE, CNAM, Paris, France. [33]Nutrition And Cancer Research Network, Jouy-en-Josas, France. [34]Mucosal Inflammation and Immunity Team, Institut Cochin, Université Paris Cité, CNRS, INSERM, Paris, France. [35]Department of Immunology, Institut Pasteur, Paris, France. [36]Laboratoire d'Immunologie Clinique, Institut Curie, INSERM U932, Paris, France. [37]Centre d'investigation Clinique en Biothérapie Gustave-Roussy Institut Curie (CIC-BT1428), Paris, France. [38]Mercer's Institute for Successful Ageing, St. James's Hospital, Trinity College, University of Dublin, Dublin, Ireland. [39]Laboratory of Single-Cell Inflammatory Responses and Multi-OMICs Networks, Imagine Institute, Université Paris Cité, INSERM UMR1163, Paris, France. [40]Labtech Single-Cell@Imagine, Imagine Institute, Paris, France. [41]Cytokine Signaling Unit, Institut Pasteur, Université Paris Cité, INSERM U1224, Paris, France. [42]Humoral Immunology Unit, Institut Pasteur, Université Paris Cité, Paris, France. [43]School of Biochemistry and Immunology, Trinity College Dublin, Dublin, Ireland. [44]School of Medicine, Trinity College Dublin, Dublin, Ireland. [45]Clinical Bioinformatics Laboratory, Imagine Institute, Université Paris Cité, INSERM UMR1163, Paris, France. [46]Fédération de Génétique et Médecine Génomique, Service de Médecine Génomique des Maladies Rares, Assistance Publique-Hôpitaux de Paris, Necker Hospital for Sick Children, Paris, France. [47]Laboratory of Immunogenetics of Pediatric Autoimmune Diseases, Imagine Institute, Université de Paris, INSERM UMR1163, Paris, France. [48]Immunoregulation Unit, Institut Pasteur, Université Paris Cité, Paris, France. [49]Institut Roche, Paris, France. [50]Ecology and Emergence of Arthropod-borne Pathogens Unit, Institut Pasteur, Université Paris Cité, CNRS UMR2000, Paris, France. [51]International Vaccine Design Center, Institute of Medical Science, University of Tokyo, Tokyo, Japan. [52]Virus and Immunity Unit, Institut Pasteur, Université Paris Cité, CNRS UMR3569, Paris, France. [53]Computational Systems Biomedicine Lab, Institut Pasteur, Université Paris Cité, Paris, France. [54]Institut Pasteur–Oncovita joint laboratory, Université Paris Cité, Paris, France. [55]ICAReB-Biobank, Centre de Ressources Biologiques, Institut Pasteur, Paris, France. [56]Imaging and Modeling Unit, Institut Pasteur, Université Paris Cité, Paris, France. [57]Rudolf Virchow Center for Integrative and Translational Bioimaging, University of Würzburg, Würzburg, Germany. [58]Center for Artificial Intelligence and Data Science, University of Würzburg, Würzburg, Germany. [59]Octant, Emeryville, CA, USA. [62]Deceased: Spencer Shorte.

## Methods

### Data generation

**The Milieu Intérieur cohort.** The Milieu Intérieur cohort comprises 1,000 healthy adults recruited to investigate genetic and non-genetic determinants of immune response variation[19]. Recruitment was conducted in Rennes (France) in 2012–2013 and individuals were selected based on a large set of relatively strict inclusion and exclusion criteria, as described elsewhere[19]. Of the 900 individuals reported in the present study, 453 are female and 447 are male, ranging from 20–69 years of age. The study has been approved by the Comité de Protection des Personnes−Ouest VI and French Agence Nationale de Sécurité du Médicament. The study protocol, including inclusion and exclusion criteria for the Milieu Intérieur study, has been registered on ClinicalTrials.gov under the study ID NCT01699893. The samples and data were formally established as the Milieu Intérieur biocollection (NCT03905993), with approvals by the Comité de Protection des Personnes Sud Méditerranée and Commission Nationale de l'Informatique et des Libertés on 11 April 2018. Research participants received compensation.

**The EIP cohort.** The EIP cohort comprises 390 healthy adults recruited to investigate human population differences in immune responses. Recruitment was conducted in Ghent (Belgium) in 2012–2013. Of the 312 individuals reported in the present study, 100 individuals reported to be born in Central Africa (AFB; age range = 20–50 years) and 212 reported to be born in Europe (EUB; age range = 20–50 years). AFB and EUB individuals presented no evidence of recent genetic admixture with populations originating from another continent, besides two AFB donors who presented 22% Near Eastern and 25% European ancestries, respectively[20]. All individuals were negative for serological tests against human immunodeficiency virus, hepatitis B or hepatitis C. The study was approved by the Ethics Committee of Ghent University, the Ethics Board of Institut Pasteur (EVOIMMUNOPOP-281297) and the French authorities Comité de Protection des Personnes, Comité Consultatif sur le Traitement de l'Information en Matière de Recherche and Commission Nationale de l'Informatique et des Libertés. Research participants received compensation.

**VirScan experimental protocol.** To investigate the virus-specific and viral peptide-specific antibody profiles in the Milieu Intérieur and EIP samples, we used PhIP-Seq using the VirScan V3 library, a pathogen-epitope scanning method combining bacteriophage display and immunoprecipitation. The detailed protocol and VirScan library are described elsewhere[16,18]. In brief, a library of linear peptides of 56 amino acids each was constructed to cover all UniProt protein sequences of viruses known to infect humans. Peptides were staggered along each protein sequence with an overlap of 28 amino acids. The phage library was inactivated and incubated with plasma samples normalized to total IgG concentration and controls (bead samples) to form IgG−phage immunocomplexes. The immunocomplexes were then captured by magnetic beads, lysed and sent for next-generation sequencing. Two replicates were performed for each individual, to assess reproducibility.

**VirScan data pre-processing.** Sequencing reads were processed as in ref. 17, with some modifications. We utilized the Bowtie 2−SAMtools pipeline[51,52] to map the sequencing reads of each sample to the bacteriophage library and count the number of reads for each viral peptide. Subsequently, the positivity of each peptide was determined by a binning strategy whereby read counts from blank controls were first used to group the peptides into hundreds of bins so that the counts formed a uniform distribution within each bin. Then, the peptides from plasma samples were allocated into the pre-defined bins and $Z$ scores were calculated for each peptide from each plasma sample. The means and standard deviations used for the $Z$ score calculations were the same for each bin and were computed using the bead control

sample read counts for the peptides belonging to that bin. After generating a matrix of 115,753 peptide $Z$ scores for 900 Milieu Intérieur or 312 EIP samples, we discarded peptides from bacteria, fungi and allergens from the VirScan library, resulting in 99,460 viral peptides. $Z$ score values were inverse hyperbolic sine (arcsinh) transformed. In contrast to log transformation, the arcsinh function is convenient when handling overdispersion due to both outliers and zero values, which were common in the VirScan $Z$ score data.

Peptides of poor quality were identified by leveraging discordance across replicates. $Z$ score values missing in only one replicate were set to missing in both replicates. Then, outliers in each replicate were defined as $Z$ scores higher than the 99.5% quantile. The absolute difference in $Z$ score between replicates was calculated for all peptides with an outlier value in at least one replicate. The distribution of absolute differences was bimodal, with the lower peak representing consistent $Z$ scores between replicates and the upper peak representing inconsistent $Z$ scores. The local minimum between the peaks was identified using the optimize function from the stats R package, and outliers were defined as all peptides with absolute differences above this minimum. The $Z$ score values of both replicates for all outlier peptides were then set to missing. The rate of missing values was 1.06% in the Milieu Intérieur cohort and 1.09% in the EIP cohort. Next, peptides with >50% missing values were removed from the dataset, leaving 98,757 for Milieu Intérieur and 98,697 for EIP. Duplicated UniProt entries were removed, leaving 97,975 peptides for Milieu Intérieur and 97,923 for EIP for the remaining analyses.

Missing values were imputed by running a principal component analysis on all $Z$ scores using the pca function from the pcaMethods package (nPcs = 10, scale = 'uv'), followed by imputation using the completeObs function from the same package. As individual samples were processed in batches on cell culture plates, samples were batch corrected using the ComBat[53] function from the sva R package, using plates as the batch variable (Supplementary Note). The final $Z$ scores were generated by calculating the mean of the two replicates for each individual. A peptide was considered significantly positive if the $Z$ scores of both replicates were >3.5. The hit variable was defined as 1 if the peptide was positive, and 0 otherwise. To generate the list of public peptides, the datasets were filtered on peptides significantly positive in >5% of tested individuals for at least two peptides per virus.

**VirScan data processing with AVARDA.** Between-species antibody cross-reactivity, unbalanced representation of viruses in the VirScan library and viral genome size can make peptide-level data challenging to interpret in some cases. To address these limitations and compare antibody profiles at the virus species level, we applied AVARDA[21], using the code available at https://github.com/drmonaco/AVARDA. Individual VirScan peptides were aligned to each other and to a master library of all viral genetic sequences translated in all reading frames using BLAST. Evidence peptides were VirScan peptides that aligned to the master library with a bit score of >80. For each virus, AVARDA calculated a maximally independent set of unrelated peptides that explained the total reactivity toward this virus. A probability of infection for each virus was calculated using binomial testing, comparing the ratio of the number of positive evidence peptides with the total number of evidence peptides with the fractional representation of the virus in the VirScan library. Finally, cross-reactivity was evaluated by ranking all viruses based on the probability of infection. Pairs of viruses were then iteratively compared, where shared reactive peptides were assigned to the virus with the most substantial evidence of infection based solely on non-shared peptides. Once all peptides were exclusively assigned to a single virus, a final probability of infection for each sample was calculated using the binomial testing procedure described above. Additionally, a breadth score was calculated, defined as the largest number of reactive peptides from a given virus species that did not share any sequence similarities.

**ELISA-based serological data.** Blood was collected in ethylenediaminetetraacetic acid (EDTA)-treated tubes, and the plasma was extracted by centrifugation. Total levels of the immunoglobulins IgG, IgM, IgE and IgA were measured with a turbidimetric test on an Olympus AU400 Chemistry Analyzer. The ELISA-based serologies were measured for IgG against the following viruses and antigens: CMV, HSV-1, HSV-2, EBV, VZV, IAV, rubella, mumps and measles (Supplementary Table 1). The data processing steps for the immunoassay-based serology data are described in more detail in ref. 2. The absorbance and emission values collected in each assay were used to call serostatus. The cutoff values used for calling a sample positive or negative were given by the manufacturer and can be found in supplementary table 2 of ref. 2.

**Luminex-based serological data.** Milieu Intérieur plasma samples were tested for antibodies to a broad panel of common respiratory pathogens and routine vaccine-preventable diseases using bead-based multiplex assays. Samples were run at a dilution of 1:200. Plates were read using a Luminex INTELLIFLEX system and the median fluorescence intensity was used for analysis. A five-parameter logistic curve was used to convert median fluorescence intensities to relative antibody units, relative to the standard curve performed on the same plate, to account for inter-assay variation. The antigens included in the 43-plex assay are listed in Supplementary Table 1.

**Viral peptide synthesis.** To validate experimental associations between PhIP-seq-based $Z$ scores and age, sex, continent of birth and smoking, the associated peptides were synthesized (Supplementary Table 1) and antibody titers against these peptides were measured by Luminex immunoassay (see next section). Peptide synthesis was performed with automated synthesizers (Genecust) and a tag was added to each peptide according to the standard protocol[54], using solid-phase 9-fluorenylmethoxycarbonyl (Fmoc) chemistry. For the stalk domain of IAV HA, we used a specific chimeric protein, designated as cH6/1, comprising a A/White-fronted Goose/Netherlands/21/1999 (H6HA) head and a A/Puerto Rico/8/1934 (H1HA) stalk[55]. The sequence coding for cH6/1 was cloned into the pαH vector under the control of the CAG promoter. The construct included a fold-on trimerization domain and hexahistidine tag. The plasmid was transiently transfected in Expi293F (Thermo Fisher Scientific) using PEI Max (Polysciences) as a transfection reagent. One day after transfection, the flask was transferred to 32 °C and 6.5 mM sodium propionate and 50 mM glucose were added. Following incubation, the cell culture supernatant was clarified and the recombinant cH6/11 protein was captured on an Ni-NTA column (Ni-advance HiFli; Protein Ark) and stored in small aliquots at −80 °C until further use.

**Peptide Luminex-based serological data.** We validated associations between PhIP-seq-based $Z$ scores and age, sex, continent of birth and smoking by measuring antibody titers against relevant viral peptides using multiplex Luminex immunoassays. To couple viral peptides to MagPlex microspheres, we adapted the protocol from 'Modification of microspheres with ADH'[56] and Wakeman et al.[54]. The first step comprises modifying the microspheres with adipic acid dihydrazide (ADH; Sigma–Aldrich). The second step comprises coupling the peptides to ADH-modified microspheres. The stock uncoupled microspheres were sonicated and vortexed for 30 s. Subsequently, $2.5 \times 10^6$ microspheres (200 μl) were transferred to an Eppendorf tube and washed once with 1 ml 0.1 M 2-($N$-morpholino)ethane sulfonic acid (MES) (pH 6.0) using a magnetic separator. The beads were then activated for 2 h on a rotator at room temperature containing 1 ml of 35 mg ml⁻¹ of ADH and 200 μl of 200 mg ml⁻¹ of 1-ethyl-3-(3-dimethylaminopropyl) carbodiimide hydrochloride (50 mg ml⁻¹; Sigma–Aldrich). The 1-ethyl-3-(3-dimethylaminopropyl) carbodiimide hydrochloride was prepared extemporaneously in 0.1 M MES (pH 6.0) immediately before use. Following activation, the beads were washed three times with 0.1 M MES (pH 4.5) and resuspended in 1 ml of 0.1 M MES (pH 4.5). The beads were stored at 4 °C overnight.

One day after the ADH modification of microspheres, they were washed once with 1 ml of 0.1 M MES (pH 6.0) using a magnetic separator and resuspended in 350 μl of 0.1 M MES (pH 6.0) with 20 μg of each peptide, 10 μl EDC and 10 μl hydroxysulfosuccinimide sodium salt (50 mg ml⁻¹; Sigma–Aldrich). This suspension was incubated for 2 h 30 m in the dark on a rotator. After incubation, the beads were washed twice with 0.1 M MES (pH 6.0) and blocked with 500 μl of 0.1 M MES (pH 6.0) containing 300 μg of each peptide for 1 h at room temperature in the dark in a rotator. After blocking, the beads were washed twice with 0.1 M MES (pH 6.0) and resuspended in 1 ml PBS-TN. The beads were stored at 4 °C. One day after the coupling process, all coupled beads were counted using a TC20 Automated Cell Counter (Bio-Rad). Serum samples were run at a 1:400 dilution. Plates were read using the Intelliflex system at a low detector sensitivity and the median fluorescence intensity was measured.

**Serostatus prediction.** We assessed the performance of different methods that predict serostatus from the VirScan data by comparing the predicted serostatus with the ELISA-based serostatus obtained in the same 900 Milieu Intérieur donors. We focused on predicting serostatus for four common viruses for which ELISA data were available: CMV, EBV (EBNA-1 and EA-D), HSV-1 and HSV-2 (Supplementary Note and Supplementary Table 2). We considered four alternative approaches: (1) the hit-based heuristic method, which assigns seropositivity for a given virus when the number of hits is >3 or >5 (as in ref. 16); (2) the hit-based optimized method, which involves searching for the number of positive hits for a given virus that maximizes prediction precision and recall; (3) the AVARDA-based optimized method, which involves searching for the threshold value of the AVARDA breadth score for a given virus that maximizes prediction precision and recall; and (4) an elastic net penalized logistic regression trained from a subset of the VirScan $Z$ score data.

To train the elastic net model, we shuffled and split the data into a training set (70% of the data) and a test set (30%) so that the ratio of seropositive to seronegative samples in both sets was the same as in the original data. We only considered VirScan peptide $Z$ scores for the tested virus as features during feature selection. Two complementary approaches were implemented to reduce overfitting: we discarded features with variance lower than a user-specified threshold, defining a first hyper-parameter, and kept the features with univariate association statistics higher than a user-specified percentile, defining a second hyper-parameter. A grid-based approach was used to optimize the two hyper-parameters and the ratio between elastic net L1 and L2 penalty, performing a fivefold cross-validation for each point of the three-dimensional grid. We visually inspected learning curves to ensure the absence of overfitting. Processing and modeling were carried out using Python 3.12.2 and the following packages: numpy 1.26.4, scipy 1.12.0, pandas 2.2.1 and scikit-learn 1.4.1.post1. All of the packages were installed in a conda 24.3.0 environment for reproducibility.

To estimate serostatus for the M1 protein of IAV, for which no ELISA data were available, we fitted a two-component Gaussian mixture to the non-transformed $Z$ scores using the mclust R package, and considered the 95% percentile of the left distribution to be the threshold for seropositivity.

**Flow cytometry data.** Ten eight-color flow cytometry panels were previously established[22] to count blood cell types, including 78 counts for 27 innate immune cell subtypes and 51 adaptive immune cell subtypes. The protocols, panel design, staining antibodies and gating strategies used to acquire and analyze flow cytometry data are detailed elsewhere[22]. In brief, cells were acquired using two MACSQuant analyzers calibrated with MACSQuant Calibration Beads (Miltenyi Biotec). Generated MQD files were converted to FCS format and analyzed with FlowJo.

Then, 313 immunophenotypes (cell counts, cell proportions, median fluorescence intensity values and ratios) were exported from FlowJo, including 78 cell counts used in this study. The exclusion of problematic and outlier values was described previously[22]. Some 74 donors failed quality control for the T cell panel and were thus excluded. The remaining missing values were imputed by random forest-based imputation using the missForest R package.

**Kappa-deleting recombination excision circles assay.** To evaluate whether B cell renewal affects antibody levels, we tested the association between all public peptide Z scores and circulating levels of kappa-deleting recombination excision circles (KRECs; that is, excised signal circular DNA segments generated in B cells during their maturation in bone marrow). KRECs serve as surrogates of new B cell output, as they persist in B cells and are diluted with cell division[57]. KREC quantification was performed as in ref. 58, with some modifications. Whole-blood genomic DNA (1–2 µg) was pre-amplified for 3 min at 95 °C and then 18 cycles of 95 °C for 15 s, 60 °C for 30 s and 68 °C for 30 s, in a 50 µl reaction containing primers, 200 µM of each dNTP, 2.5 mM $MgSO_4$ and 1.25 U Platinum Taq DNA Polymerase, High Fidelity (Thermo Fisher Scientific) in 1× buffer. The forward and reverse primers were TCAGCGCCCATTACGTTTCT and GTGAGGGACACGCAGCC for signal joint KRECs and CCCGATTAATGCTGCCGTAG and CCTAGGGAGCAGGGAGGCTT for coding joint KRECs, respectively. The probes were CCAGCTCTTACCCTAGAGTTTCTGCACGG (signal joint KRECs) and AGCTGCATTTTTGCCATATCCACTATTTGGAGTA (coding joint KRECs). Columns of 48.48 Dynamic Array Integrated Fluidic Circuits (Fluidigm) were loaded with 5 µl of a mixture containing 2.25 µl of a 1/2,000th dilution of pre-amplified DNA, 2.5 µl of 2× Takyon Low ROX Probe MasterMix (Eurogentec) and 0.25 µl of sample loading reagent. Rows were loaded with an equal mixture of 2× Assay Loading Reagent and 2× Assay Biomark containing only the two primers and the probe specific for each assay. These columns were subjected to 40 cycles of PCR (95 °C for 15 s and 60 °C for 60 s) in a Biomark HD system (Fluidigm). Coding and signal joint KRECs were normalized to 150,000 cells using quantification of the albumin gene as an endogenous control.

**Genome-wide SNP genotyping.** Details about SNP array genotyping of the Milieu Intérieur cohort are available elsewhere[22]. DNA was extracted from whole blood collected on EDTA using the Nucleon BACC3 Genomic DNA Extraction Kit (RPN8512; Cytiva). The 1,000 Milieu Intérieur individuals were genotyped using the HumanOmniExpress-24 BeadChip (Illumina), and 966 were also genotyped using the HumanExome-12 BeadChip (Illumina). After applying quality control filters, the SNP array datasets from the two genotyping platforms were merged. SNPs that were discordant in genotypes or position between the two platforms were removed, yielding a final dataset containing 732,341 genotyped SNPs. The dataset was then phased using SHAPEIT2 (ref. 59) and imputed using IMPUTE2 (ref. 60), with 1 Mb windows and a buffer region of 1 Mb. After imputation, SNPs with an information metric of ≤0.8, duplicated SNPs, SNPs with a missingness of >5% and SNPs with a minor allele frequency of ≤5% were removed, generating a final dataset of 5,699,237 SNPs. We removed 13 individuals based on relatedness and admixture[22]. Finally, the dataset was converted to GRCh38 using the LiftoverVcf function from the GATK software package[61].

Details about SNP array genotyping of the EIP cohort are available elsewhere[20]. Peripheral blood mononuclear cells were isolated from blood collected into EDTA vacutainers, monocytes were removed with CD14+ microbeads, and DNA was isolated from the monocyte-negative fraction using a standard phenol–chloroform protocol, followed by ethanol precipitation. Genotyping was performed in all individuals using the HumanOmni5-Quad BeadChip (Illumina) and whole-exome sequencing was performed with the Nextera Rapid Capture Expanded Exome kit. The SNP array genotyping and whole-exome sequencing data were processed separately and merged. For the SNP array

data, SNPs were passed through multiple quality control filters, and SNPs originating from the sex chromosomes were removed. For the whole-exome sequencing data, reads were processed according to GATK Best Practices. Discordant variants between the two datasets were removed before merging the SNP array and whole-exome sequencing datasets. After combining the two datasets, the data were phased using SHAPEIT2 and imputed using IMPUTE2, with 1 Mb windows and a buffer region of 1 Mb. After imputation and additional quality control filtering, 19,619,457 SNPs remained. The dataset was converted to GRCh38 using the LiftoverVcf function from the GATK software package[61]. Finally, four individuals were removed based on relatedness and admixture[20].

**Whole-genome sequencing.** Whole-genome sequencing was performed by the Centre National de Recherche en Génomique Humaine at the Institut de Biologie François Jacob. After quality control, 1 µg genomic DNA was used to prepare a library using the Illumina TruSeq DNA PCR-Free Library Preparation Kit, according to the manufacturer's instructions. After normalization and quality control, qualified libraries were sequenced on an Illumina HiSeq X5 platform as paired-end 150 bp reads. One lane of HiSeq X5 flow cell was produced for each sample in order to reach an average sequencing depth of ~30× for each sample. FASTQ files were mapped on the human reference genome version hs37d5, using BWA-MEM with default options[62]. BAM file integrity was verified and duplicated reads were identified with PicardTools and SAMtools. Reads were realigned and recalibrated with GATK[61] version 4.1. Sequencing reads mapping to the *HLA*, *IGH*, *IGK* and *IGL* loci were extracted from the mapped BAM files. Genotypes were called in each individual with HaplotypeCaller in GVCF mode. Multi-sample genotype calling was performed jointly on combined GVCF files with GATK GenotypeGVCFs. After variant quality score recalibration, variants that passed the tranche sensitivity threshold of 99.0% were selected. Multiallelic sites were split into several biallelic sites with bcftools norm -m-both and variants spanning deletions were filtered out. Genotypes were set to missing if the depth of coverage was <8× or the genotype quality was <20. Based on kinship coefficients estimated with KING[63], ten related individuals and one individual detected as contaminated were excluded. Finally, variants with a minor allele frequency of <0.05, a Hardy–Weinberg equilibrium P value of <$10^{-10}$ (calculated using the HWExact function from the GWASExactHW R package) or a call rate of <0.95 were discarded, resulting in a total of 30,503 common variants near and within immunoglobulin genes.

## Statistics and reproducibility

**Testing associations between VirScan Z scores and non-genetic factors.** All statistical associations were tested using multiple regression models. In all models, the dependent variable was either an asinh-transformed VirScan Z score (for a given peptide) or an AVARDA breadth score (for a given virus). The independent variables could be: (1) serological measurements based on ELISA; (2) serological measurements based on Luminex xMAP assays; or (3) age, sex, continent of birth and candidate non-genetic factors, including smoking, diet, past diseases, health biomarkers and anthropometric measures (Supplementary Table 3). The three groups of variables, (1), (2) and (3), were treated as independent families of tests and P values were adjusted for multiple testing accordingly, using the false discovery rate procedure. Tests within the Milieu Intérieur and EIP cohorts were also considered independent. As detailed below, the specific model and complete list of covariates used varied depending on the independent variables being tested.

The effect size of each independent continuous or binary variable was estimated and tested for being non-null (that is, the two-sided alternative hypothesis) using the linear regression model implemented in the glm R function. The $\beta$ value was used to determine the effect size of the independent variable. When the independent variable was

categorical with more than two levels, an analysis of covariance model was applied using the aov R function. In the association analyses of the Milieu Intérieur cohort, age and sex were systematically included as covariates. We also investigated nonlinear effects of age by testing an analysis of variance model that models age as a factor with five ten-year levels. In addition, we tested for age × sex, age × smoking and sex × smoking interactions by adding an interaction term to the linear model. The only analyzed independent variables for the EIP cohort were age and continent of birth. As all individuals in the EIP cohort were male, sex was not used as a covariate in these analyses. When age was used as the variable of interest, the continent of birth was controlled for, and vice versa. To separate genetic from non-genetic effects of continent of birth, we performed an additional analysis that also included genetic variants that influenced the antibody repertoire (Supplementary Table 7; see the section 'Estimation of the proportion of variance explained' below).

To leverage the high resolution of the VirScan peptide library while accounting for between-species antibody cross-reactivity, we first tested the association between all public peptide Z scores and non-genetic factors and then evaluated whether AVARDA breadth scores for the tested viruses were associated with the corresponding factors. We considered three scenarios: (1) both the Z scores for several peptides of a given virus and the AVARDA score for the same virus were associated with the candidate factor in the same direction, interpreted as a true association; (2) the Z scores for several peptides of a given virus were associated with the candidate factor in the same direction, but the AVARDA score for the same virus was not, interpreted as a false association due to cross-reactivity; and (3) the Z scores for several peptides of a given virus were associated with the candidate factor in opposite directions, but the AVARDA score for the same virus was not associated, interpreted as true associations obscured by opposite epitope-specific effects.

**Testing associations between VirScan scores and genetic factors.** GWAS was conducted on the asinh-transformed VirScan Z scores in the Milieu Intérieur cohort. The EIP cohort was used as a replication cohort. To correct for population stratification, a principal component analysis was run on all SNPs, and the first two principal components were included as covariates. Age was included as a covariate for both cohorts, and sex was included as a covariate for the Milieu Intérieur cohort only. The population of origin was included as an additional binary covariate for the EIP cohort. The GWAS analyses were conducted using the assocRegression function from the GWASTools R package[64], using a linear additive model. The genome-wide significance threshold was defined as $P = 1.31 \times 10^{-10}$ (that is, the minimum P value obtained by running the GWAS of the 2,608 peptide Z scores after randomly permuting donor identifiers). Manhattan plots, locusZoom plots and tables were all made using the topr R package[65].

***HLA* allele imputation and association testing.** *HLA* allele imputation was done using whole-genome sequencing data of the *HLA* locus (here defined as position 28–35 Mb in GRCh37), using all variants in the region with a minor allele frequency of ≥5%. Imputation was conducted on the Michigan Imputation Server[66], using the four-digit multi-ethnic *HLA* reference panel (version 2). Association testing was conducted similarly to individual SNP analysis but using *HLA* allele dosages instead of SNP genotypes.

**Estimation of the proportion of variance explained.** The proportion of variance explained by demographic and genetic factors was estimated for the VirScan Z scores of the 2,608 public peptides in the Milieu Intérieur cohort. Genetic factors were the most associated SNPs identified through conditional GWAS (that is, by testing associations with all variants while controlling for hitherto identified lead SNPs). This process was continued until no more SNPs with a P value below

genome-wide significance ($P < 1.31 \times 10^{-10}$) were identified, leaving a total of 17 SNPs (Supplementary Table 7). Age, sex and smoking were included as demographic factors. The contribution of each of these 20 variables to the variance of each peptide Z score was estimated using the relaimpo R package[67].

**Estimation of viral evolutionary rates.** To estimate evolutionary rates for each residue of the IAV HA and M1 proteins (Fig. 2), HA sequences for all H3 subtypes and M1 sequences for the H5N1 subtype with collection dates between 1 January 1975 and 1 January 2013 were retrieved from the GISAID EpiFlu database on 13 June 2025. Removing sequences with gaps or ambiguities resulted in 4,209 HA sequences and 3,301 M1 sequences. The accession number, virus name, collection date, originating laboratory, submitting laboratory and contributors of each individual sequence can be accessed under the accession codes EPI_SET_250807be (https://doi.org/10.55876/gis8.250807be) and EPI_SET_250807kf (https://doi.org/10.55876/gis8.250807kf), respectively. Normalized evolutionary rates were calculated for each residue with the empirical Bayesian inference from ConSurf-DB[68].

**Phylogenetic analyses.** All UniProt amino acid sequences used to build the VirScan peptide library for RSV protein G were aligned with the msa function from the msa package[69]. The 41-amino-acid-long region that was covered by the largest number of UniProt sequences was identified. Based on this shared region, a distance matrix between all UniProt sequences was computed with the DistanceMatrix function from the DECIPHER package[70], and complete-linkage clustering was used to obtain a phylogenetic tree using the hclust R function. Strain annotations were then interpolated for all VirScan peptides using the constructed tree.

### Reporting summary
Further information on research design is available in the Nature Portfolio Reporting Summary linked to this article.

## Data availability
The VirScan V3 PhIP-seq raw and processed data generated in this study have been deposited in the Institut Pasteur data repository, OWEY, and can be accessed via https://doi.org/10.48802/owey.84rn-jg72 (Milieu Intérieur) and https://doi.org/10.48802/owey.uCQ5VsxD (EIP). All association statistics obtained in this study can also be explored and downloaded from http://mirepertoire.pasteur.cloud/. All other pseudonymized datasets can be accessed on OWEY by submitting a data access request at https://redcap.pasteur.fr/surveys/?s=ND8TP8MDD3 (Milieu Intérieur) or https://redcap.pasteur.fr/surveys/?s=F3AA7J4M4W8LRNJ4 (EIP). The request will be reviewed by the respective data access committees. Data access committees inform research participants of the data access request and grant data access if the request is consistent with the informed consent signed by the participants. In particular, research on Milieu Intérieur and EIP datasets is restricted to research on the genetic and environmental determinants of human variation in immune responses. Data access is typically granted two months after request submission.

## Code availability
The custom code used to generate the results presented in this study was written in R and is available via GitHub at https://github.com/h-e-g/virscan_association.

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

## Acknowledgements

We thank the HPC Core Facility of Institut Pasteur for supporting this work. Pre-processing of the VirScan data was supported by resources from the National Academic Infrastructure for Supercomputing in Sweden, partially funded by the Swedish Research Council (2022-06725). We thank J. Vahokoski, R. J. Cox Brokstad and F. Krammer for providing the purified, chimeric cH6/1 protein. We gratefully acknowledge all data contributors for generating the genetic sequences and metadata and sharing via the GISAID Initiative, on which some of this research is based. This study is sponsored by the Institut Pasteur (Pasteur ID-RCB number: 2012-A00238-35) and was funded by the French Government's Investissements d'Avenir program, managed by the Agence Nationale de la Recherche (ANR-10-LABX-69-01). A.O. was supported by the Wenner–Gren Foundation, and D.L. by a Pasteur–Roux–Cantarini fellowship. The E.S.-L. laboratory is funded by the INCEPTION program (Investissements d'Avenir; ANR-16-CONV-0005), Ixcore Foundation for Research and HERA Project DURABLE (101102733).

## Author contributions

A.O., L.Q.-M. and E.P. conceived of and developed the study. F. Dubois and B.C. prepared the DNA samples. C.P., Z.T. and P.B. acquired the VirScan V3 data. D.D. and P.B. advised on the experiments. F. Donnadieu, L.G., C.L., E.B. and M.W. developed and performed the Luminex-based immunoassays. E.C., I.L.A. and A.T. generated the KREC data. A.J. developed the predictive algorithms. J.C. conducted viral conservation analyses under the supervision of E.S.-L. A.O. performed all of the remaining analyses, with contributions from A.J., M. Roux, M. Rotival and D.L., under the supervision of E.P. A.O. and E.P. wrote the manuscript, with input from L.Q.-M. All authors discussed the results and contributed to the final manuscript.

## Competing interests

The authors declare no competing interests.

## Additional information

**Extended data** is available for this paper at https://doi.org/10.1038/s41590-026-02432-7.

**Correspondence and requests for materials** should be addressed to Lluis Quintana-Murci or Etienne Patin.

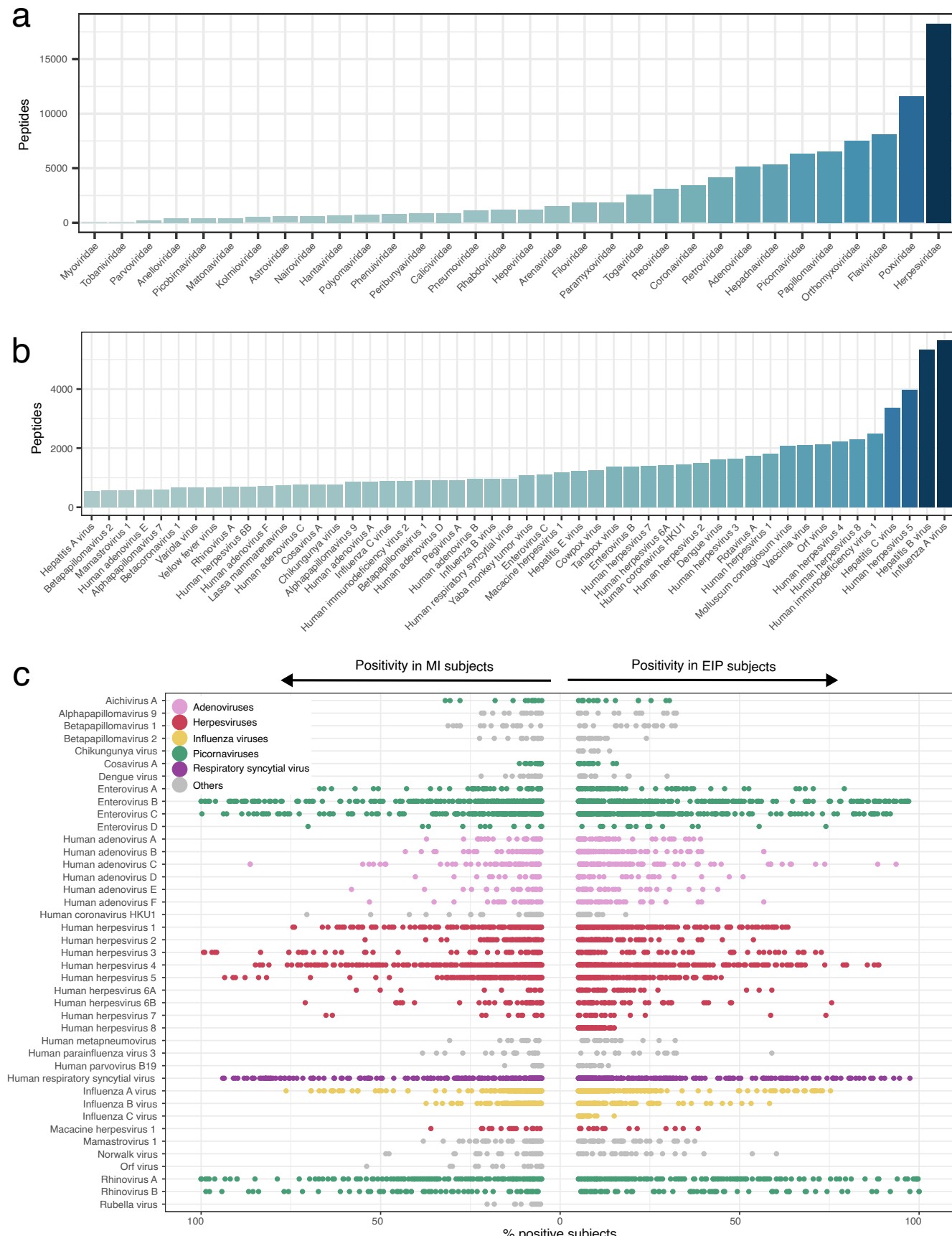

**Extended Data Fig. 1 | See next page for caption.**

**Extended Data Fig. 1 | Overview of the viruses targeted by the VirScan assay in the Milieu Intérieur and EIP cohorts. a,b,** Number of peptides in the VirScan PhIP-seq library, separated by (**a**) viral family and (**b**) species. Only the 50 most covered viruses are shown. **c**, Percentage of Milieu Intérieur (left) and EIP (right) individuals positive for 2,608 public peptides, separated by virus. Each point indicates a viral peptide, colored according to its viral family. Only viruses with at least 10 peptides showing an enrichment of >5% are included.

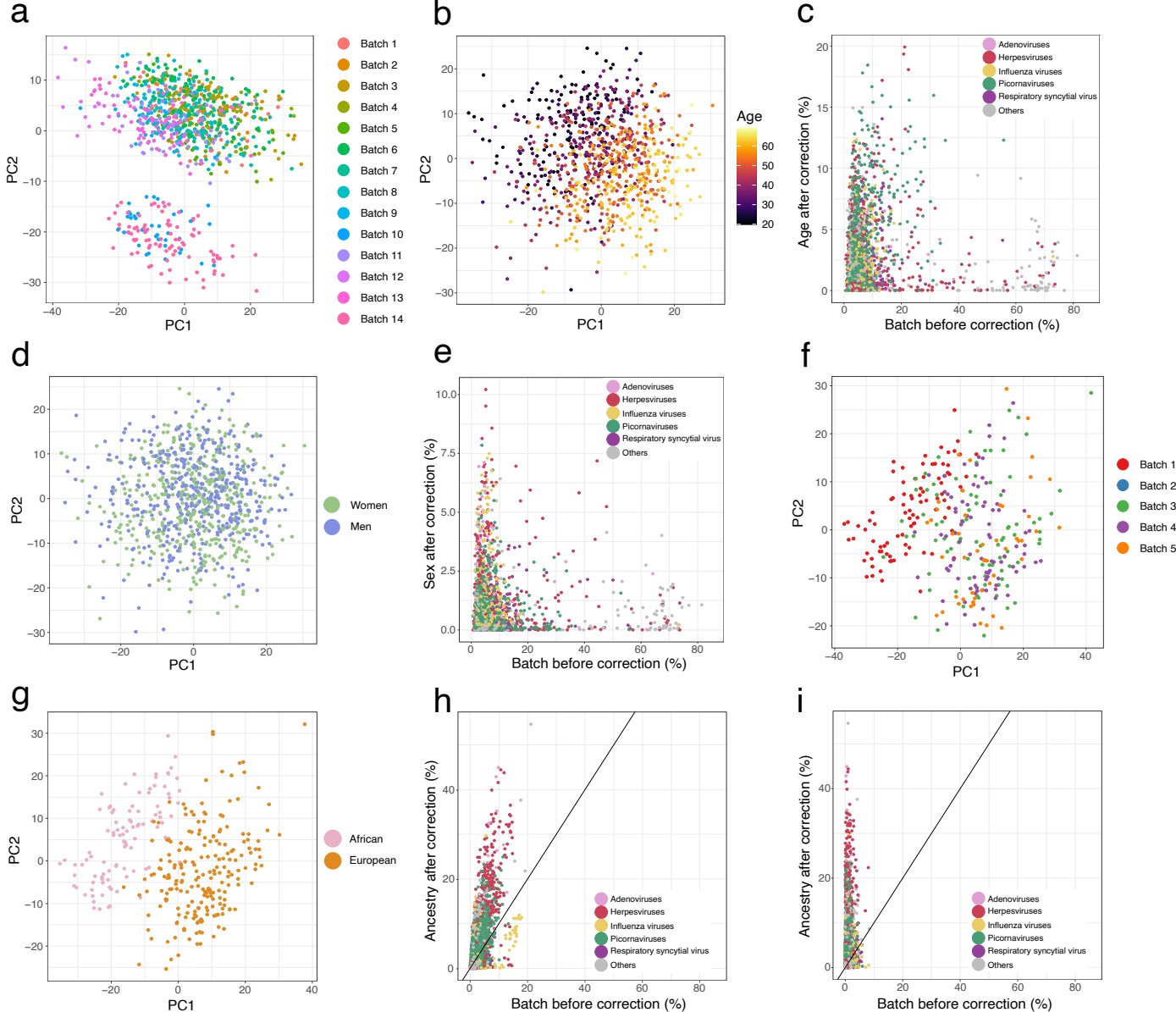

**Extended Data Fig. 2 | Effects of age, sex and continent of birth are not confounded by batch effects. a**, Principal component analysis of the Milieu Intérieur dataset, before batch correction ($n$ = 2,608 public peptide $Z$-scores). Colors indicate sequencing batches. **b**, Principal component analysis of the Milieu Intérieur dataset, after batch correction. Colors indicate age decades in years. **c**, Variance explained by batches before batch correction against that explained by age after batch correction, in the Milieu Intérieur cohort. **d**, Principal component analysis of the Milieu Intérieur dataset, after batch correction. Colors indicate sex. **e**, Variance explained by batches before batch correction against that explained by sex after batch correction, in the

Milieu Intérieur cohort. **f**, Principal component analysis of the EIP dataset, before batch correction ($n$ = 3,210 public peptide Z-scores). Colors indicate sequencing batches. **g**, Principal component analysis of the EIP dataset, after batch correction. Colors indicate continent of birth (AFB: Belgians born in Central Africa; EUB: Belgians born in Europe). **h**, Variance explained by batches before batch correction against that explained by continent of birth after batch correction, in the EIP cohort. **i**, Variance explained by batches before batch correction in EUB samples only against that explained by continent of birth after batch correction, in the EIP cohort. **h,i**, The black line indicates the identity line. **c,e,h,i**, Colors indicate viral families.

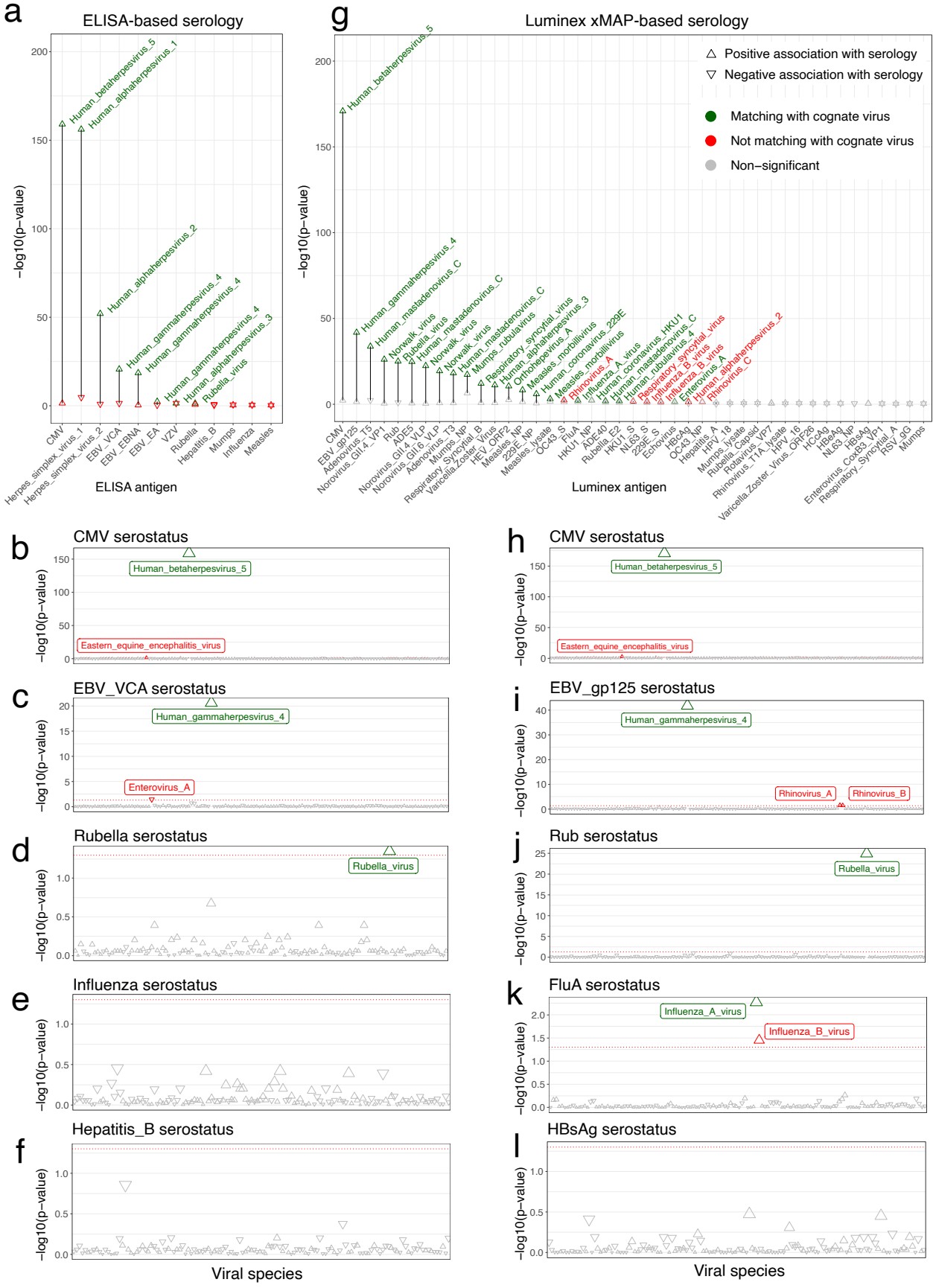

**Extended Data Fig. 3 | See next page for caption.**

**Extended Data Fig. 3 | Validation of AVARDA scores by comparisons with ELISA- and Luminex-based serostatuses in the Milieu Intérieur cohort.** **a,g,** −log$_{10}$(*P*-values) for the association between serology determined by (**a**) ELISA or (**g**) Luminex xMAP and the AVARDA breadth scores. Serology variables are plotted on the x-axis. The top two AVARDA associations are connected by a black vertical line. Significant associations (FDR < 0.05) are colored in green or red if the association is or is not for the cognate virus, respectively. Non-significant associations are colored gray. **b-f,** −log$_{10}$(*P*-values) for the association between the AVARDA breadth scores and ELISA-based serostatus for (**b**) CMV, (**c**) EBV VCA antigen, (**d**) rubella virus, (**e**) IAV, and (**f**) hepatitis B. **h-l,** −log$_{10}$ (*P*-values) for the association between the AVARDA breadth scores and Luminex-based serology for (**h**) CMV, (**i**) EBV VCA antigen, (**j**) rubella virus, (**k**) IAV, and (**l**) hepatitis B. Significant associations (FDR < 0.05) are colored in green or red if the association is or is not for the cognate virus, respectively. Non-significant associations are colored gray.

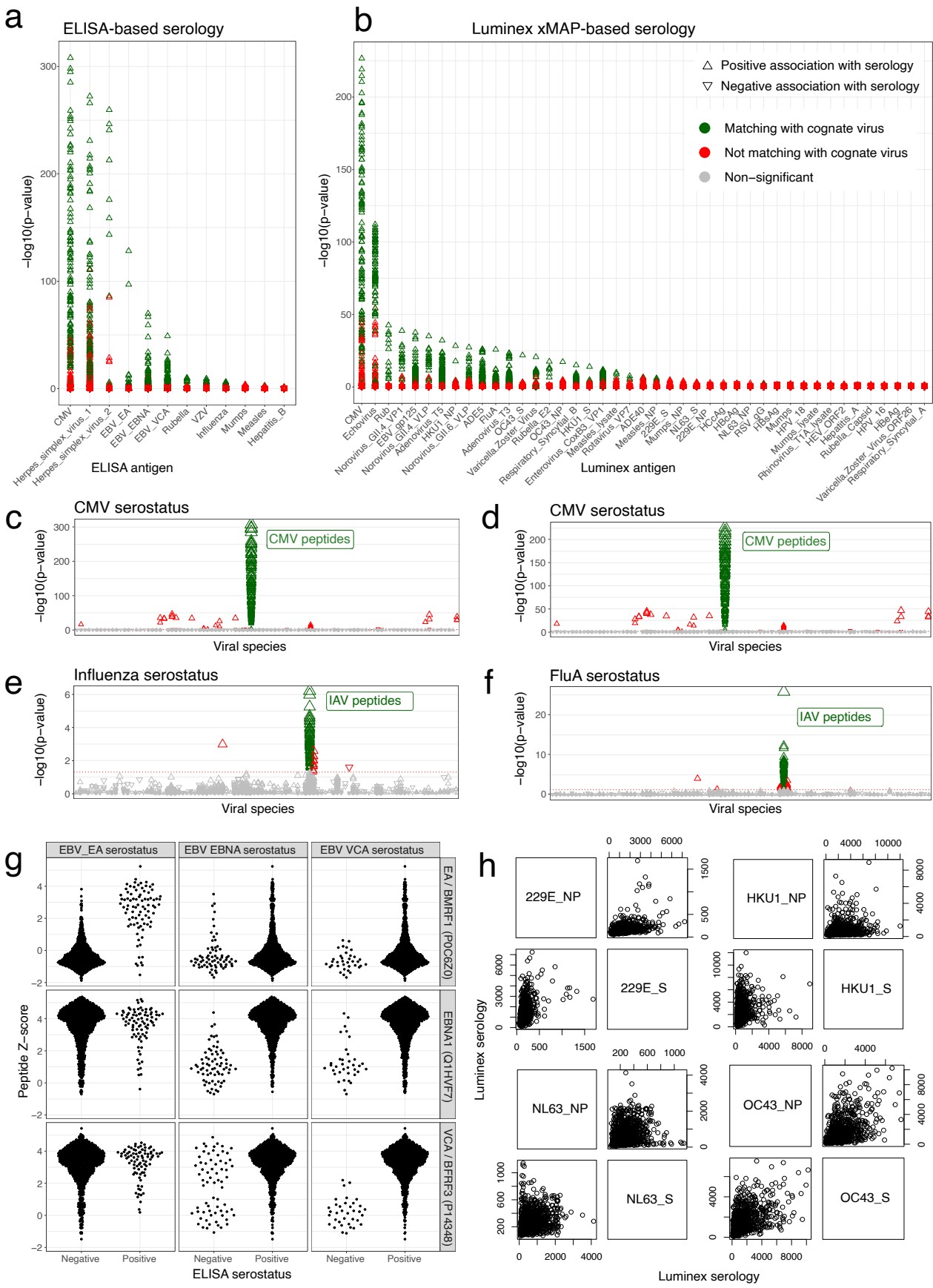

**Extended Data Fig. 4 | See next page for caption.**

**Extended Data Fig. 4 | Validation of VirScan *Z*-scores by comparisons with ELISA- and Luminex-based serostatuses in the Milieu Intérieur cohort.**
**a,b**, −log₁₀(*P*-values) for the association between the VirScan Z-scores and serology determined by (**a**) ELISA or (**b**) Luminex xMAP. Serology variables are plotted on the x-axis, and for each serology variable, the −log₁₀(*P*-values) for the association with all 2,608 public VirScan peptides are shown on the y-axis. Significant associations (FDR < 0.05) are colored in green or red if the association is or is not for the cognate virus, respectively. Non-significant associations are colored gray. **c,e**, −log₁₀(*P*-values) for the association between the VirScan *Z*-scores and ELISA-determined serostatus for (**c**) CMV and (**e**) IAV. Significant associations (FDR < 0.05) are colored in green or red if the association is or is not for the cognate virus, respectively. Non-significant associations are colored gray. **d,f**, −log₁₀(*P*-values) for the association between the VirScan *Z*-scores and Luminex-determined serology for (**d**) CMV and (**f**) IAV. Significant associations (FDR < 0.05) are colored in green or red if the association is or is not for the cognate virus, respectively. Non-significant associations are colored gray. **g**, VirScan *Z*-score distributions for the three peptides most significantly associated with ELISA-based serostatus for the EBV antigens EA, EBNA and VCA. ELISA-based serostatus for each EBV antigen is shown on the x-axis. **h**, Luminex-based serologies for the spike and nucleocapsid proteins shown for the four coronaviruses 229E, HKU1, NL63, and OC43.

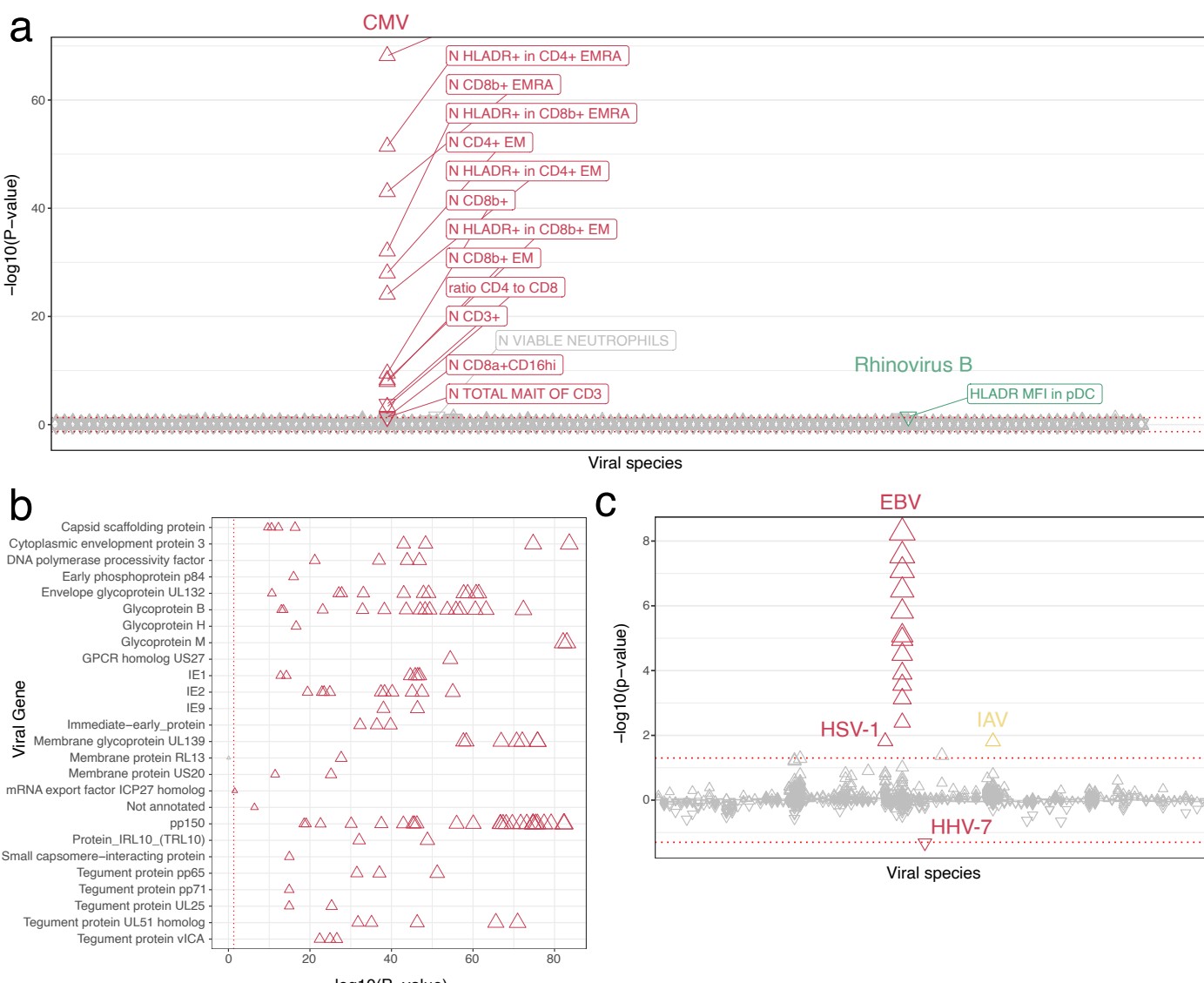

**Extended Data Fig. 5 | Associations between the anti-viral antibody repertoire and immune cell phenotypes in the Milieu Intérieur cohort. a**, −log₁₀ (adjusted *P*-values) and direction of associations between AVARDA breadth scores and immune cell phenotypes in the Milieu Intérieur cohort, by viral species. Each point indicates an association between an immune cell phenotype and the AVARDA antibody breadth score for a given virus, colored according to its viral family. The dashed red horizontal lines indicate the significance threshold ($P_{adj} < 0.05$). **b**, −log₁₀ (adjusted *P*-values) of associations between the frequency of CD4⁺ T$_{EMRA}$ cells and CMV peptide *Z*-scores, separated by viral protein. **c**, −log₁₀ (adjusted *P*-values) and direction of associations between all public peptide *Z*-scores and the HLA-DR MFI in the cDC1 population, by viral species. The dashed red horizontal lines indicate the significance threshold ($P_{adj} < 0.05$). **a-c**, The strength and direction of associations are indicated by the size and direction of triangles, respectively. Non-significant associations are colored gray.

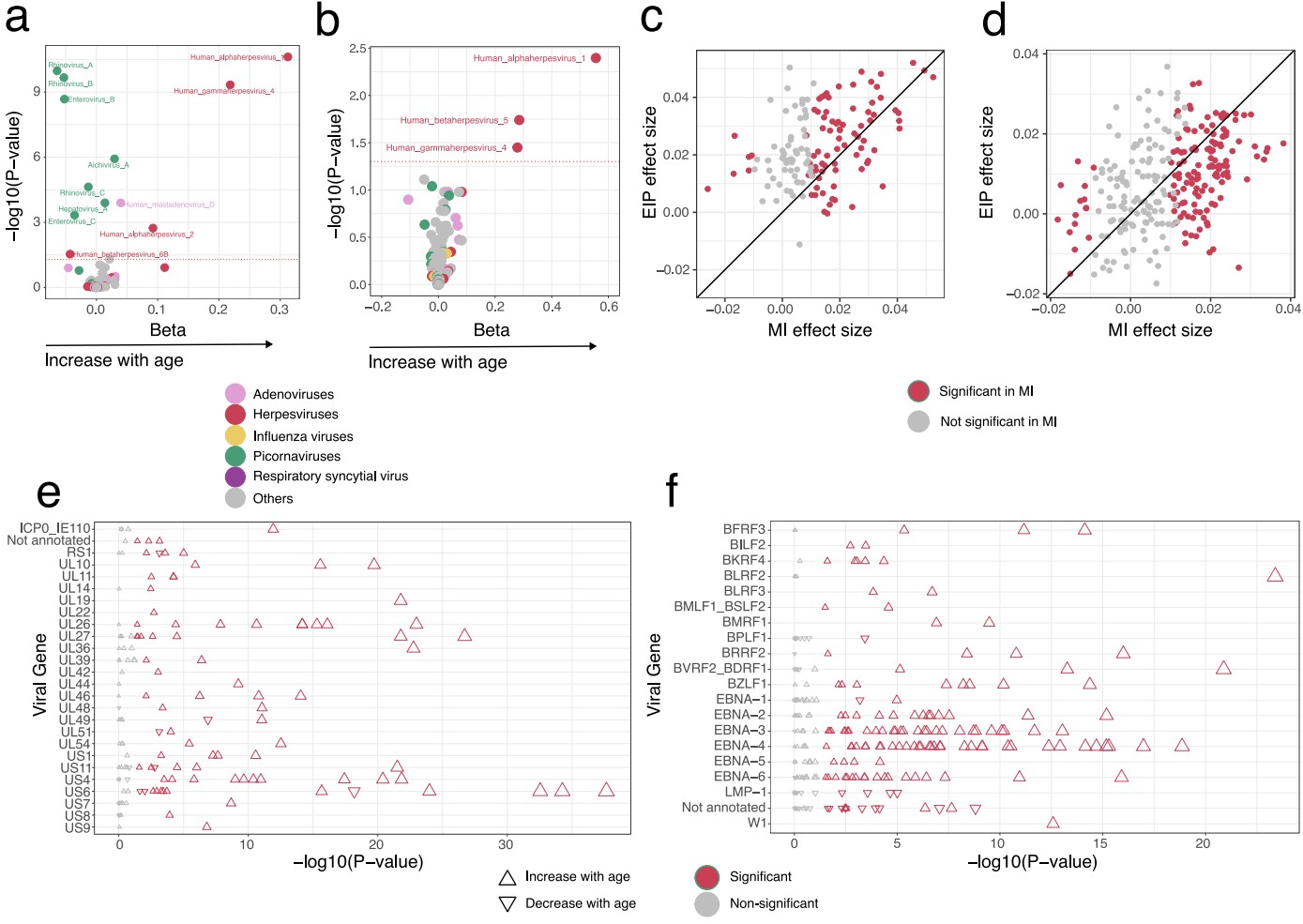

**Extended Data Fig. 6 | Additional age differences in the antiviral antibody repertoire. a,b**, $-\log_{10}$(adjusted *P*-values) against effect sizes for associations between the AVARDA breadth score and age in the Milieu Intérieur (**a**) and EIP (**b**) cohorts (two-sided Wald test). Each point indicates the AVARDA breadth score for a given a virus, colored according its viral family. **c,d**, Effect sizes for the associations between age and HSV-1 (**c**) and EBV (**d**) peptide *Z*-scores in the Milieu Intérieur and EIP cohorts. The black line indicates the identity line. **e,f**, $-\log_{10}$(adjusted *P*-values) of associations between age and (**e**) HSV-1 and (**f**) EBV peptide *Z*-scores, separated by viral protein, in the Milieu Intérieur cohort (two-sided Wald test). The strength and direction of associations are indicated by the size and direction of triangles, respectively. Non-significant associations are colored gray.

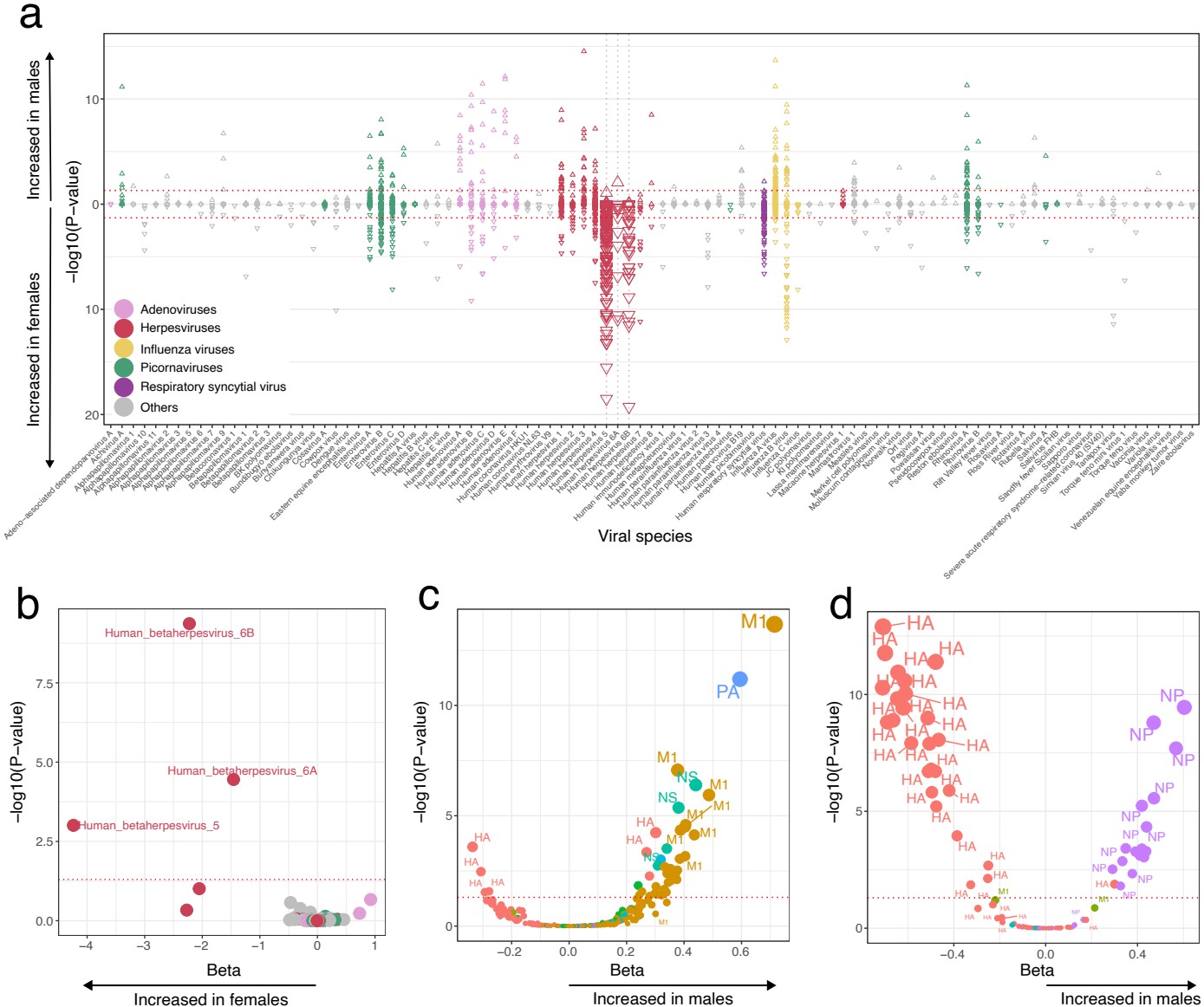

**Extended Data Fig. 7 | Sex differences in the antiviral antibody repertoire.**
**a**, −log₁₀(adjusted *P*-values) and direction of associations between all public peptide *Z*-scores and sex in the Milieu Intérieur cohort (two-sided Wald test). The dashed gray vertical lines indicate viruses for which the AVARDA breadth score is significantly associated with sex. The dashed red horizontal lines indicate the significance threshold (*P*₍adj₎ < 0.05). **b**, −log₁₀(adjusted *P*-values) against effect sizes for associations between the AVARDA breadth score and sex in the Milieu Intérieur cohort (two-sided Wald test). Each point indicates a virus, colored according to its viral family. **c,d**, −log₁₀(adjusted *P*-values) against effect sizes for associations between (**c**) IAV and (**d**) IBV peptide *Z*-scores and sex in the Milieu Intérieur cohort, colored according to the viral protein (two-sided Wald test).

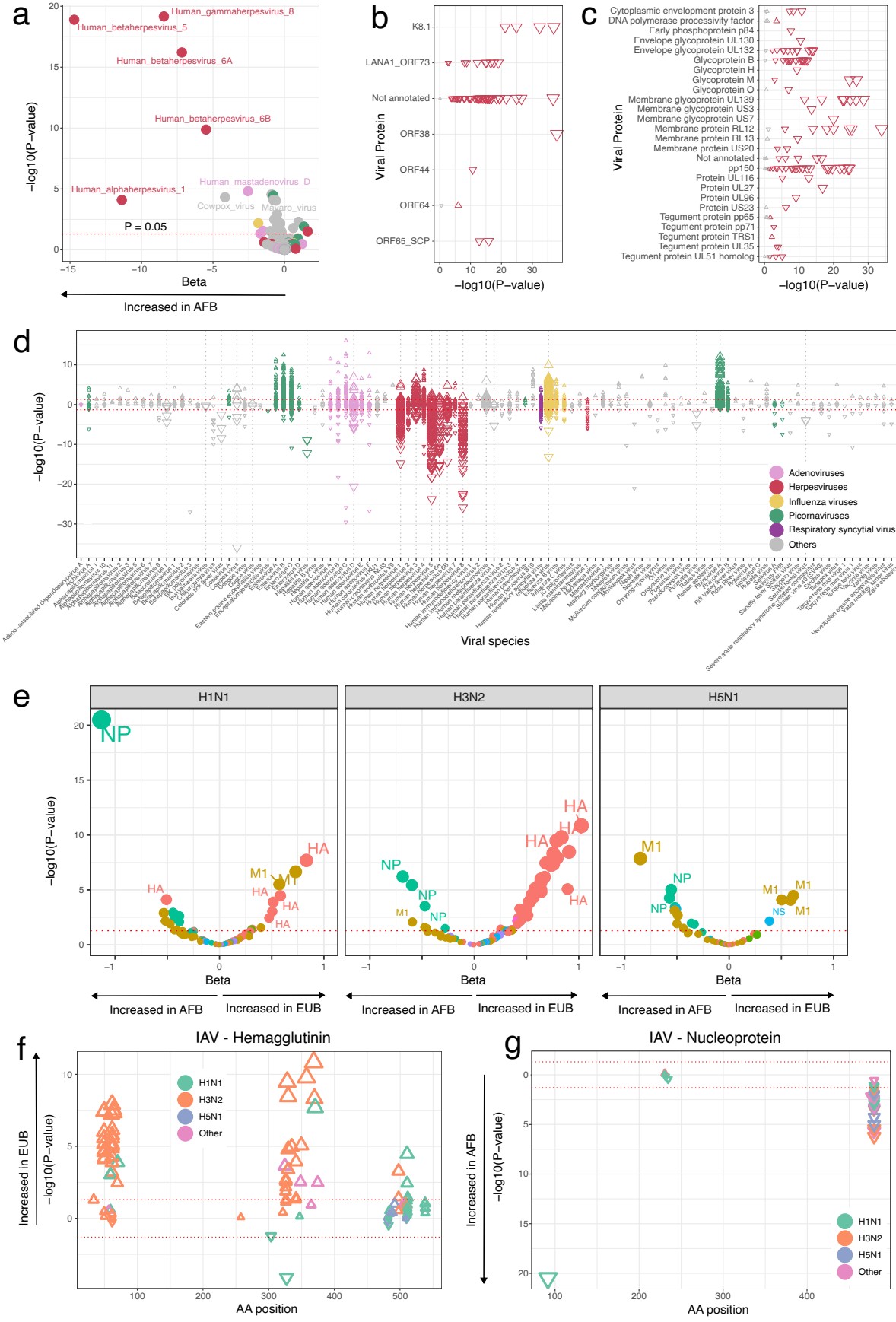

**Extended Data Fig. 8 | See next page for caption.**

**Extended Data Fig. 8 | Additional population differences in the antiviral antibody repertoire. a**, $-\log_{10}$(adjusted $P$-values) against effect sizes for associations between the AVARDA breadth score and continent of birth in the EIP cohort (two-sided Wald test). Each point indicates a virus, colored according to its viral family. **b,c**, $-\log_{10}$(adjusted $P$-values) of associations between continent of birth and HHV-8 (**b**) and CMV (**c**) peptide $Z$-scores, separated by viral protein, in the EIP cohort (two-sided Wald test). The strength and direction of associations are indicated by the size and direction of triangles, respectively. Non-significant associations are colored gray. **d**, $-\log_{10}$(adjusted $P$-values) and direction of associations between all public peptide $Z$-scores and continent of birth in the EIP cohort, separated by viral species, after adjusting on genotypes of the lead variants identified by GWAS (Supplementary Table 7) (two-sided Wald test). The dashed gray vertical lines indicate viruses for which the AVARDA breadth score is significantly associated with continent of birth. The dashed red horizontal lines indicate the significance threshold ($P_{adj} < 0.05$). **e**, $-\log_{10}$(adjusted $P$-values) against effect sizes for associations between IAV peptide $Z$-scores and continent of birth in the EIP cohort, faceted by main IAV subtypes (two-sided Wald test). Colors indicate the viral protein. **f,g**, Amino-acid positions of the midpoint of HA (**f**) and NP (**g**) peptides associated with continent of birth within the full IAV proteins for the EIP cohort (two-sided Wald test). The significance and direction of associations with age are indicated on the y-axis and by the direction of triangles, respectively. The triangle color indicates the IAV subtype.

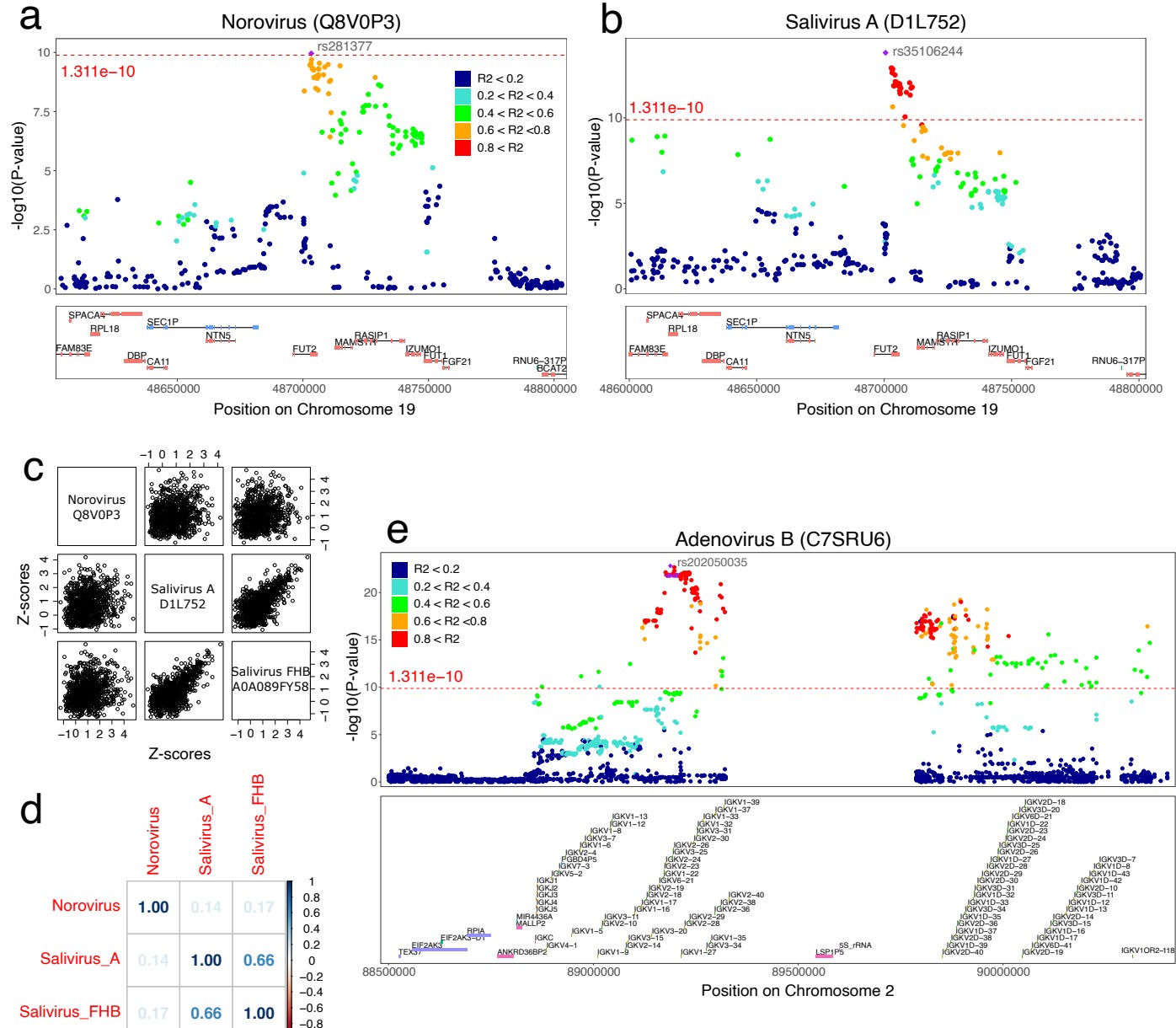

**Extended Data Fig. 9 | Association of genetic variation in the *FUT2* and *IGK* loci with the antiviral antibody repertoire. a,b**, LocusZoom plots showing associations between the *FUT2* locus and antibody reactivity against (**a**) norovirus (UniProt ID: Q8V0P3) and (**b**) salivirus A (UniProt ID: D1L752) (two-sided Wald test). The dashed red line indicates the significance threshold.

**c,d**, Scatter plots (**c**) and correlation matrix (**d**) for the three norovirus and salivirus peptide *Z*-scores most significantly associated with *FUT2* variants. **e**, LocusZoom plot showing associations between the *IGK* locus and antibody reactivity against adenovirus B (UniProt ID: C7SRU6; two-sided Wald test).

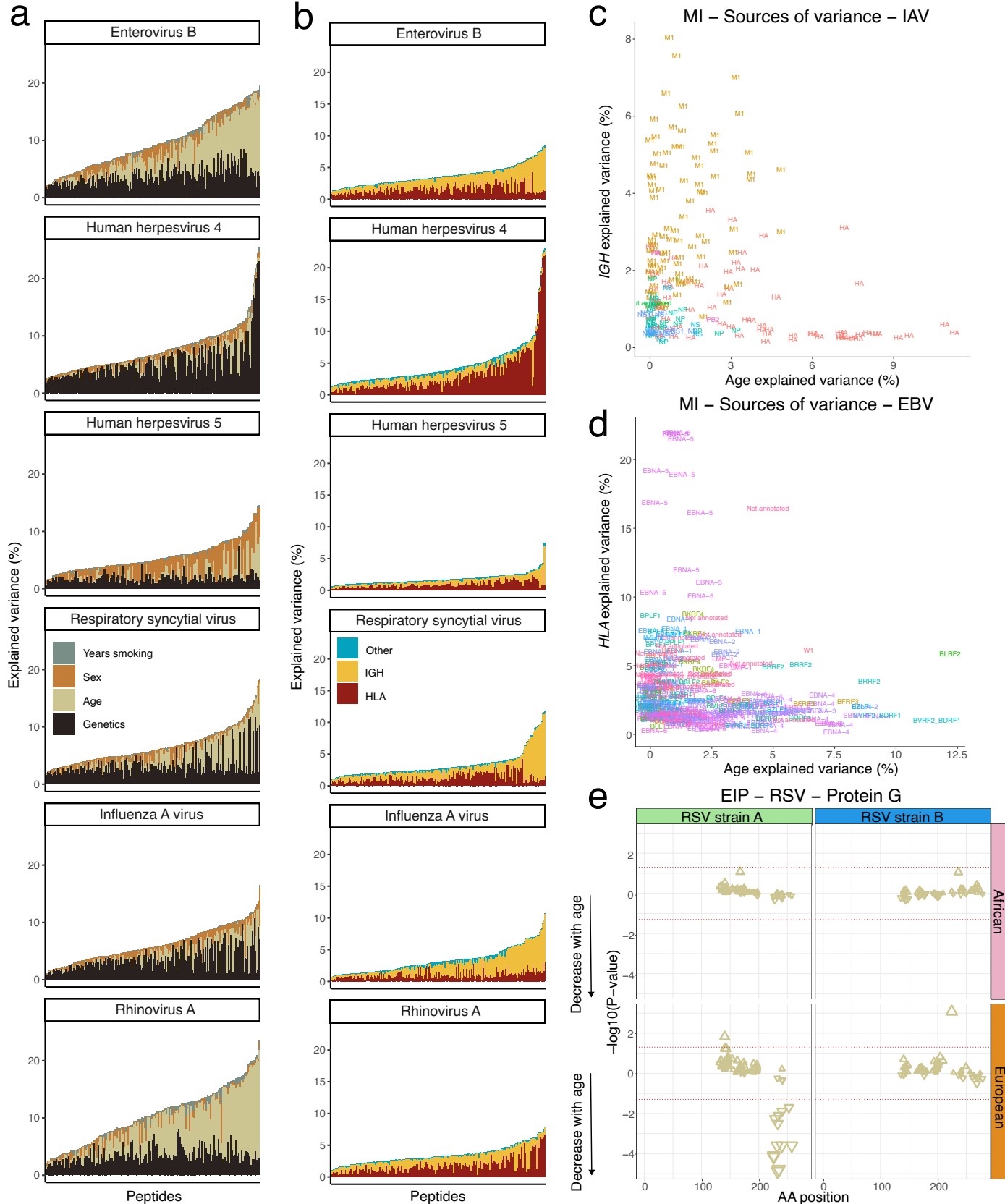

**Extended Data Fig. 10 | See next page for caption.**

**Extended Data Fig. 10 | Variance in the antiviral antibody repertoire explained by individual factors. a**, Proportion of variance explained by age, sex, smoking, and genetics for antibody reactivity against public peptides from the six viruses with the largest number of public peptides in the Milieu Intérieur cohort. Peptides are sorted by total variance explained. **b**, Proportion of variance explained by genetic factors for antibody reactivity against public peptides from the six viruses with the largest number of public peptides in the Milieu Intérieur cohort. Genetic variance is separated by genetic variation in the *HLA* and *IGH* loci and variation external to these loci. Peptides are sorted by total variance

explained. **c,d**, Variance explained by (**c**) age and *IGH* genetic variation for IAV peptides and (**d**) age and *HLA* genetic variation for EBV peptides in the Milieu Intérieur cohort, colored according to viral protein. **e**, Amino-acid positions of the midpoint of protein G peptides associated with age within the full RSV protein G for the EIP cohort (two-sided Wald test). The significance and direction of associations are indicated on the y-axis and by the direction of triangles, respectively. The dashed red horizontal lines indicate the significance threshold ($P_{adj} < 0.05$).

# Reporting Summary

## Statistics

For all statistical analyses, confirm that the following items are present in the figure legend, table legend, main text, or Methods section.

| n/a | Confirmed | |
|---|---|---|
| ☐ | ☒ | The exact sample size (*n*) for each experimental group/condition, given as a discrete number and unit of measurement |
| ☐ | ☒ | A statement on whether measurements were taken from distinct samples or whether the same sample was measured repeatedly |
| ☐ | ☒ | The statistical test(s) used AND whether they are one- or two-sided <br> *Only common tests should be described solely by name; describe more complex techniques in the Methods section.* |
| ☐ | ☒ | A description of all covariates tested |
| ☐ | ☒ | A description of any assumptions or corrections, such as tests of normality and adjustment for multiple comparisons |
| ☐ | ☒ | A full description of the statistical parameters including central tendency (e.g. means) or other basic estimates (e.g. regression coefficient) AND variation (e.g. standard deviation) or associated estimates of uncertainty (e.g. confidence intervals) |
| ☐ | ☒ | For null hypothesis testing, the test statistic (e.g. *F*, *t*, *r*) with confidence intervals, effect sizes, degrees of freedom and *P* value noted <br> *Give P values as exact values whenever suitable.* |
| ☒ | ☐ | For Bayesian analysis, information on the choice of priors and Markov chain Monte Carlo settings |
| ☒ | ☐ | For hierarchical and complex designs, identification of the appropriate level for tests and full reporting of outcomes |
| ☐ | ☒ | Estimates of effect sizes (e.g. Cohen's *d*, Pearson's *r*), indicating how they were calculated |

*Our web collection on statistics for biologists contains articles on many of the points above.*

## Software and code

Policy information about availability of computer code

| Data collection | R v.4.3.0 with packages stats, pcaMethods v.1.94.0 and sva v.3.54.0; RStudio v.2024.04.2 Build 764.pro1; Bowtie v.2; Samtools; BWA-MEM; sambamba; GATK v.4.1; custom code available at https://github.com/h-e-g/virscan_association |
|---|---|
| Data analysis | R v.4.3.0 with packages tidyverse v.2.0.0, AVARDA v.1.0, GWASExactHW v.1.01, GWASTools v.1.48.0, topr v.1.1.10, relaimpo v.2.2-7 and DECIPHER v.2.30.0; RStudio v.2024.04.2 Build 764.pro1; Python v.3.12.2 with packages numpy v.1.26.4, scipy v.1.12.0, pandas v.2.2.1 and scikit-learn v.1.4.1; plink v.2; Eagle v.2.4; Minimac4; ConSurf-DB; PyMOL v.3.0.1; custom code available at https://github.com/h-e-g/virscan_association |

For manuscripts utilizing custom algorithms or software that are central to the research but not yet described in published literature, software must be made available to editors and reviewers. We strongly encourage code deposition in a community repository (e.g. GitHub). See the Nature Portfolio guidelines for submitting code & software for further information.

# Data

Policy information about availability of data

All manuscripts must include a data availability statement. This statement should provide the following information, where applicable:
  - Accession codes, unique identifiers, or web links for publicly available datasets
  - A description of any restrictions on data availability
  - For clinical datasets or third party data, please ensure that the statement adheres to our policy

The VirScan3 PhIP-seq raw and processed data generated in this study have been deposited in the Institut Pasteur data repository, OWEY, which can be accessed via the following links: https://doi.org/10.48802/owey.84rn-jg72 (Milieu Intérieur) and https://doi.org/10.48802/owey.uCQ5VsxD (EvoImmunoPop). All association statistics obtained in this study can also be explored and downloaded from the web browser http://mirepertoire.pasteur.cloud/. All other pseudonymized datasets can be accessed on OWEY by submitting a data access request at https://redcap.pasteur.fr/surveys/?s=ND8TP8MDD3 (Milieu Intérieur) or https://redcap.pasteur.fr/surveys/?s=F3AA7J4M4W8LRNJ4 (EvoImmunoPop). The request will be reviewed by the respective data access committees (DAC). The DAC informs the research participants of the data access request and grants data access if the request is consistent with the informed consent signed by the participants. In particular, research on Milieu Intérieur and EvoImmunoPop datasets is restricted to research on the genetic and environmental determinants of human variation in immune responses. Data access is typically granted two months after request submission.

# Research involving human participants, their data, or biological material

Policy information about studies with human participants or human data. See also policy information about sex, gender (identity/presentation), and sexual orientation and race, ethnicity and racism.

| | |
|---|---|
| Reporting on sex and gender | This study evaluates how sex affects the human antibody repertoire against viruses. We report results by sex when appropriate and assess extensively how women's and men's antibody levels differ. Gender was not studied and self-reported gender was not collected. |
| Reporting on race, ethnicity, or other socially relevant groupings | This study evaluates how the continent of birth affects the human antibody repertoire against viruses. Participants from the EvoImmunoPop study are Belgian residents who were born in either sub-Saharan Africa (AFB) or Europe (EUB). The grouping was determined based on the self-declared place of birth of participants. Associations between antibody reactivity and the continent of birth were interpreted as statistical evidence for geographical differences in exposure to viruses between the two groups, rather than differences caused by race or ethnicity. AFB have been more exposed to herpesvirures, relative to EUB, whereas EUB have been more exposed to influenza A virus and rhinoviruses, owing to diverse climatic, ecological and/or health factors. This study also indicates that AFB and EUB individuals infected by the same virus differ in the viral peptides their antibodies target, implying that global serological surveys based on a single antigen may provide biased results. |
| Population characteristics | 900 individuals from the Milieu Intérieur cohort were included in the present study, including 453 females and 447 males, aged from 20 to 69 years (Fig. 1a). 312 individuals from the EvoImmunoPop study were included, including 100 individuals born in Central Africa (AFB), and 212 born in Europe (EUB), all aged from 20 to 50 years (Fig. 1b). |
| Recruitment | Regarding the Milieu Intérieur cohort, recruitment was conducted in Rennes (France) in 2012-2013, with the aim to recruit 1,000 healthy individuals, including 500 women and 500 men stratified by age in 5 decades of age ([20–29], [30–39], [40–49], [50–59] and [60–69] years, with 200 subjects per stratum). A pre-existing donor database composed of ~110,000 donors was used for pre-screening potential participants in accordance with the study criteria. Additional advertising and website recruitment campaigns were launched in order to complete age and sex strata not sufficiently represented in the donor database. Eligibility was assessed by telephone interview and confirmed during a preliminary information meeting about the objectives of the research. Interested participants that met pre-screening criteria returned for the enrollment visit. During this visit, eligibility criteria were assessed in two stages: first, based on demographical data and clinical examination; and second, by analysis of blood and urine samples that were sent for clinical laboratory testing (Thomas et al., Clin Immunol 2015). Donors were excluded if they have evidence of, or report a history of, neurological, psychiatric or any severe/chronic/recurrent pathological conditions. Other exclusion criteria included a history or evidence of alcohol abuse, recent use of illicit drugs, recent vaccine administration, and recent use of immune modulatory agents. To avoid the influence of hormonal fluctuations in women during the peri-menopausal phase, only pre- or post-menopausal women were included. To avoid the presence of genetic structure in our study population, which would impact upon the power to detect genotype-to-phenotype associations, only individuals whose parents and grandparents were born in continental France were included. Regarding the EvoImmuoPop cohort, recruitment was performed at the Center for Vaccinology (CEVAC) of Ghent University Hospital (Ghent, Belgium). Sampling of related individuals was avoided because relatedness can confound genetic analyses. Individuals with serological signs of past or ongoing infection with human immunodeficiency virus (HIV), hepatitis B virus (HBV) or hepatitis C virus (HCV) were also excluded. |
| Ethics oversight | The Milieu Intérieur study has been approved by the Comité de Protection des Personnes — Ouest 6 (Committee for the Protection of Persons) and by the French Agence Nationale de Sécurité du Médicament (ANSM) and is sponsored by the Institut Pasteur (Pasteur ID-RCB Number: 2012-A00238-35). The study protocol, including inclusion and exclusion criteria for the Milieu Intérieur study, was registered on ClinicalTrials.gov under the study ID NCT01699893. The EvoImmunoPop study was approved by the ethics committee of Ghent University (Belgium, n° B670201214647) and the relevant French authorities (CPP, CCITRS and CNIL). The EvoImmunoPop study was also monitored by the Ethics Board of Institut Pasteur (EVOIMMUNOPOP-281297). |

Note that full information on the approval of the study protocol must also be provided in the manuscript.

# Field-specific reporting

Please select the one below that is the best fit for your research. If you are not sure, read the appropriate sections before making your selection.

☒ Life sciences  ☐ Behavioural & social sciences  ☐ Ecological, evolutionary & environmental sciences

For a reference copy of the document with all sections, see nature.com/documents/nr-reporting-summary-flat.pdf

# Life sciences study design

All studies must disclose on these points even when the disclosure is negative.

| | |
|---|---|
| Sample size | We found by simulations that the sample size of the primary cohort (Milieu Intérieur; n = 900) provides 95% power to detect a medium non-genetic or genetic effect, corresponding to 0.65 standard deviation of the peptide Z-score. |
| Data exclusions | As the present study focuses on human humoral responses against viruses, we discarded peptides from bacteria, fungi, and allergens from the VirScan library, resulting in 99,460 viral peptides. We also excluded VirScan data for peptides showing largely different Z-score values between experimental replicates. Discordant values were set to missing and peptides with >50% missing values were discarded. Finally, peptides with duplicated Uniprot entries were removed, leaving 97,975 peptides in the Milieu Intérieur dataset and 97,923 in the EvoImmunoPop dataset for the remaining analyses. |
| Replication | Measurement reproducibility was assessed by performing two replicates for each individual. Measurements with low reproducibility across individuals were excluded. All the results discovered in the Milieu Intérieur cohort were tested for replication in the EvoImmunoPop cohort, whenever possible. Associations with age were not replicated for a few viruses and peptides, likely because of the smaller size of the replication cohort (EvoImmunoPop; n = 312). Associations with sex were not tested for replication, as the replication cohort includes males only. Associations with the continent of birth were tested in the EvoImmunoPop cohort only, as the Milieu Intérieur cohort includes only participants born in France. Genetic associations were all tested for replication, except at the IGH and IGK loci, which were sequenced in the Milieu Intérieur cohort only. |
| Randomization | Prior to PhIP-seq experiments, samples were distributed in processing plates according to sample identifiers. As sample identifiers were partially correlated with age and continent of birth, and the PhIP datasets were affected by plate effects (Supplementary Fig. 1), VirScan Z-scores were corrected for plate effects using the ComBat function from the sva R package, providing age and continent-of-birth variables as biological factors with true effects on the data. |
| Blinding | Investigators were blinded to group allocation during data collection. |

# Reporting for specific materials, systems and methods

We require information from authors about some types of materials, experimental systems and methods used in many studies. Here, indicate whether each material, system or method listed is relevant to your study. If you are not sure if a list item applies to your research, read the appropriate section before selecting a response.

## Materials & experimental systems

| n/a | Involved in the study |
|---|---|
| ☒ | ☐ Antibodies |
| ☒ | ☐ Eukaryotic cell lines |
| ☒ | ☐ Palaeontology and archaeology |
| ☒ | ☐ Animals and other organisms |
| ☐ | ☒ Clinical data |
| ☒ | ☐ Dual use research of concern |
| ☒ | ☐ Plants |

## Methods

| n/a | Involved in the study |
|---|---|
| ☒ | ☐ ChIP-seq |
| ☒ | ☐ Flow cytometry |
| ☒ | ☐ MRI-based neuroimaging |

# Clinical data

Policy information about clinical studies

All manuscripts should comply with the ICMJE guidelines for publication of clinical research and a completed CONSORT checklist must be included with all submissions.

| | |
|---|---|
| Clinical trial registration | NCT01699893 |
| Study protocol | The Milieu Intérieur study protocol is described in Thomas et al., Clin Immunol 2015. The EvoImmunoPop study protocol is described in Quach*, Rotival*, Pothlichet*, Loh* et al., Cell 2016. |
| Data collection | Blood samples were collected from the Milieu Intérieur healthy, fasting donors every working day from 8AM to 11AM, from September 2012 to August 2013, at Biotrial (Rennes, France). Tracking procedures were established in order to ensure delivery to Institut Pasteur (Paris) within 6 hours of blood draw. Upon receipt, samples were kept at room temperature until plasma preparation |

| Outcomes | and DNA extraction. Blood samples were collected from the EvoImmunoPop donors on EDTA-blood collection tubes at the Center for Vaccinology (CEVAC) of Ghent University Hospital (Ghent, Belgium), from 2012 to 2013. |
|---|---|
| | The aim of the present study was to evaluate how demographic and genetic factors affect the human antibody repertoire against viruses. The primary outcomes of this study were the quantitative measures of antibody reactivity against peptides of all known viruses infecting humans. To assess these measures, we employed PhIP-Seq using the VirScan V3 library, a viral peptidome scanning method based on bacteriophage display and immuno-precipitation. The phage library was incubated with plasma samples normalized to total IgG concentration and controls (bead samples) to form IgG-phage immunocomplexes. The immunocomplexes were then captured by magnetic beads, lysed, and sent to next-generation sequencing. The secondary outcomes were demographic variables, including age, sex, health-related habits, and vaccination and medical history, which were obtained through structured questionnaires, and genetic variables, which were obtained by SNP array genotyping and genotype imputation. |

# Plants

| Seed stocks | n/a |
|---|---|
| Novel plant genotypes | n/a |
| Authentication | n/a |

