## [Peer Review File · Nature Immunology]

Demographic and genetic factors shape the epitope specificity of the human antibody repertoire against viruses

Corresponding Author: Dr Etienne Patin

Version 0:

Reviewer comments:

Reviewer #1

(Remarks to the Author)

The study by Olin et al applied VirScan to 2 different cohorts of 1212 adults to understand how intrinsic and external factors influence antiviral antibody specificity. They demonstrate virus and sometimes epitope-specific impacts of age, sex, continent of birth as well as smoking status. They further link the VirScan data to GWAS analyses and identify associations between HLA, IGH, IGK and FUT2 variants with specific antibody measurements. While a lot of the findings are confirmatory of past associations (as appropriately noted by the authors), the study also makes several novel contributions to the field. The dataset is likely to be a very useful resource for future analyses across the field of immunology. The results are already available on a web-based browser and the data/code available through relevant repositories.

While the authors clearly acknowledge the limitations of VirScan (e.g linear vs structural epitopes, binding vs neutralising antibodies) and have compared VirScan with 'conventional' serology approaches, I think it is pertinent to validate some of the key observations using ELISA (the data may already be part of the ELISA work performed but it should be presented clearly as a separate supplementary figure). Specifically, it would be important to show by ELISA that men have higher antibodies to flu NP than women, and women have higher antibodies to HA. Similarly, the differences in ENBA4 and EBNA6 between different continents of birth, and the Rhinovirus B difference between smoker/non-smokers. Most of these proteins are commercially available so this should be feasible. The same can be done with peptide-specific differences between other comparisons using peptides in ELISAs.

Given that some of these findings have the potential to initiate future work to investigate their mechanistic bases, I think it is critical to validate them at this point.

Other minor points:

- It would be beneficial to also note that the effects of 'continent of birth' may also be related to genetics? For example, HLA genes are differentially represented in different populations, so the effects associated with continent of birth may not necessarily reflect differences in viral exposure.
- In the methods section for ELISA/Luminex (lines 729 and 742 onwards) please provide more details in the source of antigens used?
- There is a methods section for a KREC assay – but I was unable to locate the relevant data/results in the manuscript? Please clarify.
- Please clarify which structures were used in figure 2 (e.g. PDBxxx)
- In figure 1 c and f – should the y-axis be 'proportion of donors' instead of 'peptides'?

Reviewer #2

(Remarks to the Author)

This manuscript reports a large-scale analysis of plasma IgG binding to a library of 97,000 phage-displayed peptides from viral families of clinical significance (VirScan V3), using 900 specimens from members of a European-ancestry cohort (MI) with ages 20-69 years, and 312 specimens from individuals of a cohort of Belgian residents of either Central African or European birth (EIP). The results evaluate correlations between the concentration of plasma antibodies binding peptides from the phage displayed panel and demographic, lifestyle and genetic data from the participants. The study reports

correlations between plasma antibody binding of viral peptides and participant age, sex, and continent of birth, with strong associations between smoking status and plasma antibodies against peptides from viruses such as rhinoviruses. A very interesting part of the manuscript is its GWAS analysis of genotype associations (including germline sequence variants within the immunoglobulin heavy chain and light chain loci, which are rarely examined) with antibody binding to viral peptides, showing strongest linkage between antibody binding to viral peptides and genetic variants in the MHC, IGH, IGK and FUT2 loci.

The study reconfirms some previously reported effects of sex, age and population groups on antibody titers against certain viruses. It expands upon previously-reported genetic associations with antibody titers against particular viruses and the MHC, IGH and FUT2 loci.

Strengths of the study are its breadth of coverage of virus types with the peptide antigens, the relatively large clinical cohorts with detailed demographic and lifestyle data, and the pairing of broad serological testing with genome-wide genetic association testing. The associations detected between participant sex or age and the proteins and domains targeted by antibodies are novel, interesting and potentially clinically important. The experimental and data analysis approaches in the study appear well thought-out and executed, to the extent to which those can be assessed based on the text and figures of the manuscript and supplementary documents.

Major limitations of the study are the unclear commitment of the authors to truly share the data from the manuscript if it were to be published; the uncertain importance (clinically or otherwise) of antibody titers against isolated linear peptide antigens; the lack of functionally important data such as viral neutralization titers or other antibody functions from the PhIPSeq assay; and the limited degree to which new potential findings from the PhIPSeq assay are validated and replicated with immunoassays using intact viral antigens or antigen domains when those are implicated by the peptide binding assay. The previously-reported AVARDA algorithm used to attempt to correct for possible cross-reactivity between peptides in the library only partly addresses concerns about analytical accuracy of the assay, as the AVARDA approach seems to have been used previously to improve seropositivity status assessment for individual viruses with PhIPSeq, but not to enable quantitative comparison of antibody titers against particular structurally-intact protein domains within a viral antigen, or between different viral antigens.

Major Concerns

1. The greatest concern with the manuscript is whether there will actually be open data sharing to allow other researchers to evaluate the data, replicate the analyses and independently assess the conclusions of the study. With complicated data types and analyses such as those presented in the study, full data sharing is critically important if the data, results and conclusions are ever going to be assessed by anyone other than the authors. The data sharing proposed is inadequate. Browsing on the "mireperture" graphical display system does not substitute for making the underlying data available to other scientists.

- The OWY data sharing portal at the Institut Pasteur does not appear to be independent of the researchers who conducted this work, and the data sharing link gives this message at the site: "Please fill in the following document <https://redcap.pasteur.fr/surveys/?s=ND8TP8MDD3> in order to request access to the desired data. Milieu Intérieur Consortium members will discuss your request during the next Data Access Committee and send you back an answer (positive or negative) or request for complementary information."
- The redcap form outlines many restrictions on data access and use, requires that other researchers comply with a "publication charter" that is not cited or linked on the document.
- The redcap form does not list replication or evaluation of the present manuscript analysis as one of the valid reasons for requesting the study data
- The available dataset appears to be only for the Milieu Intérieur participants and not for the EvolImmunoPop participants in the study.
- Under what conditions would the authors not share the full and complete data from this study to enable other researchers to attempt to replicate and evaluate the analyses and conclusions? There would be very little risk of re-identification of participants or other potential harms based on serology results, basic demographics and lifestyle information.
- The data available at the site is listed as already processed z-scores or binary positive/negative results for the peptides in the PhIP-Seq analysis, rather than the actual sequence data that would enable other researchers to attempt to replicate the analysis results. The manuscript states that "raw and processed data" are available at the sharing site, which is not consistent with the description at the OWY site.
- This review represents a request to the Milieu Intérieur Consortium members to share the PhIP-Seq data and participant metadata for the purpose of allowing other scientists to re-analyze the data to attempt to replicate the results and conclusions in this manuscript. The data should be deposited in an archive outside of author control, like ImmPORT or SRA, with no or minimal restrictions on access.

2. Major and more novel results (male/female differences in influenza antigen binding to HA vs. NP or MP1, and HA head vs. stalk domain binding; rhinovirus antibody binding in smokers compared to non-smokers) should be validated using independent immunoassays with intact antigen proteins or protein domains.

3. For the major and more novel results (male/female differences in influenza antigen binding to HA vs. NP or MP1, and HA head vs. stalk domain binding; rhinovirus antibody binding in smokers compared to non-smokers), the magnitude of the differences in antibody quantities measured should be reported, rather than only the P-values. What are the effect sizes of these differences? Presumably, they could be tiny differences yet still have very low P-values given the large data set size.

4. Correction for batch effects in the experiment are mentioned. How large were any batch effects between sample batches tested, and do the sample batches have any relationship to age, sex or population group?

5. Are age-related serology associations independent of the population groups EB vs. AB that differ in their age distributions? Similarly, were the correlations between antibody levels and sex independent of age, or is there any interaction between age and sex? Or smoking and sex or age (e.g., more smoking in males?)

6. In the influenza analyses (Page 6) "These differences were not driven by age-related variations in exposure to different IAV subtypes, as both positive and negative associations were observed within the same IAV subtypes for HA and MP1"

- Could this be due to different imprinting in old and young for peptides that are conserved or vary across older flu strains or more recent ones (e.g., older participants giving relatively stronger responses to peptides that are more conserved between older and more recent strains?)

- The authors should test whether the degree of sequence conservation for these peptide regions in flu variants from prior decades is correlated with the age-related differences seen in the serology data

- Would the AVARDA algorithm be able to account for prior flu strains participants might have been infected by in prior decades?

- Does greater flu HA vs. MP1 binding in women compared to men reflect similar response to infection between men and women, but better vaccine responses in women or more frequent vaccination of women (which would probably preferentially increase the HA responses)? The influenza vaccination rates are described as not differing between women and men, but does this take into account how often and how recently women and men were vaccinated relative to the blood sampling for this study?

- How do the flu antibody levels and proteins or domains targeted differ between men and women that have been vaccinated recently, compared to between men and women who reported no vaccination?

7. It is surprising that no viral serology results were correlated with genetic variants in the immunoglobulin lambda locus, whereas kappa locus variants were highly significant. Was there any difference in the extent or interpretability of genotyping in the lambda locus compared to kappa?

8. For RSV results, what is known about the distribution of strain A in earlier decades? Is it possible that older people would have had greater exposure to it?

Reviewer #3

(Remarks to the Author)

This is a carefully executed work using state of the art analytics. The analysis uses 97000 peptide serum reactivity from more than 1000 individuals using VirScan, genomics and statistics. Below are some comments that may help in revising the current draft.

1. Population size-effect. The various associations are statistically robust, but as expected for the type of analytics and the presumed complex factors determining viral reactivity, the explained variance is modest (~3.5% for non-genetic and ~3.5% for genetic factors). This should be discussed to make sure that the readership is not misled.

2. Coverage of viral peptidome. From Extended data Fig 1, it is unclear how the difference in representation of viruses in VirScan (whether due to viral genome size or representativity/coverage) impacts the association statistics and conclusion. This should be addressed.

3. VirScan/Avarda. When looking at the implementation in the literature

([https://www.thelancet.com/journals/ebiom/article/PIIS2352-3964\(21\)00541-7/fulltext](https://www.thelancet.com/journals/ebiom/article/PIIS2352-3964(21)00541-7/fulltext)) , it is unclear how best to present Avarda. One could assume that there would be no need to use native VirScan data and rather resort to the Avarda deconvoluted results. The present work uses both outputs. The Avarda breath score needs more explanation.

4. The data on Saliviruses and FUT2 is a compelling output of the technology as a path to discover new associations. It may not be feasible for a revision of the draft, but it would have been great to see a proof of usage of the receptor. However, it is not completely clear whether the virus has been isolated.

5. The genome association analyses are conceptually similar to those described as genome-to-genome studies (original publication <https://elifesciences.org/articles/01123>). It would be worth commenting on the shared methodologies and concepts.

6. Thanks for the discussion on limits of the study as well as for the release of data and code and the UI

Decision Letter:

24th Jan 2025

Dear Etienne & Lluis,

Thank you for providing a point-by-point response to the referees' comments on your manuscript entitled, "Demographic and genetic factors shape the epitope specificity of the human antibody repertoire against viruses". As noted previously, while they find your work of considerable potential interest, they have raised quite substantial concerns that must be addressed. In light of these comments, we cannot accept the current manuscript for publication, but would be very interested in considering a revised version that addresses these serious concerns along the lines proposed in your point-by-point rebuttal.

We invite you to submit a substantially revised manuscript, however please bear in mind that we will be reluctant to approach the referees again in the absence of major revisions.

Specifically, the revision should include new experiments to validate some of the main findings in your study, including: ELISA and/or Luminex assays to measure

- (i) age differences in antibody reactivity to HA and M1 IAV epitopes,
- (ii) sex differences in antibody reactivity to HA, NP and M1 IAV epitopes,
- (iii) differences in antibody reactivity to EBNA-4 and EBNA-6 epitopes of EBV between Belgians born in Africa or Europe, and
- (iv) differences in antibody reactivity to rhinovirus B between smokers and never-/ex-smokers

Please include the additional textual clarifications & limitations as indicated in your response letter.

When you revise your manuscript, please take into account all reviewer and editor comments, please highlight all changes in the manuscript text file in Microsoft Word format.

* If you have not done so already please begin to revise your manuscript so that it conforms to our Resource format instructions at <http://www.nature.com/ni/authors/index.html>. Refer also to any guidelines provided in this letter.

The Reporting Summary can be found here:

<https://www.nature.com/documents/hr-reporting-summary.pdf>

Extended Data figures and tables are online-only (appearing in the online PDF and full-text HTML version of the paper), peer-reviewed display items that provide essential background to the Article but are not included in the printed version of the paper due to space constraints or being of interest only to a few specialists. A maximum of ten Extended Data display items (figures and tables) is typically permitted. When re-submitting your manuscript, please ensure that any supplementary figures and tables that are more critical to the manuscript's conclusions are converted to Extended data to increase these data's visibility.

Link Redacted

If you wish to submit a suitably revised manuscript we would hope to receive it within 6 months. If you cannot send it within this time, please let us know. We will be happy to consider your revision so long as nothing similar has been accepted for publication at Nature Immunology or published elsewhere.

Nature Immunology is committed to improving transparency in authorship. As part of our efforts in this direction, we are now requesting that all authors identified as 'corresponding author' on published papers create and link their Open Researcher

and Contributor Identifier (ORCID) with their account on the Manuscript Tracking System (MTS), prior to acceptance. ORCID helps the scientific community achieve unambiguous attribution of all scholarly contributions. You can create and link your ORCID from the home page of the MTS by clicking on 'Modify my Springer Nature account'. For more information please visit www.springernature.com/orcid.

Thank you for the opportunity to review your work.

Kind regards,

Laurie

Laurie A. Dempsey, Ph.D.
Senior Editor
Nature Immunology
l.dempsey@us.nature.com
ORCID: 0000-0002-3304-796X

Reviewers' Comments:

Reviewer #1 (Remarks to the Author):

The study by Olin et al applied VirScan to 2 different cohorts of 1212 adults to understand how intrinsic and external factors influence antiviral antibody specificity. They demonstrate virus and sometimes epitope-specific impacts of age, sex, continent of birth as well as smoking status. They further link the VirScan data to GWAS analyses and identify associations between HLA, IGH, IGK and FUT2 variants with specific antibody measurements. While a lot of the findings are confirmatory of past associations (as appropriately noted by the authors), the study also makes several novel contributions to the field. The dataset is likely to be a very useful resource for future analyses across the field of immunology. The results are already available on a web-based browser and the data/code available through relevant repositories.

While the authors clearly acknowledge the limitations of VirScan (e.g linear vs structural epitopes, binding vs neutralising antibodies) and have compared VirScan with 'conventional' serology approaches, I think it is pertinent to validate some of the key observations using ELISA (the data may already be part of the ELISA work performed but it should be presented clearly as a separate supplementary figure). Specifically, it would be important to show by ELISA that men have higher antibodies to flu NP than women, and women have higher antibodies to HA. Similarly, the differences in ENBA4 and EBNA6 between different continents of birth, and the Rhinovirus B difference between smoker/non-smokers. Most of these proteins are commercially available so this should be feasible. The same can be done with peptide-specific differences between other comparisons using peptides in ELISAs.

Given that some of these findings have the potential to initiate future work to investigate their mechanistic bases, I think it is critical to validate them at this point.

Other minor points:

- It would be beneficial to also note that the effects of 'continent of birth' may also be related to genetics? For example, HLA genes are differentially represented in different populations, so the effects associated with continent of birth may not necessarily reflect differences in viral exposure.
- In the methods section for ELISA/Luminex (lines 729 and 742 onwards) please provide more details in the source of antigens used?
- There is a methods section for a KREC assay – but I was unable to locate the relevant data/results in the manuscript? Please clarify.
- Please clarify which structures were used in figure 2 (e.g. PDBxxx)
- In figure 1 c and f – should the y-axis be 'proportion of donors' instead of 'peptides'?

Reviewer #2 (Remarks to the Author):

This manuscript reports a large-scale analysis of plasma IgG binding to a library of 97,000 phage-displayed peptides from viral families of clinical significance (VirScan V3), using 900 specimens from members of a European-ancestry cohort (MI) with ages 20-69 years, and 312 specimens from individuals of a cohort of Belgian residents of either Central African or European birth (EIP). The results evaluate correlations between the concentration of plasma antibodies binding peptides from the phage displayed panel and demographic, lifestyle and genetic data from the participants. The study reports correlations between plasma antibody binding of viral peptides and participant age, sex, and continent of birth, with strong associations between smoking status and plasma antibodies against peptides from viruses such as rhinoviruses. A very interesting part of the manuscript is its GWAS analysis of genotype associations (including germline sequence variants within the immunoglobulin heavy chain and light chain loci, which are rarely examined) with antibody binding to viral

peptides, showing strongest linkage between antibody binding to viral peptides and genetic variants in the MHC, IGH, IGK and FUT2 loci.

The study reconfirms some previously reported effects of sex, age and population groups on antibody titers against certain viruses. It expands upon previously-reported genetic associations with antibody titers against particular viruses and the MHC, IGH and FUT2 loci.

Strengths of the study are its breadth of coverage of virus types with the peptide antigens, the relatively large clinical cohorts with detailed demographic and lifestyle data, and the pairing of broad serological testing with genome-wide genetic association testing. The associations detected between participant sex or age and the proteins and domains targeted by antibodies are novel, interesting and potentially clinically important. The experimental and data analysis approaches in the study appear well thought-out and executed, to the extent to which those can be assessed based on the text and figures of the manuscript and supplementary documents.

Major limitations of the study are the unclear commitment of the authors to truly share the data from the manuscript if it were to be published; the uncertain importance (clinically or otherwise) of antibody titers against isolated linear peptide antigens; the lack of functionally important data such as viral neutralization titers or other antibody functions from the PhIPSeq assay; and the limited degree to which new potential findings from the PhIPSeq assay are validated and replicated with immunoassays using intact viral antigens or antigen domains when those are implicated by the peptide binding assay. The previously-reported AVARDA algorithm used to attempt to correct for possible cross-reactivity between peptides in the library only partly addresses concerns about analytical accuracy of the assay, as the AVARDA approach seems to have been used previously to improve seropositivity status assessment for individual viruses with PhIPSeq, but not to enable quantitative comparison of antibody titers against particular structurally-intact protein domains within a viral antigen, or between different viral antigens.

Major Concerns

1. The greatest concern with the manuscript is whether there will actually be open data sharing to allow other researchers to evaluate the data, replicate the analyses and independently assess the conclusions of the study. With complicated data types and analyses such as those presented in the study, full data sharing is critically important if the data, results and conclusions are ever going to be assessed by anyone other than the authors. The data sharing proposed is inadequate. Browsing on the "mirepore" graphical display system does not substitute for making the underlying data available to other scientists.

- The OWY data sharing portal at the Institut Pasteur does not appear to be independent of the researchers who conducted this work, and the data sharing link gives this message at the site: "Please fill in the following document <https://redcap.pasteur.fr/surveys/?s=ND8TP8MDD3> in order to request access to the desired data. Milieu Intérieur Consortium members will discuss your request during the next Data Access Committee and send you back an answer (positive or negative) or request for complementary information."
- The redcap form outlines many restrictions on data access and use, requires that other researchers comply with a "publication charter" that is not cited or linked on the document.
- The redcap form does not list replication or evaluation of the present manuscript analysis as one of the valid reasons for requesting the study data
- The available dataset appears to be only for the Milieu Intérieur participants and not for the EvolImmunoPop participants in the study.
- Under what conditions would the authors not share the full and complete data from this study to enable other researchers to attempt to replicate and evaluate the analyses and conclusions? There would be very little risk of re-identification of participants or other potential harms based on serology results, basic demographics and lifestyle information.
- The data available at the site is listed as already processed z-scores or binary positive/negative results for the peptides in the PhIP-Seq analysis, rather than the actual sequence data that would enable other researchers to attempt to replicate the analysis results. The manuscript states that "raw and processed data" are available at the sharing site, which is not consistent with the description at the OWY site.
- This review represents a request to the Milieu Intérieur Consortium members to share the PhIP-Seq data and participant metadata for the purpose of allowing other scientists to re-analyze the data to attempt to replicate the results and conclusions in this manuscript. The data should be deposited in an archive outside of author control, like ImmPORT or SRA, with no or minimal restrictions on access.

2. Major and more novel results (male/female differences in influenza antigen binding to HA vs. NP or MP1, and HA head vs. stalk domain binding; rhinovirus antibody binding in smokers compared to non-smokers) should be validated using independent immunoassays with intact antigen proteins or protein domains.

3. For the major and more novel results (male/female differences in influenza antigen binding to HA vs. NP or MP1, and HA head vs. stalk domain binding; rhinovirus antibody binding in smokers compared to non-smokers), the magnitude of the differences in antibody quantities measured should be reported, rather than only the P-values. What are the effect sizes of these differences? Presumably, they could be tiny differences yet still have very low P-values given the large data set size.

4. Correction for batch effects in the experiment are mentioned. How large were any batch effects between sample batches tested, and do the sample batches have any relationship to age, sex or population group?

5. Are age-related serology associations independent of the population groups EB vs. AB that differ in their age

distributions? Similarly, were the correlations between antibody levels and sex independent of age, or is there any interaction between age and sex? Or smoking and sex or age (e.g., more smoking in males?)

6. In the influenza analyses (Page 6) “These differences were not driven by age-related variations in exposure to different IAV subtypes, as both positive and negative associations were observed within the same IAV subtypes for HA and MP1”
- Could this be due to different imprinting in old and young for peptides that are conserved or vary across older flu strains or more recent ones (e.g., older participants giving relatively stronger responses to peptides that are more conserved between older and more recent strains?)
 - The authors should test whether the degree of sequence conservation for these peptide regions in flu variants from prior decades is correlated with the age-related differences seen in the serology data
 - Would the AVARDA algorithm be able to account for prior flu strains participants might have been infected by in prior decades?
 - Does greater flu HA vs. MP1 binding in women compared to men reflect similar response to infection between men and women, but better vaccine responses in women or more frequent vaccination of women (which would probably preferentially increase the HA responses)? The influenza vaccination rates are described as not differing between women and men, but does this take into account how often and how recently women and men were vaccinated relative to the blood sampling for this study?
 - How do the flu antibody levels and proteins or domains targeted differ between men and women that have been vaccinated recently, compared to between men and women who reported no vaccination?
7. It is surprising that no viral serology results were correlated with genetic variants in the immunoglobulin lambda locus, whereas kappa locus variants were highly significant. Was there any difference in the extent or interpretability of genotyping in the lambda locus compared to kappa?
8. For RSV results, what is known about the distribution of strain A in earlier decades? Is it possible that older people would have had greater exposure to it?

Reviewer #3 (Remarks to the Author):

This is a carefully executed work using state of the art analytics. The analysis uses 97000 peptide serum reactivity from more than 1000 individuals using VirScan, genomics and statistics. Below are some comments that may help in revising the current draft.

1. Population size-effect. The various associations are statistically robust, but as expected for the type of analytics and the presumed complex factors determining viral reactivity, the explained variance is modest (~3.5% for non-genetic and ~3.5% for genetic factors). This should be discussed to make sure that the readership is not misled.
2. Coverage of viral peptidome. From Extended data Fig 1, it is unclear how the difference in representation of viruses in VirScan (whether due to viral genome size or representativity/coverage) impacts the association statistics and conclusion. This should be addressed.
3. VirScan/Avarda. When looking at the implementation in the literature ([https://www.thelancet.com/journals/ebiom/article/PIIS2352-3964\(21\)00541-7/fulltext](https://www.thelancet.com/journals/ebiom/article/PIIS2352-3964(21)00541-7/fulltext)), it is unclear how best to present Avarda. One could assume that there would be no need to use native VirScan data and rather resort to the Avarda deconvoluted results. The present work uses both outputs. The Avarda breath score needs more explanation.
4. The data on Saliviruses and FUT2 is a compelling output of the technology as a path to discover new associations. It may not be feasible for a revision of the draft, but it would have been great to see a proof of usage of the receptor. However, it is not completely clear whether the virus has been isolated.
5. The genome association analyses are conceptually similar to those described as genome-to-genome studies (original publication <https://elifesciences.org/articles/01123>). It would be worth commenting on the shared methodologies and concepts.
6. Thanks for the discussion on limits of the study as well as for the release of data and code and the UI

Version 1:

Reviewer comments:

Reviewer #1

(Remarks to the Author)

The authors have addressed all of my comments and have considerably strengthened the manuscript.

Reviewer #2

(Remarks to the Author)

In the revisions of this manuscript, the authors clarified the nature of their data sharing plans, and attempted to validate their main conclusions from the prior manuscript, using a Luminex assay with intact proteins. They also evaluated the impact of batch effects on their results.

The authors' description of their data sharing plans and the constraints imposed by regulatory requirements have alleviated my prior concerns.

Major concerns persist in the revised manuscript. In the response to reviews and description of new experiments attempting to validate the VirScan results using Luminex or ELISA assays, the authors state:

“Specifically, we confirmed that:

- (i) antibody titers against the less conserved globular head (β Luminex = -0.033 ; P Luminex = 4.42×10^{-76}) and the more conserved stalk (β Luminex = 0.014 ; P Luminex = 8.62×10^{-15}) domains of IAV hemagglutinin change in opposite directions with age;
- (ii) antibody titers against less conserved positions 200-250 of the IAV M1 protein decrease with age (β Luminex = -0.0068 , P Luminex = 4.04×10^{-4});
- (iii) antibodies preferentially target the IAV HA protein in females (β Luminex = -0.10 ; P Luminex = 0.038), whereas antibodies in males target M1 (β Luminex = 0.092 ; P Luminex = 0.02) and NP 2 (β Luminex = 0.10 ; P Luminex = 0.03) proteins from IAV and IBV, respectively;
- (iv) antibodies of AFB preferentially target EBNA-4 peptides deriving from the AG876 strain (β = -1.28 , P = 9.12×10^{-14}) whereas antibodies of EUB target EBNA-6 peptides from the B95-8 strain (β = 1.02 , P = 8.2×10^{-13}).”

Unfortunately, several aspects of these new results don't increase confidence in the reliability or significance of many of the findings of the study.

One of the stronger conclusions in the original manuscript was that smoking was associated with differences in titers against rhinovirus peptides. The authors' experiment to further test this result did not reproduce it, and an explanation is offered that this is because testing of exactly the peptides that gave the original result in the VirScan data was not possible due to time constraints. The validation assay used viral lysates as the antigen source, and it is not obvious why this kind of assay wouldn't be expected to correlate with rhinovirus exposure, infection history and serological responses. If the original result was indeed restricted to a very few peptides, it is unclear what the immunological or clinical significance of this finding could be.

For the attempted confirmations of the VirScan results with ELISA or Luminex assays, the authors list in Suppl. Table 1 these influenza antigens used in the Luminex panel to assess for age-related titer changes:

- Chimeric protein comprising of A/White-fronted Goose/Netherlands/21/1999 (H6HA head) and A/Puerto Rico/8/1934 (H1HA stalk) (Krammer et al., J Virol 2013)
- Influenza A H3N2 (A/Beijing/32/1992) Hemagglutinin / HA1 Protein (His Tag)

For point (i), testing based on this limited number of antigens (and the additional influenza peptides listed in Suppl. Table 1) is not clearly interpretable or informative for assessing whether there is an age association with antibody titers binding to the head vs. stalk of influenza HA. The head vs. stalk antigens in the validation assays come from completely different influenza virus types (an H1 stalk from 1934 in the chimeric H6 head/H1 stalk protein, being compared to an H3 HA1 from 1992). There is extensive literature about imprinting effects in influenza antibody responses, related to the birth year and age of research participants, which can differ between HA types (particularly between group 1 viruses including H1 and H2 types, and group 2 viruses including H3 types), and which can result in individuals in particular age ranges differing in their antibody titers to particular H1 or H3 antigens, not necessarily related to differences in head vs. stalk binding titers. The experiments in the revised manuscript don't really resolve these complexities, and don't support general conclusions about age-related changes in head vs. stalk antigen domains. The sentence in the abstract “We demonstrate that age, sex, and continent of birth extensively affect not only the viruses but also the specific viral epitopes targeted by the antibody repertoire” does not seem well-supported by these data in relation to age effects on influenza epitope binding antibodies.

For point (ii), similar concerns apply to age effects on the influenza M1 antibody analysis, where the validation experiments for age effects include only a single peptide from a particular M1 sequence, and it is not clear how many M1 sequences from different historical viruses are represented in the original VirScan result.

In (iii), effects of sex on antibody titers to influenza HA versus M1, these results are difficult to interpret without knowledge of what fraction of males versus females were vaccinated, and how recently. Increased female antibody responses to influenza vaccination (primarily increasing HA titers) compared to males are well documented (summarized in Cook, Vaccine 2008, PMID 185244330). Decreased female severity of influenza infection is also reported, and may be causally related to higher HA titers prior to infection. Males could have higher antibody responses to M1 antigen due to having more severe infections. But the accuracy of measurement of M1 antibodies with the VirScan platform needs to be documented further, as older literature (summarized in Krammer, Nat Rev Immunol, 19, pages 383–397 (2019)) reports low rates of seropositivity of individuals for M1 antibodies, and low seroconversion after known infection. The revised manuscript doesn't add clarity or fully address these topics.

In (iv) the effects of continent of birth on herpesvirus titers are convincing and consistent with other literature, and the EBNA peptide results are consistent with the geographical distribution of the strains of virus from which they are derived, and are validated by the additional Luminex experiments.

The authors also evaluated the extent to which batch effects may have affected the results due to confounding with participant demographics.

For the MI cohort, they note:

“If biological effects are confounded by batch effects, we expect the two proportions to be correlated. However, we found weak correlations between the two statistics: antibody levels with the largest biological effects typically showed limited proportions of variance explained by batch effects, and vice versa (Pearson’s coefficient $r_{Age} = 0.042$, Supplementary Fig. 1c; $r_{Sex} = -0.0063$, Supplementary Fig. 1e). Notably, peptides largely explained by batch effects were enriched for peptides attributed to the viral family ‘Others’, which includes several uncommon viruses, raising questions about the biological relevance of these peptides (Supplementary Fig. c,e).”

For the EIP cohort, they note:

“We estimated the proportion of variance explained by batches in EUB only ($n = 212$ out of 312 EIP samples, distributed on 4 out of 5 batches), which should estimate solely batch effects. Reassuringly, the correlation between the variance explained by batches in the uncorrected EUB data and that explained by continent of birth in the full batch-corrected data was low ($r_{Continent} = 0.043$; Supplementary Fig. 1i).”

However, it is not clear in the revised analysis whether the magnitude of the batch effect correlations with biological effects are much smaller than the correlations between biological variables (such as age or sex) and the VirScan peptide antibody measurements. In Suppl Fig. 1c, the percentage of variation explained by batch effects is plotted on a different, compressed axis scale compared to the percentage of variation explained by age, making the age effects appear more substantial and making the batch effects appear diminished. An even greater scale mismatch is used in Suppl Fig. 1e for the sex vs. batch effect comparison.

The serological associations with continent of origin compared to batch effects show a more modest impact of batch effects, and the authors have plotted these on the same axis scale. Here there is less concern about artifacts dominating the results.

The other concerns raised in points 5-8 of the review have been addressed adequately in the revised manuscript, or clarify that key data such as vaccination timing are not available.

Reviewer #3

(Remarks to the Author)

Thanks for the careful revision of the manuscript.

Decision Letter:

24th Oct 2025

Dear Etienne,

Thank you for providing a rebuttal response to the comments of referee #2 on your Resource manuscript entitled, "Demographic and genetic factors shape the epitope specificity of the human antibody repertoire against viruses". We are very interested in the possibility of publishing your study in Nature Immunology, but would like you to first revise the manuscript as noted to experimental/analysis points i-iv, as noted in your response to these concerns, in the form of a revised manuscript before we make a final decision on publication.

We therefore invite you to revise your manuscript taking into account all reviewer and editor comments. Please highlight all changes in the manuscript text file in Microsoft Word format.

* If you have not done so already please begin to revise your manuscript so that it conforms to our Resource format instructions at <http://www.nature.com/ni/authors/index.html>. Refer also to any guidelines provided in this letter.

* Please include a revised version of any required reporting checklist. It will be available to referees to aid in their evaluation of the manuscript goes back for peer review. They are available here:

Reporting summary:

Please note, Extended Data figures and tables are online-only (appearing in the online PDF and full-text HTML version of the paper), peer-reviewed display items that provide essential background to the Article but are not included in the printed version of the paper due to space constraints or being of interest only to a few specialists. A maximum of ten Extended Data display items (figures and tables) is typically permitted. When re-submitting your manuscript, please ensure that any supplementary figures and tables that are more critical to the manuscript's conclusions are converted to Extended data to increase these data's visibility.

Link Redacted

We hope to receive your revised manuscript within two weeks. If you cannot send it within this time, please let us know. We will be happy to consider your revision so long as nothing similar has been accepted for publication at Nature Immunology or published elsewhere.

Nature Immunology is committed to improving transparency in authorship. As part of our efforts in this direction, we are now requesting that all authors identified as 'corresponding author' on published papers create and link their Open Researcher and Contributor Identifier (ORCID) with their account on the Manuscript Tracking System (MTS), prior to acceptance. ORCID helps the scientific community achieve unambiguous attribution of all scholarly contributions. You can create and link your ORCID from the home page of the MTS by clicking on 'Modify my Springer Nature account'. For more information please visit www.springernature.com/orcid.

Kind regards,

Laurie

Laurie A. Dempsey, Ph.D.
Senior Editor
Nature Immunology
l.dempsey@us.nature.com
ORCID: 0000-0002-3304-796X

Referee expertise:

Referee #1:

Referee #2:

Referee #3:

Reviewers' Comments:

Reviewer #1 (Remarks to the Author):

The authors have addressed all of my comments and have considerably strengthened the manuscript.

Reviewer #2 (Remarks to the Author):

In the revisions of this manuscript, the authors clarified the nature of their data sharing plans, and attempted to validate their main conclusions from the prior manuscript, using a Luminex assay with intact proteins. They also evaluated the impact of batch effects on their results.

The authors' description of their data sharing plans and the constraints imposed by regulatory requirements have alleviated my prior concerns.

Major concerns persist in the revised manuscript. In the response to reviews and description of new experiments attempting to validate the VirScan results using Luminex or ELISA assays, the authors state:

“Specifically, we confirmed that:

- (i) antibody titers against the less conserved globular head ($\beta_{\text{Luminex}} = -0.033$; $P_{\text{Luminex}} = 4.42 \times 10^{-76}$) and the more conserved stalk ($\beta_{\text{Luminex}} = 0.014$; $P_{\text{Luminex}} = 8.62 \times 10^{-15}$) domains of IAV hemagglutinin change in opposite directions with age;
- (ii) antibody titers against less conserved positions 200-250 of the IAV M1 protein decrease with age ($\beta_{\text{Luminex}} = -0.0068$, $P_{\text{Luminex}} = 4.04 \times 10^{-4}$);
- (iii) antibodies preferentially target the IAV HA protein in females ($\beta_{\text{Luminex}} = -0.10$; $P_{\text{Luminex}} = 0.038$), whereas antibodies in males target M1 ($\beta_{\text{Luminex}} = 0.092$; $P_{\text{Luminex}} = 0.02$) and NP 2 ($\beta_{\text{Luminex}} = 0.10$; $P_{\text{Luminex}} = 0.03$) proteins from IAV and IBV, respectively;
- (iv) antibodies of AFB preferentially target EBNA-4 peptides deriving from the AG876 strain ($\beta = -1.28$, $P = 9.12 \times 10^{-14}$) whereas antibodies of EUB target EBNA-6 peptides from the B95-8 strain ($\beta = 1.02$, $P = 8.2 \times 10^{-13}$).”

Unfortunately, several aspects of these new results don't increase confidence in the reliability or significance of many of the findings of the study.

One of the stronger conclusions in the original manuscript was that smoking was associated with differences in titers against rhinovirus peptides. The authors' experiment to further test this result did not reproduce it, and an explanation is offered that this is because testing of exactly the peptides that gave the original result in the VirScan data was not possible due to time constraints. The validation assay used viral lysates as the antigen source, and it is not obvious why this kind of assay wouldn't be expected to correlate with rhinovirus exposure, infection history and serological responses. If the original result was indeed restricted to a very few peptides, it is unclear what the immunological or clinical significance of this finding could be.

For the attempted confirmations of the VirScan results with ELISA or Luminex assays, the authors list in Suppl. Table 1 these influenza antigens used in the Luminex panel to assess for age-related titer changes:

- Chimeric protein comprising of A/White-fronted Goose/Netherlands/21/1999 (H6HA head) and A/Puerto Rico/8/1934 (H1HA stalk) (Krammer et al., J Virol 2013)
- Influenza A H3N2 (A/Beijing/32/1992) Hemagglutinin / HA1 Protein (His Tag)

For point (i), testing based on this limited number of antigens (and the additional influenza peptides listed in Suppl. Table 1) is not clearly interpretable or informative for assessing whether there is an age association with antibody titers binding to the head vs. stalk of influenza HA. The head vs. stalk antigens in the validation assays come from completely different influenza virus types (an H1 stalk from 1934 in the chimeric H6 head/H1 stalk protein, being compared to an H3 HA1 from 1992). There is extensive literature about imprinting effects in influenza antibody responses, related to the birth year and age of research participants, which can differ between HA types (particularly between group 1 viruses including H1 and H2 types, and group 2 viruses including H3 types), and which can result in individuals in particular age ranges differing in their antibody titers to particular H1 or H3 antigens, not necessarily related to differences in head vs. stalk binding titers. The experiments in the revised manuscript don't really resolve these complexities, and don't support general conclusions about age-related changes in head vs. stalk antigen domains. The sentence in the abstract “We demonstrate that age, sex, and continent of birth extensively affect not only the viruses but also the specific viral epitopes targeted by the antibody repertoire” does not seem well-supported by these data in relation to age effects on influenza epitope binding antibodies.

For point (ii), similar concerns apply to age effects on the influenza M1 antibody analysis, where the validation experiments for age effects include only a single peptide from a particular M1 sequence, and it is not clear how many M1 sequences from different historical viruses are represented in the original VirScan result.

In (iii), effects of sex on antibody titers to influenza HA versus M1, these results are difficult to interpret without knowledge of what fraction of males versus females were vaccinated, and how recently. Increased female antibody responses to influenza vaccination (primarily increasing HA titers) compared to males are well documented (summarized in Cook, Vaccine 2008, PMID 185244330). Decreased female severity of influenza infection is also reported, and may be causally related to higher HA titers prior to infection. Males could have higher antibody responses to M1 antigen due to having more severe infections. But the accuracy of measurement of M1 antibodies with the VirScan platform needs to be documented further, as older literature (summarized in Krammer, Nat Rev Immunol, 19, pages 383–397 (2019)) reports low rates of seropositivity of individuals for M1 antibodies, and low seroconversion after known infection. The revised manuscript doesn't add clarity or fully address these topics.

In (iv) the effects of continent of birth on herpesvirus titers are convincing and consistent with other literature, and the EBNA

peptide results are consistent with the geographical distribution of the strains of virus from which they are derived, and are validated by the additional Luminex experiments.

The authors also evaluated the extent to which batch effects may have affected the results due to confounding with participant demographics.

For the MI cohort, they note:

“If biological effects are confounded by batch effects, we expect the two proportions to be correlated. However, we found weak correlations between the two statistics: antibody levels with the largest biological effects typically showed limited proportions of variance explained by batch effects, and vice versa (Pearson’s coefficient $r_{Age} = 0.042$, Supplementary Fig. 1c; $r_{Sex} = -0.0063$, Supplementary Fig. 1e). Notably, peptides largely explained by batch effects were enriched for peptides attributed to the viral family ‘Others’, which includes several uncommon viruses, raising questions about the biological relevance of these peptides (Supplementary Fig. c,e).”

For the EIP cohort, they note:

“We estimated the proportion of variance explained by batches in EUB only ($n = 212$ out of 312 EIP samples, distributed on 4 out of 5 batches), which should estimate solely batch effects. Reassuringly, the correlation between the variance explained by batches in the uncorrected EUB data and that explained by continent of birth in the full batch-corrected data was low ($r_{Continent} = 0.043$; Supplementary Fig. 1i).”

However, it is not clear in the revised analysis whether the magnitude of the batch effect correlations with biological effects are much smaller than the correlations between biological variables (such as age or sex) and the VirScan peptide antibody measurements. In Suppl Fig. 1c, the percentage of variation explained by batch effects is plotted on a different, compressed axis scale compared to the percentage of variation explained by age, making the age effects appear more substantial and making the batch effects appear diminished. An even greater scale mismatch is used in Suppl Fig. 1e for the sex vs. batch effect comparison.

The serological associations with continent of origin compared to batch effects show a more modest impact of batch effects, and the authors have plotted these on the same axis scale. Here there is less concern about artifacts dominating the results.

The other concerns raised in points 5-8 of the review have been addressed adequately in the revised manuscript, or clarify that key data such as vaccination timing are not available.

Reviewer #3 (Remarks to the Author):

Thanks for the careful revision of the manuscript.

Version 2:

Decision Letter:

Our ref: NI-RS39162B

28th Nov 2025

Dear Etienne & Lluis,

Thank you for submitting your revised manuscript "Demographic and genetic factors shape the epitope specificity of the human antibody repertoire against viruses" (NI-RS39162B). We looked at your revisions and acknowledge that these have addressed the final concerns voiced by the referees, thus we'll be happy in principle to publish it in Nature Immunology, pending minor revisions to comply with our editorial and formatting guidelines.

We will now perform detailed checks on your paper and will send you a checklist detailing our editorial and formatting requirements in about a week. Please do not upload the final materials and make any revisions until you receive this additional information from us.

If you had not uploaded a Word file for the current version of the manuscript, we will need one before beginning the editing process; please email that to immunology@us.nature.com at your earliest convenience.

Thank you again for your interest in Nature Immunology Please do not hesitate to contact me if you have any questions.

Kind regards,

Laurie

Laurie A. Dempsey, Ph.D.
Senior Editor
Nature Immunology
l.dempsey@us.nature.com
ORCID: 0000-0002-3304-796X

Reviewer #1

The study by Olin et al applied VirScan to 2 different cohorts of 1212 adults to understand how intrinsic and external factors influence antiviral antibody specificity. They demonstrate virus and sometimes epitope-specific impacts of age, sex, continent of birth as well as smoking status. They further link the VirScan data to GWAS analyses and identify associations between HLA, IGH, IGK and FUT2 variants with specific antibody measurements. While a lot of the findings are confirmatory of past associations (as appropriately noted by the authors), the study also makes several novel contributions to the field. The dataset is likely to be a very useful resource for future analyses across the field of immunology. The results are already available on a web-based browser and the data/code available through relevant repositories.

We thank the reviewer for their critical review of our manuscript and for their insightful comments. We also appreciate their positive comment regarding the potential of our dataset to serve as a valuable resource for future immunological analyses.

While the authors clearly acknowledge the limitations of VirScan (e.g linear vs structural epitopes, binding vs neutralising antibodies) and have compared VirScan with 'conventional' serology approaches, I think it is pertinent to validate some of the key observations using ELISA (the data may already be part of the ELISA work performed but it should be presented clearly as a separate supplementary figure). Specifically, it would be important to show by ELISA that men have higher antibodies to flu NP than women, and women have higher antibodies to HA. Similarly, the differences in ENBA4 and EBNA6 between different continents of birth, and the Rhinovirus B difference between smoker/non-smokers. Most of these proteins are commercially available so this should be feasible. The same can be done with peptide-specific differences between other comparisons using peptides in ELISAs. Given that some of these findings have the potential to initiate future work to investigate their mechanistic bases, I think it is critical to validate them at this point.

We thank the reviewer for raising this important point. As requested by Reviewer 1 (as well as Reviewer 2), we have now validated our key PhIP-seq-based findings using multiplex immunoassays (Luminex®). Specifically, we have assessed (i) age differences in antibody reactivity to HA and M1 IAV epitopes, (ii) sex differences in reactivity to HA, NP and M1 epitopes of IAV and IBV, (iii) population differences in reactivity to EBNA-4 and EBNA-6 epitopes of EBV between Belgians born in Africa (AFB) or Europe (EUB), and (iv) differences in reactivity to rhinovirus B between smokers and never-/ex-smokers. As noted by the reviewer, several of these associations are peptide-specific and could not be assessed using commercially available proteins. Therefore, we employed either commercial antigens (HA1 domain for IAV; HA and NP proteins for IBV; rhinovirus B lysate) or newly synthesized peptides (specific HA stalk and M1 peptides from IAV; lineage-specific EBV peptides). Details about the peptides or proteins tested can be found **in the new Table S1**.

We quantified antibody titers against the 10 candidate antigens by multiplexed Luminex® assays on all available MI ($n = 1,000$) and EIP ($n = 410$) samples. We were able to replicate most of our findings. Specifically, we confirmed that: (i) antibody titers against the less conserved globular head ($\beta_{\text{Luminex}} = -0.033$; $P_{\text{Luminex}} = 4.42 \times 10^{-76}$) and the more conserved stalk ($\beta_{\text{Luminex}} = 0.014$; $P_{\text{Luminex}} = 8.62 \times 10^{-15}$) domains of IAV hemagglutinin change in opposite directions with age; (ii) antibody titers against less conserved positions 200-250 of the IAV M1 protein decrease with age ($\beta_{\text{Luminex}} = -0.0068$, $P_{\text{Luminex}} = 4.04 \times 10^{-4}$); (iii) antibodies preferentially target the IAV HA protein in females ($\beta_{\text{Luminex}} = -0.10$; $P_{\text{Luminex}} = 0.038$), whereas antibodies in males target M1 ($\beta_{\text{Luminex}} = 0.092$; $P_{\text{Luminex}} = 0.02$) and NP

($\beta_{\text{Luminex}} = 0.10$; $P_{\text{Luminex}} = 0.03$) proteins from IAV and IBV, respectively; (iv) antibodies of AFB preferentially target EBNA-4 peptides deriving from the AG876 strain ($\beta = -1.28$, $P = 9.12 \times 10^{-14}$) whereas antibodies of EUB target EBNA-6 peptides from the B95-8 strain ($\beta = 1.02$, $P = 8.2 \times 10^{-13}$). However, we were not able to confirm by Luminex that immunoreactivity to rhinoviruses is strongly associated with smoking, likely because we used, for time reasons, a commercial assay based on a whole-virus lysate of rhinovirus, instead of synthesizing the specific peptides identified by our VirScan analyses. We have thus toned down our claims regarding associations with smoking, by removing the sentence describing these results in the abstract (l. 41-42), and by stating that these results were not validated by Luminex in the Results (l. 252-253) and the Discussion (l. 389-391).

The new validation results are now reported in the main text (l. 167-169, 171-173, 192-196, 229-231) and in Figs. 2h, 3d,h, and two new sections in the Methods have been added (l. 830-872).

Other minor points:

- It would be beneficial to also note that the effects of ‘continent of birth’ may also be related to genetics? For example, HLA genes are differentially represented in different populations, so the effects associated with continent of birth may not necessarily reflect differences in viral exposure.

Following the reviewer’s recommendation, we have now tested whether the observed continent-of-birth effects on antibody reactivity could be explained, at least in part, by genetics. Specifically, we have adjusted the statistical models assessing continent-of-birth effects for the 17 lead GWAS variants identified in our study (i.e., *HLA*, *IGH*, *IGK* and *FUT2* loci; **new Table S7**). After adjustment, the marked differences in antibody reactivity between African-born and European-born individuals remain largely unchanged (see **new Extended Data Fig. 4d**). These findings suggest that the strong population differences in antibody reactivity initially observed are more likely driven by differential viral exposure rather than genetic variation. **We report these new analyses in the revised main text (l. 217-220)**. We thank the reviewer for this constructive suggestion, which has helped to strengthen our conclusions.

- In the methods section for ELISA/Luminex (lines 729 and 742 onwards) please provide more details in the source of antigens used.

We have now provided the requested details **in the new Table S1**.

- There is a methods section for a KREC assay – but I was unable to locate the relevant data/results in the manuscript. Please clarify.

In the initial version of the manuscript, we tested the association between KREC levels and VirScan Z-scores but did not observe any significant associations. As these results were not explicitly mentioned, **we now report them in the revised version of the Results (l. 131-136) and in a new section of the Supplementary Note, along with the new Supplementary Fig. 4**. In the Supplementary Note, we systematically describe associations between the PhIP-seq data and various immune cell phenotypes measured in the Milieu Intérieur cohort. These analyses show that inter-individual variation in B-cell phenotypes does not correlate with humoral responses against specific viruses.

- Please clarify which structures were used in figure 2 (e.g. PDBxxx)

The requested PDB identifier has now been added **to the revised version of Fig. 2 legend (l. 512).**

- In figure 1 c and f – should the y-axis be ‘proportion of donors’ instead of ‘peptides’?

The reviewer is absolutely right, thanks for pointing this out. **We have corrected the new version of Fig. 1 accordingly.**

Reviewer #2

This manuscript reports a large-scale analysis of plasma IgG binding to a library of 97,000 phage-displayed peptides from viral families of clinical significance (VirScan V3), using 900 specimens from members of a European-ancestry cohort (MI) with ages 20-69 years, and 312 specimens from individuals of a cohort of Belgian residents of either Central African or European birth (EIP). The results evaluate correlations between the concentration of plasma antibodies binding peptides from the phage displayed panel and demographic, lifestyle and genetic data from the participants. The study reports correlations between plasma antibody binding of viral peptides and participant age, sex, and continent of birth, with strong associations between smoking status and plasma antibodies against peptides from viruses such as rhinoviruses. A very interesting part of the manuscript is its GWAS analysis of genotype associations (including germline sequence variants within the immunoglobulin heavy chain and light chain loci, which are rarely examined) with antibody binding to viral peptides, showing strongest linkage between antibody binding to viral peptides and genetic variants in the MHC, IGH, IGK and FUT2 loci.

The study reconfirms some previously reported effects of sex, age and population groups on antibody titers against certain viruses. It expands upon previously-reported genetic associations with antibody titers against particular viruses and the MHC, IGH and FUT2 loci.

Strengths of the study are its breadth of coverage of virus types with the peptide antigens, the relatively large clinical cohorts with detailed demographic and lifestyle data, and the pairing of broad serological testing with genome-wide genetic association testing. The associations detected between participant sex or age and the proteins and domains targeted by antibodies are novel, interesting and potentially clinically important. The experimental and data analysis approaches in the study appear well thought-out and executed, to the extent to which those can be assessed based on the text and figures of the manuscript and supplementary documents.

Major limitations of the study are the unclear commitment of the authors to truly share the data from the manuscript if it were to be published; the uncertain importance (clinically or otherwise) of antibody titers against isolated linear peptide antigens; the lack of functionally important data such as viral neutralization titers or other antibody functions from the PhIPSeq assay; and the limited degree to which new potential findings from the PhIPSeq assay are validated and replicated with immunoassays using intact viral antigens or antigen domains when those are implicated by the peptide binding assay. The previously-reported AVARDA algorithm used to attempt to correct for possible cross-reactivity between peptides in the library only partly addresses concerns about analytical accuracy of the assay, as the AVARDA approach seems to have been used previously to improve seropositivity status assessment for individual viruses with PhIPSeq, but not to enable quantitative comparison of antibody titers against particular structurally-intact protein domains within a viral antigen, or between different viral antigens.

We thank the reviewer for their critical review of our manuscript, as well as for their positive comments on the strengths of our study and for highlighting key points that required clarification.

Major Concerns

1. The greatest concern with the manuscript is whether there will actually be open data sharing to allow other researchers to evaluate the data, replicate the analyses and independently assess the conclusions of the study. With complicated data types and analyses such as those presented in the study, full data sharing is critically important if the data,

results and conclusions are ever going to be assessed by anyone other than the authors. The data sharing proposed is inadequate. Browsing on the “mirepertore” graphical display system does not substitute for making the underlying data available to other scientists.

- The OWEY data sharing portal at the Institut Pasteur does not appear to be independent of the researchers who conducted this work, and the data sharing link gives this message at the site: “Please fill in the following document*

https://redcap.pasteur.fr/surveys/?s=ND8TP8MDD3 in order to request access to the desired data. Milieu Intérieur Consortium members will discuss your request during the next Data Access Committee and send you back an answer (positive or negative) or request for complementary information.”

- The redcap form outlines many restrictions on data access and use, requires that other researchers comply with a “publication charter” that is not cited or linked on the document.*

- The redcap form does not list replication or evaluation of the present manuscript analysis as one of the valid reasons for requesting the study data*

- The available dataset appears to be only for the Milieu Intérieur participants and not for the EvoImmunoPop participants in the study.*

- Under what conditions would the authors not share the full and complete data from this study to enable other researchers to attempt to replicate and evaluate the analyses and conclusions? There would be very little risk of re-identification of participants or other potential harms based on serology results, basic demographics and lifestyle information.*

- The data available at the site is listed as already processed z-scores or binary positive/negative results for the peptides in the PhIP-Seq analysis, rather than the actual sequence data that would enable other researchers to attempt to replicate the analysis results. The manuscript states that “raw and processed data” are available at the sharing site, which is not consistent with the description at the OWEY site.*

- This review represents a request to the Milieu Intérieur Consortium members to share the PhIP-Seq data and participant metadata for the purpose of allowing other scientists to re-analyze the data to attempt to replicate the results and conclusions in this manuscript. The data should be deposited in an archive outside of author control, like ImmPORT or SRA, with no or minimal restrictions on access.*

We fully agree with the reviewer that reproducibility and open science must be central priorities in research. For this reason, the Milieu Intérieur consortium has long been committed to facilitating direct access to the datasets generated in the Milieu Intérieur cohort. Since 2013, our data access committee (DAC) has accepted all 70 data access requests submitted to date – listed on our data sharing webpage on the Milieu Intérieur website. We would also like to clarify that the “mirepertore” web interface was never intended as a substitute for full data sharing. It was developed solely to facilitate the exploration and download of our statistical results.

However, we are legally unable to share the data without conditions – as requested by the reviewer. This restriction stems from French and international laws governing the use of personal data from research participants. In compliance with these regulations, we are required:

- (1) To continuously inform research participants on every project using their personal data, so they retain the right to withdraw at any time. To this end, we are required to publish on our dedicated website (<https://www.milieuinterieur.fr/en/>) the title, non-technical summary, and names and affiliations of researcher for each approved project.

(2) To ensure that secondary use of the data is conducted in full respect of the conditions stipulated in participants' informed consent. Specifically, the applicant is not authorized to (i) use the data for diagnostic purposes, (ii) store the data outside of secure servers, (iii) reidentify research participants, and (iv) share the data with a third party. Furthermore, research on Milieu Intérieur and EvoImmunoPop datasets is restricted to research on the genetic and environmental determinants of immune variation. Access conditions for Milieu Intérieur data and EvoImmunoPop are specified in our online data request forms, at <https://redcap.pasteur.fr/surveys/?s=ND8TP8MDD3> (Milieu Intérieur) and <https://redcap.pasteur.fr/surveys/?s=F3AA7J4M4W8LRNJ4> (EvoImmunoPop).

In line with these legal requirements, data from both Milieu Intérieur and EvoImmunoPop are hosted on OWEY, the secure institutional repository developed by Institut Pasteur. Access is open, contingent on submission of a short data access request to the corresponding Data Access Committee (DAC). Importantly, the DAC includes external members to ensure impartial review. As noted above, all requests received thus far have been approved by Milieu Intérieur and EvoImmunoPop DACs.

Besides these legal considerations, we thank the reviewer for pointing out that the language referring to a “publication charter” in our online form was unclear. In no way does the Milieu Intérieur consortium request co-authorship for secondary use of its data, including studies aiming to replicate or re-evaluate published findings. **This text has now been clarified accordingly** (<https://redcap.pasteur.fr/surveys/?s=ND8TP8MDD3>). Furthermore, we have updated the access request form to clarify that the Milieu Intérieur data can be accessed for the purpose of replicating published results.

Lastly, we wish to clarify that both raw (i.e., fastq files) and processed (i.e., full matrix of Z-scores) VirScan data for Milieu Intérieur and EvoImmunoPop were deposited on OWEY, although the associated landing pages did not clearly reflect it. **These pages have now been updated for clarity** (<https://doi.org/10.48802/owey.84rn-jg72>). Furthermore, we have revised the **Data availability section** of the main text accordingly (l. 1095-1102).

We thank the reviewer for their critical evaluation of our data sharing framework and for highlighting areas that required clarification. We hope these explanations and corresponding revisions demonstrate our strong and ongoing commitment to making these datasets as openly accessible as legal and ethical regulations permit.

2. Major and more novel results (male/female differences in influenza antigen binding to HA vs. NP or MPI, and HA head vs. stalk domain binding; rhinovirus antibody binding in smokers compared to non-smokers) should be validated using independent immunoassays with intact antigen proteins or protein domains.

We thank the reviewer for raising this important point. As requested by Reviewer 2 (as well as Reviewer 1), we have now validated our key PhIP-seq-based findings using multiplex immunoassays (Luminex®). Specifically, we have assessed (i) age differences in antibody reactivity to HA and M1 IAV epitopes, (ii) sex differences in reactivity to HA, NP and M1 epitopes of IAV and IBV, (iii) population differences in reactivity to EBNA-4 and EBNA-6 epitopes of EBV between Belgians born in Africa (AFB) or Europe (EUB), and (iv) differences in reactivity to rhinovirus B between smokers and never-/ex-smokers. As noted by the reviewer, several of these associations are peptide-specific and could not be assessed using commercially available proteins. Therefore, we employed either commercial antigens (HA1

domain for IAV; HA and NP proteins for IBV; rhinovirus B lysate) or newly synthesized peptides (specific HA stalk and M1 peptides from IAV; lineage-specific EBV peptides). Details about the peptides or proteins tested can be found **in the new Table S1**.

We quantified antibody titers against the 10 candidate antigens by multiplexed Luminex® assays on all available MI ($n = 1,000$) and EIP ($n = 410$) samples. We were able to replicate most of our findings. Specifically, we confirmed that: (i) antibody titers against the less conserved globular head ($\beta_{\text{Luminex}} = -0.033$; $P_{\text{Luminex}} = 4.42 \times 10^{-76}$) and the more conserved stalk ($\beta_{\text{Luminex}} = 0.014$; $P_{\text{Luminex}} = 8.62 \times 10^{-15}$) domains of IAV hemagglutinin change in opposite directions with age; (ii) antibody titers against less conserved positions 200-250 of the IAV M1 protein decrease with age ($\beta_{\text{Luminex}} = -0.0068$, $P_{\text{Luminex}} = 4.04 \times 10^{-4}$); (iii) antibodies preferentially target the IAV HA protein in females ($\beta_{\text{Luminex}} = -0.10$; $P_{\text{Luminex}} = 0.038$), whereas antibodies in males target M1 ($\beta_{\text{Luminex}} = 0.092$; $P_{\text{Luminex}} = 0.02$) and NP ($\beta_{\text{Luminex}} = 0.10$; $P_{\text{Luminex}} = 0.03$) proteins from IAV and IBV, respectively; (iv) antibodies of AFB preferentially target EBNA-4 peptides deriving from the AG876 strain ($\beta = -1.28$, $P = 9.12 \times 10^{-14}$) whereas antibodies of EUB target EBNA-6 peptides from the B95-8 strain ($\beta = 1.02$, $P = 8.2 \times 10^{-13}$). However, we were not able to confirm by Luminex that immunoreactivity to rhinoviruses is strongly associated with smoking, likely because we used, for time reasons, a commercial assay based on a whole-virus lysate of rhinovirus, instead of synthesizing the specific peptides identified by our VirScan analyses. We have thus toned down our claims regarding associations with smoking, by removing the sentence describing these results in the abstract (l. 41-42), and by stating that these results were not validated by Luminex in the Results (l. 252-253) and the Discussion (l. 389-391).

The new validation results are now reported in the main text (l. 167-169, 171-173, 192-196, 229-231) and in Figs. 2h, 3d,h, and two new sections in the Methods have been added (l. 830-872).

3. For the major and more novel results (male/female differences in influenza antigen binding to HA vs. NP or MP1, and HA head vs. stalk domain binding; rhinovirus antibody binding in smokers compared to non-smokers), the magnitude of the differences in antibody quantities measured should be reported, rather than only the P-values. What are the effect sizes of these differences? Presumably, they could be tiny differences yet still have very low P-values given the large data set size.

Following the reviewer's request, **we have now reported effect sizes throughout the revised Results section**. The magnitude of these effects can also be appreciated in the updated versions of Figs. 2 and 3, which show both PhIP-seq- and Luminex-based measurements of antibody reactivity. In addition, all effect sizes are available for exploration and download via our web interface: <https://mirepertoire.pasteur.cloud/>.

We would like to highlight that, for more than 20% of viral peptides, the factors discovered in this study collectively explain more than 10% of variance in antibody reactivity (Fig. 6a). Nevertheless, we agree that, overall, the variance explained by the factors identified remains limited. **We now acknowledge this limitation in the revised manuscript (l. 421-425)**, emphasizing the need for future studies to explore additional sources of inter-individual variation in antibody levels.

4. Correction for batch effects in the experiment are mentioned. How large were any batch effects between sample batches tested, and do the sample batches have any relationship to age, sex or population group?

We thank the reviewer for this important comment. To address this concern, we now describe the magnitude of batch effects in our data and assess their relationship with age, sex and continent-of-birth effects, in a **new section of the Supplementary Note and the new Supplementary Fig. 1**, which are now cited in the main text (l. 100-101). We first show that batch effects are indeed substantial in the PhIP-seq data before batch correction: samples clustered by sequencing batch and/or processing plate on the first two principal components from a PCA of the uncorrected data (**new Supplementary Fig. 1a,f**). However, after batch correction with ComBat, PC1 and PC2 are instead associated with biological factors—age in the MI cohort and continent of birth in the EIP cohort (**new Supplementary Fig. 1b,g**)—demonstrating that batch correction was effective and preserved genuine biological signals.

Given that sample allocation across processing plates was not perfectly randomized with respect to age, sex, and continent of birth, we next evaluated whether these biological effects can be confounded by technical batch effects. For each peptide Z-score, we compared the proportion of variance explained by batches before batch correction, with that explained by age, sex, or continent of birth after batch correction. Reassuringly, these analyses showed that, the proportion of variance explained by age or sex in the MI data, or continent of birth in the EIP data, is not correlated with that explained by batches, as shown in **new Supplementary Fig. 1c,e,i**. In other words, peptide Z-scores with strong effects of age, sex, or continent of birth typically showed limited batch effects. We also observed that peptides with strong batch effects were enriched for the viral family ‘Others’, which includes uncommon viruses not discussed in the manuscript.

These new analyses, together with the replication of strong age, sex and continent-of-birth effects by Luminex immunoassays (see Results section), indicate that the **major biological effects identified in our study (age, sex and continent of birth) are largely independent of batch effects**, supporting the view that our findings are not confounded by technical artifacts.

5. Are age-related serology associations independent of the population groups EUB vs. AFB that differ in their age distributions? Similarly, were the correlations between antibody levels and sex independent of age, or is there any interaction between age and sex? Or smoking and sex or age (e.g., more smoking in males?)

All the associations reported in this study were tested with multiple regression models that include age as covariate, except when the effect of age was tested. Therefore, the associations identified for sex, continent of birth, and smoking are independent of age effects. **This is now more explicitly explained in the revised Methods (l. 997-1021)**. Moreover, we have now further clarified in the revised manuscript that **we observed no significant age × sex interactions (l. 138-140)**. We have also evaluated whether peptide Z-scores are affected by interactions between sex and smoking, or between age and smoking, but found no significant interactions. We note, however, that we did detect interactions between age and the continent-of-birth in the EIP cohort: anti-RSV antibodies of EUB change with age only in EUB, likely because AFB have not been recurrently exposed to RSV infections. **These findings are now shown in the main text (l. 342-343) and the new Extended Data Fig. 6e**. We thank the reviewer for these suggestions, which led to an improvement of the manuscript.

*6. In the influenza analyses (Page 6) “These differences were not driven by age-related variations in exposure to different IAV subtypes, as both positive and negative associations were observed within the same IAV subtypes for HA and MPI”
- Could this be due to different imprinting in old and young for peptides that are conserved or vary across older flu strains or more recent ones (e.g., older participants giving relatively*

stronger responses to peptides that are more conserved between older and more recent strains?)

- The authors should test whether the degree of sequence conservation for these peptide regions in flu variants from prior decades is correlated with the age-related differences seen in the serology data

- Would the AVARDA algorithm be able to account for prior flu strains participants might have been infected by in prior decades?

We fully agree with the reviewer that the opposite age associations observed along the IAV HA and M1 proteins are consistent with a scenario in which age-related changes in anti-IAV antibody levels depend on the sequence conservation of targeted viral epitopes. Specifically, it is plausible that antibodies targeting the most conserved IAV epitopes tend to increase with age, as individuals are recurrently exposed to similar antigens (i.e., immune recall), whereas antibodies directed against less conserved domains may decline with age (i.e., waning), due to their reduced ability to recognize rapidly-evolving IAV antigens.

To test this hypothesis, we have now collaborated with Jack Crook and Etienne Simon-Lorière (Evolutionary Genomics of RNA Viruses lab, Institut Pasteur), new co-authors of this study, to estimate per-residue HA and M1 evolutionary rates using all strains circulating between 1975 and 2012 and deposited on the GISAID database. As expected, these new analyses show that the HA globular head is less conserved than the stalk domain, **as now presented in new Fig. 2d**.

More importantly, the antibodies whose reactivity decreases most with age tend to target less conserved HA peptides ($\rho = 0.53$, $P = 7.4 \times 10^{-5}$; **see new Fig. 2e**) and M1 peptides (Spearman's $\rho = -0.70$, $P = 1.8 \times 10^{-4}$), supporting the reviewer's hypothesis. **We now report these results and discuss their implications in the revised manuscript (l. 39-41, 165-167, 170-175, 366-367)**. We warmly thank the reviewer for suggesting these very relevant analyses, which substantially strengthen our manuscript.

Finally, regarding the AVARDA algorithm: we do not expect it to account for prior flu strains, as the iterative peptide removal implemented in AVARDA relies solely on sequence identity among peptides and does not incorporate information about strain circulating year.

- Does greater flu HA vs. M1 binding in women compared to men reflect similar response to infection between men and women, but better vaccine responses in women or more frequent vaccination of women (which would probably preferentially increase the HA responses)? The influenza vaccination rates are described as not differing between women and men, but does this take into account how often and how recently women and men were vaccinated relative to the blood sampling for this study?

Unfortunately, we did not collect data on the timing of the last flu vaccination or the number of flu vaccines received. **We now acknowledge this limitation in the revised version of the manuscript (l. 375-378)**.

- How do the flu antibody levels and proteins or domains targeted differ between men and women that have been vaccinated recently, compared to between men and women who reported no vaccination?

As suggested by the reviewer, we have now tested whether self-declared flu vaccination status is associated with PhIP-seq Z-scores in the MI cohort. Antibody reactivity differed significantly between vaccinated and unvaccinated participants for only three peptides, all derived from IBV ($P_{\text{adj}} < 0.05$). This contrasts with the strong association observed between

self-declared vaccination status and total anti-IAV antibody titers measured by ELISA in the same individuals ($P = 2.62 \times 10^{-14}$; see l. 180). These results suggest that vaccination status was reliably reported by MI donors, but IAV and IBV linear peptides included in the VirScan library may not adequately capture the epitopes typically targeted by commercial flu vaccines. As the anti-viral antibody repertoires measured by VirScan appear to be more reflective of natural infection than vaccination, we found these results difficult to integrate in the manuscript and considered them beyond the scope of our study.

7. It is surprising that no viral serology results were correlated with genetic variants in the immunoglobulin lambda locus, whereas kappa locus variants were highly significant. Was there any difference in the extent or interpretability of genotyping in the lambda locus compared to kappa?

We agree with the reviewer that the absence of association with the *IGL* locus is unexpected, especially given the associations detected at the *IGK* locus. To explore this further, we have analyzed whole genome sequencing (WGS) data for the two loci, obtained for 936 Milieu Intérieur donors. We have now confirmed that the WGS data provides comparable coverage of the *IGK* and *IGL* loci. We have found that the genetic variant density is actually higher for *IGL* (8,030 SNPs in 2,884,611 bp, corresponding to 2.78 SNPs / kb) than for *IGK* (4,074 SNPs in 3,117,361 bp, corresponding to 1.31 SNPs / kb), suggesting that the absence of association at *IGL* is therefore genuine. We also note that the genetic associations we identified at the *IGK* locus were restricted to a single protein of adenovirus B.

8. For RSV results, what is known about the distribution of strain A in earlier decades? Is it possible that older people would have had greater exposure to it?

We have searched the literature to assess whether the French and Belgian populations have experienced differential exposure to RSV-A or RSV-B over the past five decades. We have found two longitudinal sero-epidemiological studies of RSV subtypes, one conducted in Western France from 1982 to 1990 (Freymuth et al., *J Clin Microbiol* 1991) and another in Belgium from 1996 to 2006 (Zlateva et al., *J Clin Microbiol* 2007). Both studies showed that the dominant RSV subtype alternated every two to three years, suggesting that all MI and EIP EUB participants have likely been exposed to both subtypes during childhood and/or adulthood. Therefore, temporal differences in subtype exposure are unlikely to account for the observed age-related differences in antibody reactivity to RSV-A and RSV-B. Furthermore, we would like to emphasize that such temporal variation cannot explain the strain-specific effects of *IGH* genetic variants on anti-RSV antibody responses. **We have now reported these observations in the main text (l. 344-346).**

Reviewer #3

This is a carefully executed work using state of the art analytics. The analysis uses 97000 peptide serum reactivity from more than 1000 individuals using VirScan, genomics and statistics. Below are some comments that may help in revising the current draft.

We thank the reviewer for their critical review of our manuscript and their insightful comments. We also appreciate their positive appraisal of the care we have taken in analyzing the data.

1. Population size-effect. The various associations are statistically robust, but as expected for the type of analytics and the presumed complex factors determining viral reactivity, the explained variance is modest (~3.5% for non-genetic and ~3.5% for genetic factors). This should be discussed to make sure that the readership is not misled.

We fully agree with the reviewer: it is important to clarify that the overall proportion of variance explained by the factors identified is modest. **We now explicitly discuss this limitation in the revised manuscript** and note that further studies will be required to explore the sources of unexplained inter-individual variation in antibody levels (**I. 421-425**). We would like to point out, however, that the reported 3.5% reflects an average across all peptides, and that the variance explained by our models exceeds 10% for over 20% of peptides (Fig. 6a). Furthermore, our analyses uncover how these modest effects can differentially affect antibody responses to specific epitopes within the same virus or viral protein, potentially paving the way for future mechanistic studies of antibody generation and maintenance.

2. Coverage of viral peptidome. From Extended data Fig 1, it is unclear how the difference in representation of viruses in VirScan (whether due to viral genome size or representativity/coverage) impacts the association statistics and conclusion. This should be addressed.

The reviewer is right to point out that the unbalanced representation of different viruses in the VirScan library could affect our conclusions. In particular, viruses represented by a large number of peptides, such as IAV and CMV, may be more likely to yield significant associations than those with fewer peptides. To address this, we used the AVARDA algorithm, which estimates viral-level serostatus by iteratively discarding redundant peptides, while accounting for the unequal representation of individual viruses in the library (Monaco et al., eBioMedicine 2022). For this reason, we systematically report both AVARDA and individual Z-score results (see detailed response to point 3 below). **This clarification has now been added to the revised version of the main text (I. 114-117).**

3. VirScan/Avarda. When looking at the implementation in the literature ([https://www.thelancet.com/journals/ebiom/article/PIIS2352-3964\(21\)00541-7/fulltext](https://www.thelancet.com/journals/ebiom/article/PIIS2352-3964(21)00541-7/fulltext)), it is unclear how best to present Avarda. One could assume that there would be no need to use native VirScan data and rather resort to the Avarda deconvoluted results. The present work uses both outputs. The Avarda breath score needs more explanation.

As suggested by the reviewer, we now provide a more detailed explanation of why we did not rely solely on AVARDA results. **This has been included in the revised version of the Results (I. 127-131) and Methods section (I. 1022-1033).**

A central aim of this study is to characterize inter-individual variation in antibody reactivity against different peptides from the same virus—an analysis that is not possible when using virus-level AVARDA scores only. We thus chose to analyze both peptide Z-scores and AVARDA breadth scores, to leverage the high resolution of the VirScan peptide library while accounting for potential between-species cross-reactivity and the unbalanced representation of different viruses.

More specifically, for each candidate factor (such as age), we tested associations with all public peptide Z-scores and the AVARDA breadth scores for the corresponding viruses. We considered three scenarios: (i) Z-scores for several peptides from a given virus and the AVARDA score for the same virus were associated with the candidate factor in the same direction, interpreting this as evidence of a true association; (ii) Z-scores for several peptides from a given virus were associated with the candidate factor in the same direction, but the AVARDA score was not, interpreting this as evidence of a potential false positive due to cross-reactivity; and (iii) Z-scores for several peptides from a given virus were associated with the candidate factor in opposite directions, but the AVARDA score showed no association, interpreting this as evidence of true associations obscured by opposite epitope-specific effects. **We have also clarified how the AVARDA breadth score is computed (l. 790-804).**

We hope that these additions make our rationale and the value of jointly analyzing peptide-level and virus-level aggregated scores from PhIP-seq data more transparent.

4. The data on Saliviruses and FUT2 is a compelling output of the technology as a path to discover new associations. It may not be feasible for a revision of the draft, but it would have been great to see a proof of usage of the receptor. However, it is not completely clear whether the virus has been isolated.

We thank the reviewer for this inspiring comment. Nevertheless, we agree that such experiments fall beyond the scope of the present study and would be difficult to implement, particularly in the absence of an isolated virus.

5. The genome association analyses are conceptually similar to those described as genome-to-genome studies (original publication <https://elifesciences.org/articles/01123>). It would be worth commenting on the shared methodologies and concepts.

We now refer to genome-to-genome association studies and highlight how such approaches could complement human viral exposome studies like ours. **This is presented in the main text (l. 424).**

6. Thanks for the discussion on limits of the study as well as for the release of data and code and the UI

We warmly thank the reviewer for highlighting our efforts to transparently discuss the limitations of our study, and to make the code and data as accessible and easy to use as possible.

Point by point responses to the Reviewers' comments

REVIEWER #1

The authors have addressed all of my comments and have considerably strengthened the manuscript.

RESPONSE: We thank the reviewer for their positive feedback. We are pleased to note that the reviewer finds the revised manuscript to be considerably strengthened.

REVIEWER #2

In the revisions of this manuscript, the authors clarified the nature of their data sharing plans, and attempted to validate their main conclusions from the prior manuscript, using a Luminex assay with intact proteins. They also evaluated the impact of batch effects on their results.

RESPONSE: We thank the reviewer for their very careful revision of our revised manuscript.

The authors' description of their data sharing plans and the constraints imposed by regulatory requirements have alleviated my prior concerns.

RESPONSE: We are pleased to note that the reviewer now considers our data sharing plan justified.

Major concerns persist in the revised manuscript. In the response to reviews and description of new experiments attempting to validate the VirScan results using Luminex or ELISA assays, the authors state:

“Specifically, we confirmed that:

(i) antibody titers against the less conserved globular head (β Luminex = -0.033 ; P Luminex = 4.42×10^{-76}) and the more conserved stalk (β Luminex = 0.014 ; P Luminex = 8.62×10^{-15}) domains of IAV hemagglutinin change in opposite directions with age;

(ii) antibody titers against less conserved positions 200-250 of the IAV M1 protein decrease with age (β Luminex = -0.0068 , P Luminex = 4.04×10^{-4});

(iii) antibodies preferentially target the IAV HA protein in females (β Luminex = -0.10 ; P Luminex = 0.038), whereas antibodies in males target M1 (β Luminex = 0.092 ; P Luminex = 0.02) and NP 2 (β Luminex = 0.10 ; P Luminex = 0.03) proteins from IAV and IBV, respectively;

(iv) antibodies of AFB preferentially target EBNA-4 peptides deriving from the AG876 strain ($\beta = -1.28$, $P = 9.12 \times 10^{-14}$) whereas antibodies of EUB target EBNA-6 peptides from the B95-8 strain ($\beta = 1.02$, $P = 8.2 \times 10^{-13}$).”

Unfortunately, several aspects of these new results don't increase confidence in the reliability or significance of many of the findings of the study.

One of the stronger conclusions in the original manuscript was that smoking was associated with differences in titers against rhinovirus peptides. The authors' experiment to further test this result did not reproduce it, and an explanation is offered that this is because testing of exactly the peptides that gave the original result in the VirScan data was not possible due to time constraints. The validation assay used viral lysates as the antigen source, and it is not obvious why this kind of assay wouldn't be expected to correlate with rhinovirus exposure, infection history and serological responses. If the original result was indeed restricted to a very few peptides, it is unclear what the immunological or clinical significance of this finding could be.

RESPONSE: We agree that the lack of replication in the validation assay may raise questions about the significance of our findings. However, we would like to clarify the following:

(i) The association between smoking and anti-rhinovirus antibody reactivity is supported by prior evidence using a custom PhIP-seq library in 1,443 individuals from the Lifelines-DEEP cohort (Andreu-Sanchez et al., *Immunity* 2023). Notably, the five peptides most strongly associated with smoking in our MI cohort – derived from a rhinovirus polyprotein containing capsid proteins ($\beta > 0.040$, $P_{\text{adj}} < 1.0 \times 10^{-10}$) – included the peptide most associated with smoking status in the LifeLines-DEEP cohort ($\beta = 1.09$, $P_{\text{adj}} = 3.29 \times 10^{-9}$), confirming that these signals are both genuine and peptide-specific;

(ii) Among the rhinovirus peptides included in our VirScan library, 29% show associations with smoking, which explains why peptide-specific associations are not replicated using a viral lysate in the Luminex assay. Smoking effects on antibodies against a viral lysate can be seen as an average of these effects on antibody titers against many individual epitopes; peptide-specific effects are therefore diluted (or lost) when antibodies targeting other epitopes are unaffected. In addition, preparation of viral lysates can alter epitope structure and antibody recognition, further contributing to differences between assays;

(iii) Our results do not support the notion that smoking affects only a few peptides, as the proportion of rhinovirus peptides showing associations with smoking is substantial (29%);

(iv) The main objective of our study is precisely to discover peptide-specific effects that are missed by conventional approaches; and

(v) Assessing the clinical significance of these findings, for example by assessing whether antibodies against the 63 smoking-associated rhinovirus peptides are neutralizing, would require long experiments that, we believe, fall outside the scope of this resource-focused study.

To address the Reviewer's concerns, we have now revised the manuscript to (i) explicitly state that our findings are consistent with a previous study (Andreu-Sanchez et al., *Immunity* 2023); (ii) report the proportion of rhinovirus peptides targeted by smoking-associated antibodies to strengthen the immunological significance of our findings; and (iii) explain more clearly why lysate-based measurements do not reproduce peptide-specific associations (l. 257-265; l. 398).

For the attempted confirmations of the VirScan results with ELISA or Luminex assays, the authors list in Suppl. Table 1 these influenza antigens used in the Luminex panel to assess for age-related titer changes:

- Chimeric protein comprising of A/White-fronted Goose/Netherlands/21/1999 (H6HA head) and A/Puerto Rico/8/1934 (H1HA stalk) (Krammer et al., *J Virol* 2013)
- Influenza A H3N2 (A/Beijing/32/1992) Hemagglutinin / HA1 Protein (His Tag)

For point (i), testing based on this limited number of antigens (and the additional influenza peptides listed in Suppl. Table 1) is not clearly interpretable or informative for assessing whether there is an age association with antibody titers binding to the head vs. stalk of influenza HA. The head vs. stalk antigens in the validation assays come from completely different influenza virus types (an H1 stalk from 1934 in the chimeric H6 head/H1 stalk protein, being compared to an H3 HA1 from 1992). There is extensive literature about

imprinting effects in influenza antibody responses, related to the birth year and age of research participants, which can differ between HA types (particularly between group 1 viruses including H1 and H2 types, and group 2 viruses including H3 types), and which can result in individuals in particular age ranges differing in their antibody titers to particular H1 or H3 antigens, not necessarily related to differences in head vs. stalk binding titers. The experiments in the revised manuscript don't really resolve these complexities, and don't support general conclusions about age-related changes in head vs. stalk antigen domains. The sentence in the abstract "We demonstrate that age, sex, and continent of birth extensively affect not only the viruses but also the specific viral epitopes targeted by the antibody repertoire" does not seem well-supported by these data in relation to age effects on influenza epitope binding antibodies.

RESPONSE: We agree with the reviewer that imprinting effects can differ substantially between HA types, and that our conclusions would be best supported by comparing age effects on antibody levels against the stalk and head from the same IAV virus. To address this point, we have now compared antibody reactivity to both HA domains from the same IAV strain, focusing on H3N2 and H1N1 subtypes, quantified by Luminex and/or VirScan assays. Supporting our initial conclusions, we found that (i) antibodies targeting the hemagglutinin head and stalk of a H3N2 strain from 1975 strongly decrease and increase with age, respectively, particularly among individuals more than 40 years of age (HA head: $\beta_{\text{Luminex}} = -0.035$, $P_{\text{Luminex}} = 3.50 \times 10^{-21}$; stalk: $\beta_{\text{VirScan}} = 0.019$, $P_{\text{VirScan}} = 1.34 \times 10^{-14}$); and (ii) antibodies targeting the head and stalk of H1N1 from 1934 slightly decrease and strongly increase with age, respectively (head: $\beta_{\text{Luminex}} = -0.012$, $P_{\text{Luminex}} = 0.013$; stalk: $\beta_{\text{Luminex}} = 0.014$, $P_{\text{Luminex}} = 8.62 \times 10^{-15}$).

We thank the reviewer for this constructive suggestion, which has strengthened our conclusions. We have now incorporated these new results into the revised manuscript (l. 173-179; l. 375-377).

For point (ii), similar concerns apply to age effects on the influenza M1 antibody analysis, where the validation experiments for age effects include only a single peptide from a particular M1 sequence, and it is not clear how many M1 sequences from different historical viruses are represented in the original VirScan result.

RESPONSE: We agree with the reviewer that systematic experimental validation of VirScan-based findings would strengthen our conclusions. However, validating all 27 associations between age and anti-M1 antibody levels with Luminex assays would require several additional months of experimental work. We would like to emphasize here that the VirScan approach has been widely used in multiple studies where no or limited experimental validation was performed (Mina et al., *Science* 2019; Consiglio et al., *Cell* 2020; Shrock et al., *Science* 2020; Venkataraman et al., *Immunity* 2022; Andreu-Sanchez et al., *Immunity* 2023). In line with these studies, the main objective of our work is not to re-establish the accuracy of PhIP-seq measurements, which is now well demonstrated, but to show that this method uncovers peptide-specific associations that are typically missed by conventional assays.

To address the reviewer's comment, we have now explored further the VirScan data by comparing anti-M1 antibodies targeting positions 150-200 or 200-250 from the same IAV strains. These new analyses confirm, again, our initial conclusions; for example, antibodies against peptides at positions 150-200 and 200-250 from the H1N1 Jamesburg/1942 strain

increase and decrease with age, respectively (150-200 region: $\beta_{\text{VirScan}} = 0.013$; $P_{\text{VirScan}} = 2.4 \times 10^{-5}$; 200-250 region: $\beta_{\text{VirScan}} = -0.015$; $P_{\text{VirScan}} = 2.4 \times 10^{-7}$).

We thank the reviewer for this suggestion, which provides more direct evidence that strain differences do not explain the opposite age effects observed on antibodies targeting the two M1 domains. These new results have been incorporated into the revised manuscript (l. 179-181).

In (iii), effects of sex on antibody titers to influenza HA versus M1, these results are difficult to interpret without knowledge of what fraction of males versus females were vaccinated, and how recently. Increased female antibody responses to influenza vaccination (primarily increasing HA titers) compared to males are well documented (summarized in Cook, Vaccine 2008, PMID 185244330). Decreased female severity of influenza infection is also reported, and may be causally related to higher HA titers prior to infection. Males could have higher antibody responses to M1 antigen due to having more severe infections. But the accuracy of measurement of M1 antibodies with the VirScan platform needs to be documented further, as older literature (summarized in Krammer, Nat Rev Immunol, 19, pages 383–397 (2019)) reports low rates of seropositivity of individuals for M1 antibodies, and low seroconversion after known infection. The revised manuscript doesn't add clarity or fully address these topics.

RESPONSE: We would like to emphasize that the proportions of vaccinated females and males were already reported in the previous version of the manuscript (20.2% vs. 18.6%, respectively; l. 197). As these proportions do not differ significantly, vaccination status cannot account for the sex differences in antibody levels observed in our study. We agree with the reviewer's interpretation that observed sex differences in anti-M1 antibodies may reflect differences in infection severity (since M1 is immunogenic in natural infection only), a point we had cautiously mentioned in the previous version of the manuscript (l. 378-380).

Regarding previous studies on M1 seropositivity, direct comparisons with our results were not initially possible because we did not compute serostatus from VirScan Z-scores, which requires defining a threshold for seropositivity. To address this point, we have now analyzed VirScan Z-scores for the H1N1 and H3N2 M1 peptides most associated with sex. We fitted a two-component Gaussian mixture to the non-transformed Z-scores, and obtained distributions that discriminate seronegative from seropositive donors (**Fig. A** for the reviewer here below). Using the 95% percentile of the first distribution as the threshold for seropositivity, we estimated that 11.6% of females and 17.9% of males were seropositive for H1N1 M1 ($P = 0.0085$), and 10.0% and 16.1% for H3N2 M1 ($P = 0.0077$), respectively. These estimates are consistent with those obtained with conventional serological assays: 6% of individuals develop anti-M1 antibodies after a unique experimental infection by H3N2 (Cretescu et al., *Infect Immun* 1978).

Fig. A. Two-component Gaussian mixtures fitted on VirScan antibody reactivity Z-scores, for sex-associated peptides from (a) H1N1 and (b) H3N2 M1 proteins. The gray and red dashed lines indicate the two fitted Gaussian distributions. The black vertical line and red bars indicate the seropositivity threshold and individuals inferred as seropositive.

These additional analyses further confirm the reliability of our PhIP-seq-based approach. We have now incorporated these results in the new version of the revised manuscript (1. 199-201).

In (iv) the effects of continent of birth on herpesvirus titers are convincing and consistent with other literature, and the EBNA peptide results are consistent with the geographical distribution of the strains of virus from which they are derived, and are validated by the additional Luminex experiments.

RESPONSE: We appreciate the reviewer’s positive assessment of our results on continent-of-birth effects.

The authors also evaluated the extent to which batch effects may have affected the results due to confounding with participant demographics.

For the MI cohort, they note:

“If biological effects are confounded by batch effects, we expect the two proportions to be correlated. However, we found weak correlations between the two statistics: antibody levels with the largest biological effects typically showed limited proportions of variance explained by batch effects, and vice versa (Pearson’s coefficient $r_{Age} = 0.042$, Supplementary Fig. 1c; $r_{Sex} = -0.0063$, Supplementary Fig. 1e). Notably, peptides largely explained by batch effects were enriched for peptides attributed to the viral family ‘Others’, which includes several uncommon viruses, raising questions about the biological relevance of these peptides (Supplementary Fig. c,e).”

For the EIP cohort, they note:

“We estimated the proportion of variance explained by batches in EUB only ($n = 212$ out of 312 EIP samples, distributed on 4 out of 5 batches), which should estimate solely batch effects. Reassuringly, the correlation between the variance explained by batches in the uncorrected EUB data and that explained by continent of birth in the full batch-corrected data was low ($r_{Continent} = 0.043$; Supplementary Fig. 1i).”

However, it is not clear in the revised analysis whether the magnitude of the batch effect correlations with biological effects are much smaller than the correlations between biological variables (such as age or sex) and the VirScan peptide antibody measurements. In Suppl Fig. 1c, the percentage of variation explained by batch effects is plotted on a different, compressed

axis scale compared to the percentage of variation explained by age, making the age effects appear more substantial and making the batch effects appear diminished. An even greater scale mismatch is used in Suppl Fig. 1e for the sex vs. batch effect comparison. The serological associations with continent of origin compared to batch effects show a more modest impact of batch effects, and the authors have plotted these on the same axis scale. Here there is less concern about artifacts dominating the results.

RESPONSE: The reviewer is concerned that age and sex effects are weak compared to the batch effects observed in the MI data. However, we would like to clarify that both age or sex explain up to 5-10% of the variance in antibody reactivity, which is in fact larger than what was previously estimated in this cohort for immune cell composition, immune-gene expression, and cytokine production (Patin et al., *Nat Immunol* 2018; Piasecka et al., *PNAS* 2018; Saint-André et al., *Nature* 2024). Furthermore, we also emphasize that batch effects on VirScan Z-scores are reduced to zero after correction and therefore do not impact any of our results, provided they do not confound true biological effects. To verify this, we have computed the correlation between batch effects before correction and age or sex effects after correction and found, reassuringly, this correlation to be low (Suppl. Fig. 1c,e).

To address further the reviewer’s concern, we have now verified that our results regarding age and sex effects on the antibody repertoire are unchanged after excluding peptides for which the variance in antibody reactivity explained by batch effects exceeds 20%. These new analyses further demonstrate that our main findings are not based on Z-scores strongly affected by batch effects (**Figs. B and C** for the reviewer here below).

Fig. B. Age effects on the epitope-specific antiviral antibody repertoire, after excluding the peptide Z-scores most associated with batch effects. $-\log_{10}(\text{adjusted } P\text{-values})$ and direction of associations between all public peptide Z-scores and age in the MI cohort, by viral species. The dashed gray vertical lines indicate viruses for which the AVARDA breadth score is significantly associated with age. The dashed red horizontal lines indicate the significance threshold ($P_{\text{adj}} < 0.05$). Peptides for which the variance in antibody reactivity explained by batch effects exceeds 20% (Suppl. Fig. 1c) were excluded.

Fig. C. Sex differences in the antiviral antibody repertoire, after excluding the peptide Z-scores most associated with batch effects. $-\log_{10}(\text{adjusted } P\text{-values})$ and direction of associations between all public peptide Z-scores and sex in the MI cohort, by viral species. The dashed red horizontal lines indicate the significance threshold ($P_{\text{adj}} < 0.05$). Peptides for which the variance in antibody reactivity explained by batch effects exceeds 20% (Suppl. Fig. 1e) were excluded.

The other concerns raised in points 5-8 of the review have been addressed adequately in the revised manuscript, or clarify that key data such as vaccination timing are not available.

RESPONSE: We are pleased to note that the reviewer considers that all other concerns have been adequately addressed.

REVIEWER #3

Thanks for the careful revision of the manuscript.

RESPONSE: We thank the reviewer for their positive feedback. We are pleased to note that the reviewer finds the new manuscript to be carefully revised.